# ROR2 regulates self-renewal and maintenance of hair follicle stem cells

Anthony Veltri [1,2], Christopher M. R. Lang [1,2], Gaia Cangiotti [1], Chim Kei Chan [1] & Wen-Hui Lien [1] ✉

Hair follicles undergo cycles of regeneration fueled by hair follicle stem cells (HFSCs). While β-catenin-dependent canonical Wnt signaling has been extensively studied and implicated in HFSC activation and fate determination, very little is known about the function of β-catenin-independent Wnt signaling in HFSCs. In this study, we investigate the functional role of ROR2, a Wnt receptor, in HFSCs. By analyzing *Ror2*-depleted HFSCs, we uncover that ROR2 is not only essential to regulate Wnt-activated signaling that is responsible for HFSC activation and self-renewal, but it is also required to maintain proper ATM/ATR-dependent DNA damage response, which is indispensable for the long-term maintenance of HFSCs. In analyzing HFSCs lacking β-catenin, we identify a compensatory role of ROR2-PKC signaling in protecting *β-catenin*-null HFSCs from the loss of stem cell pool. Collectively, our study unveils a previously unrecognized role of ROR2 in regulation of stem cell self-renewal and maintenance.

In mammals, Wnt signaling functions in tissue morphogenesis, stem cell activation, and tumor development[1,2]. The binding of secreted Wnt ligands to receptors and/or co-receptors initiates diverse signaling cascades that can be divided into β-catenin-dependent canonical and β-catenin-independent non-canonical Wnt signaling pathways[1]. These pathways might act independently or cooperatively to orchestrate various cellular functions.

Canonical Wnt signaling (referred as Wnt/β-catenin signaling) is activated when a Wnt ligand binds to Frizzled (Fzd) and LRP-5/6, which triggers activation of Dishevelled (Dvl) proteins, leading to the inhibition of the destruction complex, composed by Axin, caseine kinase 1α (CK1α), adenomatous polyposis coli (APC) and glycogen synthase kinase 3β (GSK3β), thereby stabilizing β-catenin[3,4]. Stabilized β-catenin protein then translocates to the nucleus where it binds to lymphoid-enhancing factor/T-cell factor (LEF/TCF) proteins to activate target gene expression[5]. Unlike Wnt/β-catenin signaling, β-catenin-independent Wnt pathways involve multiple intracellular signaling cascades that might be cross-connected. The induction of non-canonical Wnt signaling may trigger the release of intracellular calcium, which in turn activates downstream protein kinases, such as calcium/calmodulin-dependent protein kinase II

(CaMKII) and protein kinase C (PKC)[6-8]. Non-canonical Wnt signaling can also be transduced via Rho family of the small GTPases, which activate c-Jun N-terminal kinase (JNK) and the downstream activating protein-1 (AP-1) complex for transcriptional regulation, or directly modulate cytoskeleton organization that orchestrates planar cell polarity (PCP) and cell migration[9-13].

Receptor tyrosine kinase-like orphan receptor 2 (ROR2) was initially identified along with ROR1 as a tyrosine kinase of the Trk family[14], and then recognized as one of Wnt (co)receptors due to its ability to interact with non-canonical Wnts, including Wnt4, Wnt5a, and Wnt11[15,16]. Genetic studies show that *Ror2*^−/− mice displayed striking similarities to *Wnt5a*^−/− mice, suggesting they may function in the same signaling pathway[10,17]. In vertebrates, ROR2 is required for Wnt5a-induced cell migration, a function that involves activation of JNK, PKC, actin-binding protein Filamin A and Rho-family of the GTPase[17-21]. The interaction of Wnt-ROR2 leads to phosphorylation of Dvl that induces the activation of AP-1 and Rac1[22,23]. In addition, ROR2 was shown to interact with and be phosphorylated by CK1and GSK3, both of kinases that also play imperative roles in Wnt/β-catenin signaling[20,24-26]. In multiple systems, Wnt5a was shown to inhibit β-catenin-mediated canonical Wnt signaling[27-29]. The nature by which ROR2 mediates

[1]de Duve Institute, Université catholique de Louvain, 1200 Brussels, Belgium. [2]These authors contributed equally: Anthony Veltri, Christopher M. R. Lang. ✉e-mail: wen-hui.lien@uclouvain.be

Wnt5a-dependent antagonism of Wnt/β-catenin signaling remains controversial. Under certain circumstances, Wnt5a inhibits Wnt3a-induced β-catenin signaling via ROR2[22,30–33]; in others, ROR2 is not required for this inhibition[10,26,34]. In contrast, ROR2 has also been reported to potentiate Wnt/β-catenin signaling. In osteosarcoma cells, ROR2 enhances the transcriptional response to Wnt1[35]; in lung carcinoma cells, ROR2 activates Wnt3a-induced canonical Wnt signaling as a co-receptor with Fzd2[31]. Overexpressing ROR2 enhances β-catenin-mediated transcription; conversely, knocking down ROR2 decreases it in renal cancer cells[36]. It is noteworthy that studies that analyzed the effect of ROR2 in Wnt/β-catenin signaling have mostly depended on protein overexpression and the reporter assay for Wnt/β-catenin signaling activity. The physiological effect of ROR2 in Wnt signaling activities requires further investigation.

All cells in the body are exposed to DNA damage caused by exogenous factors, including mutagens, or endogenous processes, such as oxidative stress. Thus, in order to maintain genomic integrity that ensures tissue homeostasis and prevents the development of deleterious diseases such as cancer, DNA repair mediated by DNA damage response (DDR) is of vital importance, particularly for adult stem cells (SCs) as they persist for extended periods in adult tissues[37,38]. Ataxia Telangiectasia Mutated (ATM) and ATM- and Rad3-related (ATR) are DDR kinases that are primarily activated by genomic damage; however, the functions of ATM and ATR in DNA damage specificities are distinct[39,40]. In response to DNA damage, ATM and ATR get activated and phosphorylate downstream signaling proteins, such as checkpoint kinase 2 (CHK2) for ATM, and checkpoint kinase 1 (CHK1) for ATR, thereby regulating cell cycle checkpoints, DNA repair or apoptosis[39–41]. Independently from the DDR, both ATM and ATR can also be activated by the excess production of reactive oxygen species (ROS), which could be caused by imbalanced oxidative metabolism and/or accumulation of unrepaired DNA damage[42–45]. In response to ROS, ATM induces activation of 5′ AMP-activated protein kinase (AMPK)[43,46] and nuclear factor erythroid 2-related factor 2 (NRF2)[47,48] to modulate oxidative metabolism and induce antioxidant response to combat ROS, respectively. Deficiency in ATM-dependent DDR in SCs can lead to impaired self-renewal or apoptosis that could influence the numbers of stem cells and their functionality[49–51].

Hair follicle (HF) is an excellent model system to study the underlying mechanisms that regulate stem cell activation and maintenance[52]. Mature HFs progress through cycles of growth (anagen), degeneration (catagen), and then rest (telogen)[53,54]. Hair follicle stem cells (HFSCs), located at bulge of the resting HF, remain quiescent during telogen phase of the hair cycle. At the onset of anagen, some of HFSCs become proliferative and migrate downward to replenish the lower HF[55]. The activation of HFSCs is tightly regulated by micro-environmental signals coming from their niche cells[56–61]. At anagen onset, upregulated Wnt/β-catenin signaling directs transcriptional regulation that governs HFSC activation[62–64]. HFSCs lacking β-catenin are unable to undergo hair regeneration[62,63,65], but they can be maintained in their native niche without losing HFSC identity[63]. However, it remains unclear what is the mechanism that maintains β-catenin-null HFSC pool in a rich milieu where neighboring cells secrete various signal molecules along the hair cycle.

In this study, we identify a surprising role for ROR2 in regulation of stem cell self-renewal and maintenance. Using a genetic mouse model, we showed that depletion of Ror2 in HFSCs caused a delay in HFSC activation, downregulation of HFSC stemness genes, and eventually led to loss of a HFSC population. By depleting Ror2 in primary cultured HFSCs, we uncovered that ROR2 is essential for the activation of Wnt-induced signaling and ATM/ATR-dependent DDR, thereby regulating HFSC migration, proliferation, and maintenance. Lastly, in analyzing HFSCs lacking β-catenin, we identified the necessity of ROR2 and the downstream PKC in protecting β-catenin-null HFSCs from the loss of stem cell pool and wrong fate differentiation. Together our results reveal that ROR2 not only regulates Wnt signaling that governs HFSC activation and migration, but also serves as a protective mechanism to ensure long-term maintenance of HFSCs.

## Results

### Deletion of *Ror2* in the bulge results in a delay of HFSC activation and the loss of a HFSC population

While HFSCs reside in a niche expressing Wnt ligands throughout the hair cycle[66], very little is known about the function of Wnt signaling in HFSCs apart from β-catenin-dependent regulation. To tackle this unsolved question, we focused on a Wnt receptor, ROR2, which is capable to transduce β-catenin-dependent and -independent Wnt signals[17–21,31,35]. We first examined the expression of ROR2 in HFs using immunostaining of whole-mount skin. As shown, ROR2 is fairly well expressed in HFSCs and neighboring cells in the bulge (Bu) but relatively low in the secondary hair germ (HG) (Fig. 1a and Supplementary Fig. 1a). Notably, ROR2 expression in the bulge is higher in HFs at anagen onset than those in telogen (Supplementary Fig. 1b). By purifying integrin α6$^{high}$/CD34$^+$ HFSCs from mouse HFs at telogen (referred as quiescent HFSC; qHFSC) and anagen onset (referred as activated HFSC; aHFSCs) using fluorescence-activated cell sorting (FACS), we confirmed that ROR2 protein level was elevated in aHFSCs as compared to qHFSCs (Fig. 1b and Supplementary Fig. 1c). To determine if the upregulated ROR2 is functionally significant for HFSC activation, we generated *K15CrePGR;Ror2$^{fl/fl}$;ROSA26$^{LSL−YFP}$* mice by crossing mouse line carrying inducible Cre recombinase driven by K15 promoter (*K15CrePGR*) to the mice carrying conditional *Ror2* alleles (*Ror2$^{fl/fl}$*) and a Cre-activated YFP reporter (*ROSA26$^{LSL−YFP}$*), and then depleted *Ror2* gene in HFSCs and their progenies at postnatal (P) days 18–25 (Fig. 1c). We observed that ROR2 expression was compromised in FACS-purified *Ror2* conditional knockout (*Ror2* cKO) HFSCs (Fig. 1d and Supplementary Fig. 1d), as well as diminished in the bulge of *Ror2* cKO HFs while ROR1 remained fairly expressed (Supplementary Fig. 1e). By analyzing the hair cycle progression, we found that *Ror2* cKO HFs showed a delay in anagen entry as compared to control (*Ror2* Ctrl) HFs (Fig. 1e). The delay in anagen entry was continuously detected in the following hair cycle at anagen onset as evidenced by delayed anagen entry of the *Ror2* cKO HF and hair coat recovery of the *Ror2* cKO animal (Supplementary Fig. 1f). To address whether the delay of anagen entry in *Ror2* cKO HFs was caused by a defect in HFSC activation, we performed 24-hour administration of EdU to mice at anagen onset. Immunostaining of *Ror2* Ctrl and cKO skin for CD34 and EdU showed a significant decline in EdU+ proliferative HFSCs in YFP+ bulge of *Ror2* cKO HFs (Fig. 1f), suggesting that *Ror2* cKO HFSCs displayed reduced cell proliferation.

To further verify the role of ROR2 in HFSC activation, we also analyzed the effect of *Ror2* depletion in HFSCs that were synchronized and activated by hair depilation at the 2nd telogen phase (Supplementary Fig. 2a). Histology analysis and EdU labeling experiments with depilated *Ror2* Ctrl and cKO skin showed a delay in depilation-induced HFSC activation and anagen entry in *Ror2* cKO skin (Fig. 2a and Supplementary Fig. 2b), which is in accordance with what we found in hair cycling during homeostasis (Fig. 1e, f). The HFSC proliferation defect could be recapitulated in FACS-purified HFSCs from *Ror2* Ctrl and cKO HFs at anagen onset. Specifically, when primary HFSCs were cultured with fibroblast feeder cells in the medium that promotes HFSC proliferation, *Ror2* cKO HFSCs exhibited fewer colony numbers than the Ctrl HFSCs (Fig. 2b), indicating an impaired ability of *Ror2* cKO HFSCs to proliferate. Given that Wnt/β-catenin signaling is essential for HFSC activation and that over-expressed ROR2 is capable to transduce canonical Wnt signaling in the cultured cells[31,35], the delayed activation of *Ror2* cKO HFSCs could be partly caused by the impairment in β-catenin-dependent Wnt signaling activity. To address this possibility, we examined the

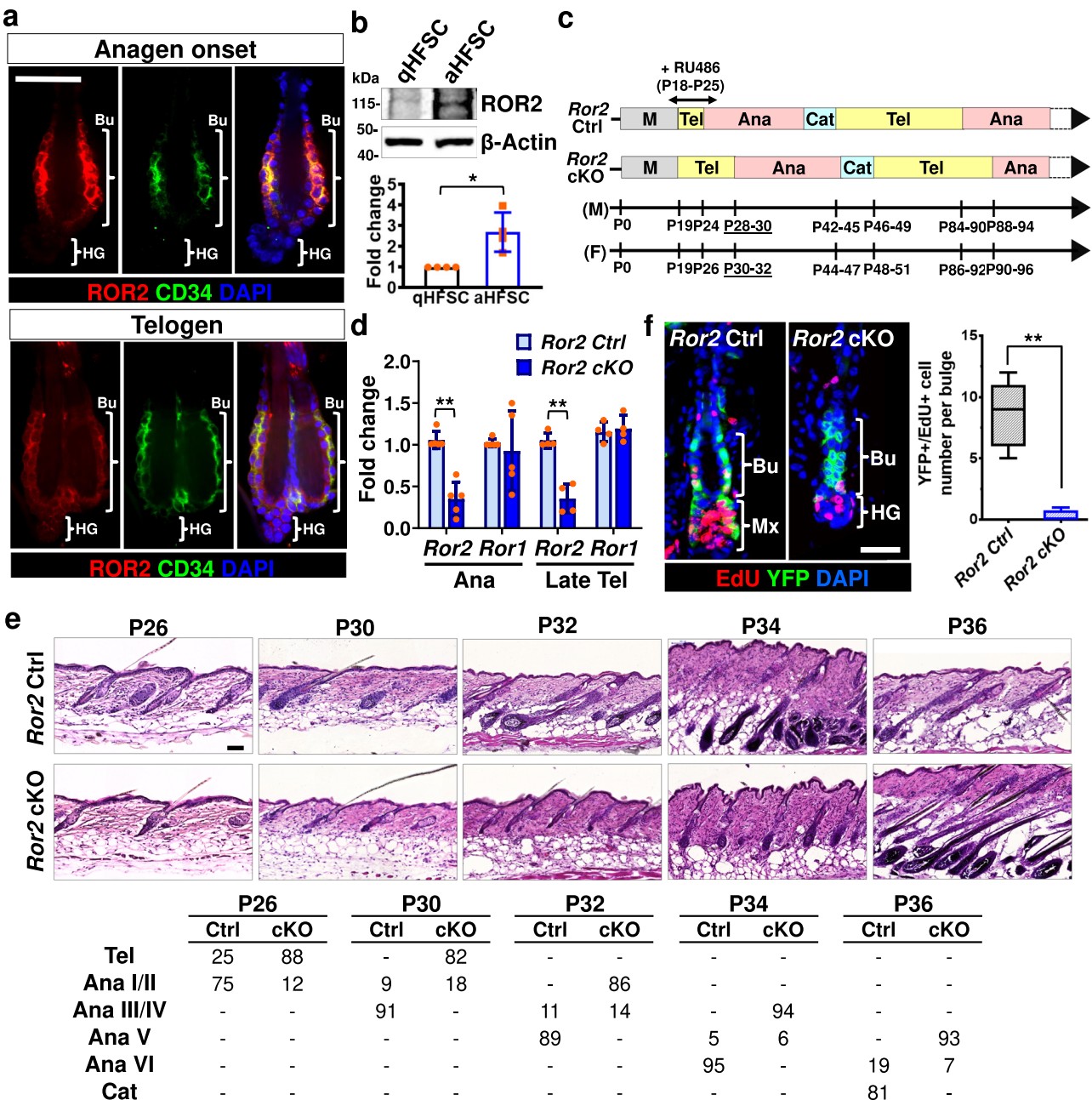

**Fig. 1 | *Ror2* depletion in bulge causes a delay of HFSC activation and hair cycle progression. a** Whole-mount immunofluorescence staining of mouse HFs at anagen onset and telogen for ROR2 (red) and CD34 (green). Bu, bulge; HG, hair germ. **b** ROR2 expression is upregulated in activated HFSCs. Immunoblotting analyses of FACS-purified HFSCs from P55 (for quiescent HFSC; qHFSC) and P22-23 (for activated HFSC; aHFSC) mouse back skins for ROR2 and β-Actin. Quantification of ROR2 protein levels from independent experiments is shown as the graph below. Data were normalized to β-Actin. Values were calculated by comparing to qHFSC and reported as average ± s.d.; *n* = 4 blots from independent FACS-sorted HFSCs; *p = 0.0117. **c** Schematic diagram illustrating delayed hair cycle entry in *Ror2* cKO mice as compared to their Ctrl littermates. Mice were treated with RU486 to activate Cre recombinase in HFSCs at P18-25. M morphogenesis, Tel telogen, Ana anagen, Cat catagen, M males, F females. **d** Real-time PCR analyses with FACS-purified *Ror2* Ctrl and cKO HFSCs from anagen onset (Ana) and late telogen (Late Tel). Values were normalized to *Ror2* Ctrl HFSC mRNA. Data are reported as average ± s.d.; *n* = 5 (Ana) or 4 (Late Tel) biological independent animals; **p < 0.005. **e** H & E staining of *Ror2* Ctrl and cKO mouse back skins at indicated postnatal (P) days (top). Quantification of percentage of HFs at indicated hair cycle stages showing a delay of anagen entry of *Ror2* cKO HFs (bottom). **f** *Ror2* cKO HFSCs display a delay in activation at anagen onset. After 24 h EdU labeling, P32 female skins were immunostained (left) and quantified (right). Mx, matrix. Data are reported as the median (the line within the box), the 25th to 75th percentiles (bottom and top lines of the box), and the 10th to 90th percentiles (bottom and top whiskers); *n* = 7 (Ctrl) or 4 (cKO) HFs examined over one pair of animals; **p = 0.0031. All *p* values were calculated using unpaired two-sided *t*-test. Scale bars in **a**, **e**, **f** represent 50 μm. All results are representative of at least two independent experiments. Source data are provided as a Source Data file.

expression of Wnt/β-catenin target genes[63] in FACS-purified *Ror2* Ctrl and cKO HFSCs from 3 days post-depilated (3dpd) skin by performing quantitative reverse transcription PCR (RT-qPCR) analysis. As shown in Fig. 2c, the expression of genes that respond to β-catenin-

dependent Wnt signaling were significantly downregulated in depilation-activated *Ror2* cKO HFSCs. This downregulation was also detected in *Ror2* cKO HFSCs at anagen onset during hair cycling (Fig. 2d). These results suggest that the activation of Wnt/β-catenin

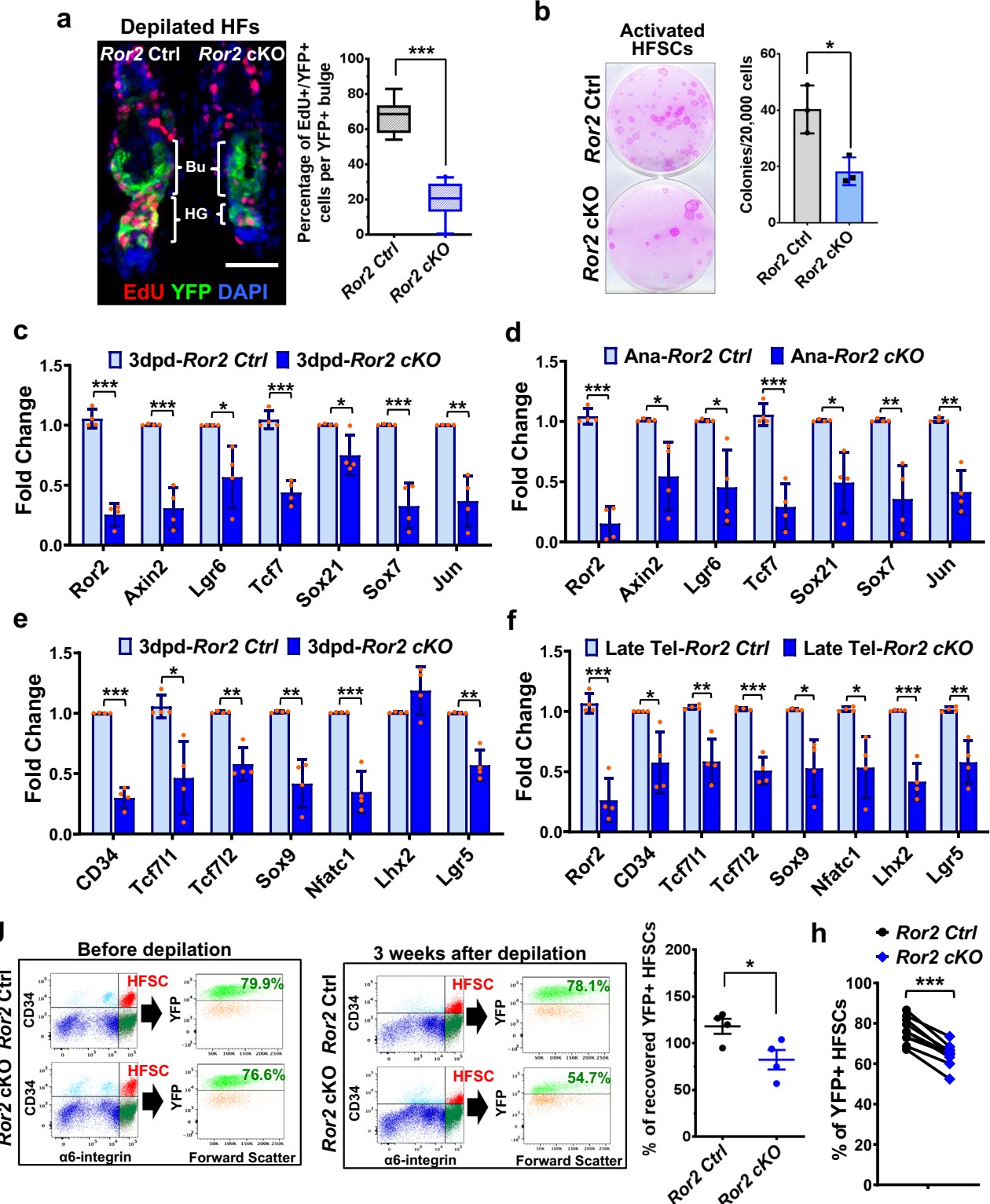

signaling that facilitates HFSC proliferation and anagen progression is dependent on ROR2, and the reduction in canonical Wnt signaling activity upon *Ror2* depletion leads to the delay of HFSC activation.

In addition to Wnt/β-catenin target genes, we also found that depilation-activated *Ror2* cKO HFSCs showed significant downregulation of a broad set of HFSC stemness genes as compared to their Ctrl HFSCs (Fig. 2e). The downregulation of these genes was also found in *Ror2* cKO HFSCs at late 2nd telogen when HFSCs were prepared for the activation (Fig. 2f). These results led us to postulate that ROR2 may

also be required for the maintenance of HFSC pool during HF regeneration. To test this hypothesis, we depilated the hair coat of mice and then used FACS to measure the percentage of YFP+ cells in integrin α6$^{high}$/CD34$^+$ HFSC population before depilation and 3 weeks after depilation when newly formed HFs returned to telogen. As shown in Fig. 2g, YFP+ *Ror2* Ctrl HFSCs were fully maintained in the stem cell niche after HF regeneration was completed, but only ~82% of YFP+ HFSCs were recovered from the previous *Ror2* cKO HFSC pool, suggesting that some *Ror2* cKO HFSCs were lost during HF regeneration. A

**Fig. 2 | *Ror2* depletion results in downregulation of Wnt/β-catenin target genes and stemness genes, which in turn lead to a delay of HFSC activation and the loss of a HFSC population. a** Depilation-activated *Ror2* cKO HFSCs display a delay in activation. Two days post-depilation and 24 h post-EdU labeling, depilated skins were immunostained (left) and quantified (right). Data are reported as the median (the line within the box), the 25th to 75th percentiles (bottom and top lines of the box) and the 10th to 90th percentiles (bottom and top whiskers); *n* = 8 (Ctrl) or 10 (cKO) HFs examined over one pair of animals; \*\*\**p* < 0.0001. Scale bar represents 50 μm. **b** Colony formation efficiency (CFE) of *Ror2* Ctrl and cKO HFSCs from anagen onset back skins. Colonies from FACS-purified HFSCs are stained with Rhodamine B (left). Quantification of CFE (right). Numbers of colonies which sizes ≥ 3 mm² were counted. Data are reported as average ± s.d.; *n* = 3 independent wells; \**p* = 0.0269. Results are representative of at least two independent experiments. **c, d** Real-time PCR analysis for Wnt/β-catenin target genes with FACS-purified *Ror2* Ctrl and cKO

HFSCs 3 days post-depilation (**c**) or at anagen onset (**d**). Values were normalized to *Ror2* Ctrl HFSC mRNA. **e, f** Real-time PCR analysis for HFSC stemness genes with FACS-purified *Ror2* Ctrl and cKO HFSCs 3 days post-depilation (**e**) or at late telogen (**f**). Data in **c–f** are reported as average ± s.d.; *n* = 4 biological independent animals; \**p* < 0.05, \*\**p* < 0.005, \*\*\**p* < 0.0005. **g** FACS analysis for YFP+ cells in integrin α6^high^/CD34⁺ HFSC population from *Ror2* Ctrl and cKO HFs before depilation (left) and 3 weeks after depilation (middle). Dot plot (right) shows percentages of recovered YFP+ HFSCs 3 weeks after depilation. Data are reported as mean ± SEM; *n* = 4 biological independent animals; \**p* = 0.0365. **h** FACS analysis for YFP+ cells in integrin α6^high^/CD34⁺ HFSC population from *Ror2* Ctrl and cKO HFs at the 2nd telogen. The paired dot plot shows reduced percentages of YFP+ HFSCs in *Ror2* cKO HFs versus *Ror2* Ctrl HFs. *n* = 10 independent pairs of littermates; \*\*\**p* = 0.0003. All *p* values were calculated using unpaired two-sided *t*-test. Source data are provided as a Source Data file.

---

similar phenomenon was detected in *Ror2* cKO HFSCs during hair cycling. By examining the percentage of YFP+ cells in α6^high^/CD34⁺ HFSC population of *Ror2* Ctrl and cKO HFs at the 2nd telogen phase, we discovered that *Ror2* cKO HFs showed lower percentage of YFP+ cells in their HFSC pools than the paired *Ror2* Ctrl HFs (Fig. 2h), indicating that some *Ror2* cKO HFSCs did not get recovered after hair cycling. However, we noticed that if any *Ror2* cKO HFSCs could survive through the 1st depilation, they were able to sustain the subsequent HF regeneration upon repetitive depilation (Supplementary Fig. 3a). This might explain why YFP+ *Ror2* cKO HFSCs could be fairly maintained in some aged HFs (Supplementary Fig. 3b). This result suggests that loss of ROR2 could be compensated by other factors in the recovered *Ror2* cKO HFSCs.

Altogether, our results implicate that ROR2 is required for HFSC activation via regulating Wnt/β-catenin signaling activity, and also for the maintenance of a HFSC population in vivo.

### Ror2 is essential for HFSC self-renewal and long-term maintenance

As we found that HFSC pool was reduced in *Ror2* cKO HFs during HF regeneration, we evaluated the capacity of HFSC maintenance by performing colony formation assays and followed by the long-term culture with FACS-purified HFSCs from *Ror2* Ctrl and cKO HFs at telogen. We found that telogen *Ror2* cKO HFSCs also displayed reduced colony numbers in culture medium promoting proliferation (Fig. 3a), indicating a defect in self-renewal ability. When single HFSC colonies that grew on a feeder layer were selected and serially passaged in culture, *Ror2* cKO HFSCs could not be maintained long-term (Fig. 3b). Moreover, upon removal from feeder cells, *Ror2* cKO HFSCs lost ability to proliferate and displayed a differentiated morphology (Fig. 3c). These differentiation-like *Ror2* cKO HFSCs not only expressed lower cyclin D1 (encoded by *Ccnd1*), corresponding to their lower proliferation, but also lost the expression of HFSC markers, *Cd34* and *Krt15* (Fig. 3d), implicating that ROR2 is essential to maintain HFSC identity. Taken together, the results from purified HFSCs in culture confirm our in vivo studies demonstrating that ROR2 plays a critical role in HFSC self-renewal and maintenance.

### ROR2 not only transduces non-canonical Wnt signaling and regulates HFSC migration, but also modulates Wnt/β-catenin signaling via regulating GSK3β stability

As purified *Ror2* cKO HFSCs could not be maintained in culture, we generated *Ror2*⁻/⁻ HFSCs by culturing HFSCs carrying conditional *Ror2* alleles (*Ror2*^fl/fl^) and YFP reporter (*ROSA26*^LSL−YFP^), and transducing them with lentiviral Cre recombinase in culture (Supplementary Fig. 4a). In this way, we were able to examine the alterations of signal transduction and gene expression in HFSCs upon loss of *Ror2*. We first confirmed the efficiency of *Ror2* depletion in HFSCs. As shown, the expression of ROR2 protein was completely abolished in Cre-expressing (*Ror2*⁻/⁻) HFSCs while the expression of ROR1 was getting

elevated in *Ror2*⁻/⁻ HFSCs during passaging (left panel, Fig. 4a). We then characterized *Ror2*⁻/⁻ HFSCs by examining their responses to Wnt stimulation. Previous studies have shown that ROR2 mediates Wnt5a-induced cell migration via activation of JNK, PKC, and small GTPases[17–21]. Here we examined the control (*Ror2*⁺/⁺) and *Ror2*⁻/⁻ HFSCs for Wnt5a-induced activation of JNK and PKC, as well as for the activities of Rac1 and Cdc42. By performing western blotting analysis with Wnt-stimulated HFSCs, we showed that Dvl2 was phosphorylated (shifted upper bands, arrowhead in the left panel of Fig. 4a) and downstream effectors JNK and PKC proteins of classical and novel subfamilies (referred as PKC in our study) were activated upon Wnt5a stimulation in *Ror2*⁺/⁺ HFSCs, but these activities were attenuated in *Ror2*⁻/⁻ HFSCs (+Wnt5a, left panel, Fig. 4a). Using an established small GTPase activation assay that employs glutathione S-transferase (GST) fusion proteins recognizing active forms (GTP-bound forms) of Rac1 and Cdc42, we found that regardless of Wnt5a stimulation, activities of Rac1 and Cdc42 were prominently reduced in *Ror2*⁻/⁻ HFSCs (Fig. 4b). Of note, although ROR1 was upregulated in cultured *Ror2*⁻/⁻ HFSCs, this upregulation was seemly insufficient to compensate for Wnt5a-ROR2-mediated signaling regulation. Together, these data demonstrate the essential role of ROR2 in transducing Wnt5a-dependent signaling in HFSCs.

Not only Wnt5a, but Wnt3a, a canonical Wnt ligand, could also activate JNK in HFSCs in a ROR2-independent manner (+Wnt3a, left panel, Fig. 4a). Intriguingly, *Ror2* depletion could prominently compromise Wnt3a-induced GSK3β inactivation and Axin1 degradation (+Wnt3a, right panel, Fig. 4a). As shown, both Wnt3a and Wnt5a stimulation could induce GSK3β inactivation, monitored by the increase in Ser-9 phosphorylation; however, this inactivation was abolished in *Ror2*⁻/⁻ HFSCs (right panel, Fig. 4a). Strikingly, the overall inactive form of GSK3β was lower than those in control HFSCs (right panel, Fig. 4a) while phosphorylation of GSK3β at Tyr-216 was not altered (Supplementary Fig. 4b). Since GSK3β is considered as a constitutively active kinase[67,68], reduction in the level of its inactive form suggests the elevation of GSK3β kinase activity. Indeed, the elevated activity of GSK3β in *Ror2*⁻/⁻ HFSCs was confirmed by an increase in p-β-catenin^S33/37/T41^ (right panel, Fig. 4a). An increase in p-LRP6^S1490^ is the other evidence of elevated GSK3β activity in *Ror2*⁻/⁻ HFSCs (right panel, Fig. 4a). In line with this observation, we also discovered that *Ror2*⁻/⁻ HFSCs displayed significant reduction in Wnt3a-induced expression of canonical Wnt target genes as compared to *Ror2*⁺/⁺ HFSCs (Fig. 4c). Inhibition of GSK3β activity by the treatment of CHIR-99021 (GSK3 inhibitor; GSK3i) increased the total pool of β-catenin (Supplementary Fig. 4c) and restored the expression of canonical Wnt target genes in *Ror2*⁻/⁻ HFSCs (Fig. 4d), further highlighting that the downregulation of Wnt/β-catenin signaling in *Ror2*⁻/⁻ HFSCs was caused by unleashed GSK3β activity that enhanced β-catenin degradation. These results support our in vivo finding showing that Wnt/β-catenin target genes were downregulated in activated *Ror2* cKO HFSCs (Fig. 2c, d), and suggest that downregulation of Wnt/β-catenin

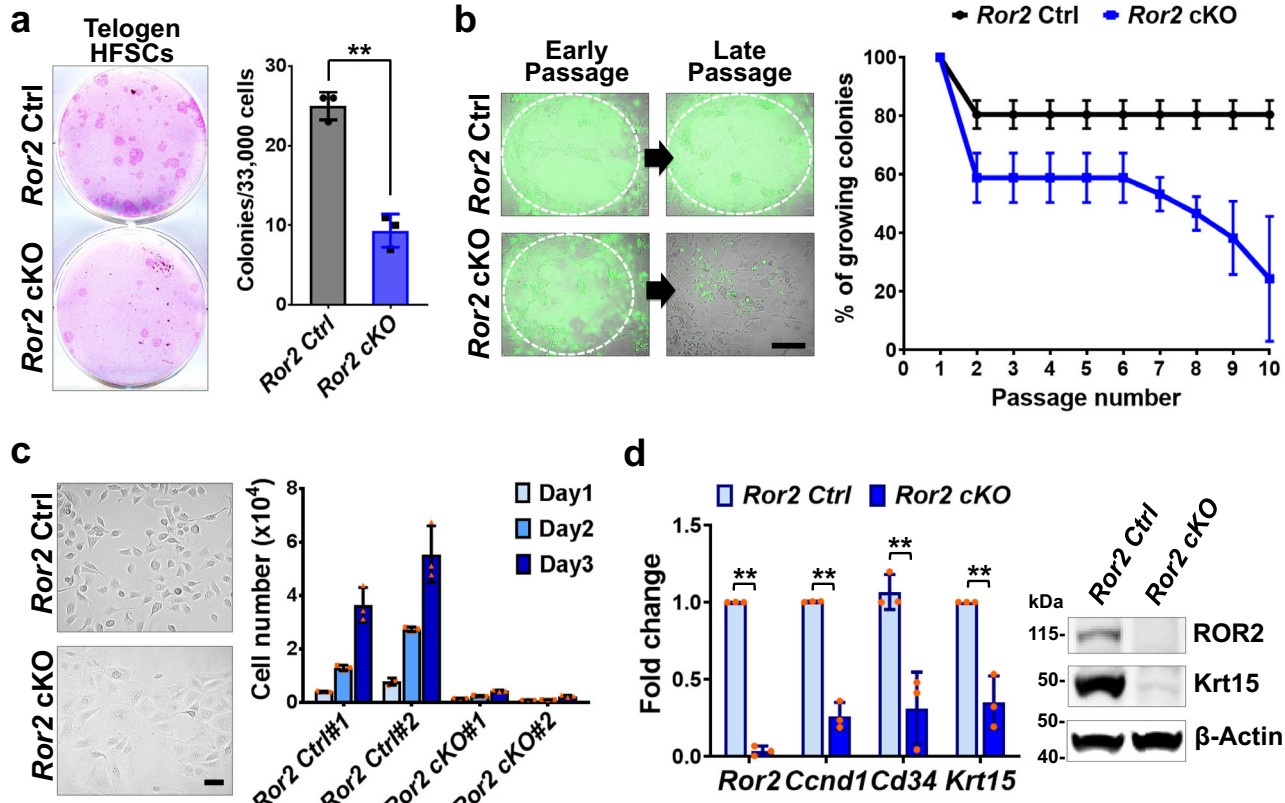

**Fig. 3 | ROR2 is required for HFSC self-renewal and long-term maintenance.**
**a** CFE of *Ror2* Ctrl and cKO HFSCs from telogen-phase back skins. Colonies from FACS-purified HFSCs are stained with Rhodamine B (left). Quantification of CFE (right). Numbers of colonies which sizes ≥ 3 mm² were counted. Data are reported as average ± s.d.; *n* = 3 independent wells; **p = 0.0007. Results are representative of at least two independent experiments. **b** *Ror2* cKO HFSCs could not be passaged and maintained for long-term. Individual *Ror2* Ctrl or cKO HFSC colonies were isolated, cultured and passaged. *Ror2* depletion was verified for individual colonies. (Left) *Ror2* Ctrl HFSCs could be passaged and maintained with feeders as evidenced by YFP expression, whereas YFP+ *Ror2* cKO HFSCs got lost along passaging. White dashed lines denote the outline of HFSC colonies. Scale bar represents 200 μm. (Right) Plot graph showing that *Ror2* cKO HFSCs could not be long-term maintained

in culture. Data are reported as average ± s.d; *n* = 3 groups, 6 colonies per group were examined. **c** Cultured *Ror2* cKO HFSCs display differentiation-like morphology and reduced proliferation ability. (Left) Differential interference contrast (DIC) images of long-term cultured *Ror2* Ctrl and cKO HFSCs. (Right) Proliferation abilities of *Ror2* Ctrl and cKO HFSCs were measured based on the daily counts of cell numbers. Data are reported as average ± s.d; *n* = 3 wells of individual samples. Scale bar represents 50 μm. **d** Cultured *Ror2* cKO HFSCs show reduced expression of HFSC markers, *Cd34* and *Krt15*. qPCR of mRNAs (left) and immunoblotting (right) from higher passaged *Ror2* Ctrl and cKO HFSCs. Data are reported as average ± s.d; *n* = 3 independent HFSC colonies; *Ror2*, **p < 0.0001; *Ccnd1*, **p = 0.0001; *Cd34*, **p = 0.0075; *Krt15*, **p = 0.0028. All *p* values were calculated using unpaired two-sided *t*-test. Source data are provided as a Source Data file.

signaling could be one, but not the only, cause of delayed HFSC activation.

Reduction in JNK, PKC, and small GTPases activities suggested that *Ror2⁻/⁻* HFSCs might have defects in Wnt5a-induced cell migration. To explore this possibility, we conducted transwell cell migration assay with *Ror2⁺/⁺* and *Ror2⁻/⁻* HFSCs using either Wnt5a or serum as a chemoattractant. As shown in Fig. 4e, Wnt5a significantly promoted cell migration of *Ror2⁺/⁺* HFSCs, but this migratory effect was completely abolished in *Ror2⁻/⁻* HFSCs. Even upon stimulation with serum, which contains additional migration stimuli, *Ror2⁻/⁻* HFSCs were still unable to migrate efficiently (Fig. 4e), suggesting that *Ror2⁻/⁻* HFSCs are defective in cell movement. Indeed, when examined by scratch wound migration assay, *Ror2⁻/⁻* HFSCs displayed impaired ability in collective cell migration (Supplementary Fig. 4d), indicating the irreplaceable role of ROR2 in regulation of HFSC motility.

Moreover, consistent with our in vivo findings (Fig. 2e, f), we also discovered that cultured *Ror2⁻/⁻* HFSCs were unable to maintain the expression of HFSC stemness genes. In fact, except for *Nfatc1*, which is associated with HFSC quiescence, *Ror2⁻/⁻* HFSCs expressed lower levels of HFSC stemness genes as well as HF fate-related genes (Fig. 4f). Taken together, our analyses with cultured HFSCs lacking *Ror2* not only confirm the previous discoveries that ROR2 mediates Wnt5-dependent signaling and cell migration via activation of JNK, PKC, and small

GTPases, but also demonstrate that endogenous ROR2 is required to maintain canonical Wnt-induced β-catenin transcriptional activation via regulating GSK3β stability.

## ROR2 is required for the activation of ATM/ATR-dependent DNA damage response in HFSCs

In addition to cell migration defects, by conducting EdU labeling experiments we also observed that fewer *Ror2⁻/⁻* HFSCs underwent S-phase DNA synthesis as evidenced by a reduced number of *Ror2⁻/⁻* HFSCs showing EdU incorporation (Fig. 5a). Interestingly, cell cycle analysis by flow cytometry showed a decrease in the portion of *Ror2⁻/⁻* HFSCs in G₀/G₁ phase and the accumulation of *Ror2⁻/⁻* HFSCs in both S and G₂/M phases (Fig. 5b). The results from EdU labeling and cell cycle analysis pointed out that *Ror2⁻/⁻* HFSCs showed slower S phase progression and G₂/M cell cycle arrest. Slowing of DNA replication is the hallmark of the S-phase DNA damage checkpoint, and G₂/M cell cycle arrest is also a consequence of DNA damage[69,70]; thus, our results suggested that *Ror2⁻/⁻* HFSCs were mounting a DNA damage response. DNA damage can be triggered by exogenous factors or endogenous sources. Given the equal culture condition of *Ror2⁺/⁺* and *Ror2⁻/⁻* HFSCs, the endogenous processes, such as oxidative stress, were likely to be the source causing DNA damage in *Ror2⁻/⁻* HFSCs. To address these possibilities, we performed immunostaining for γH2AX, a biomarker

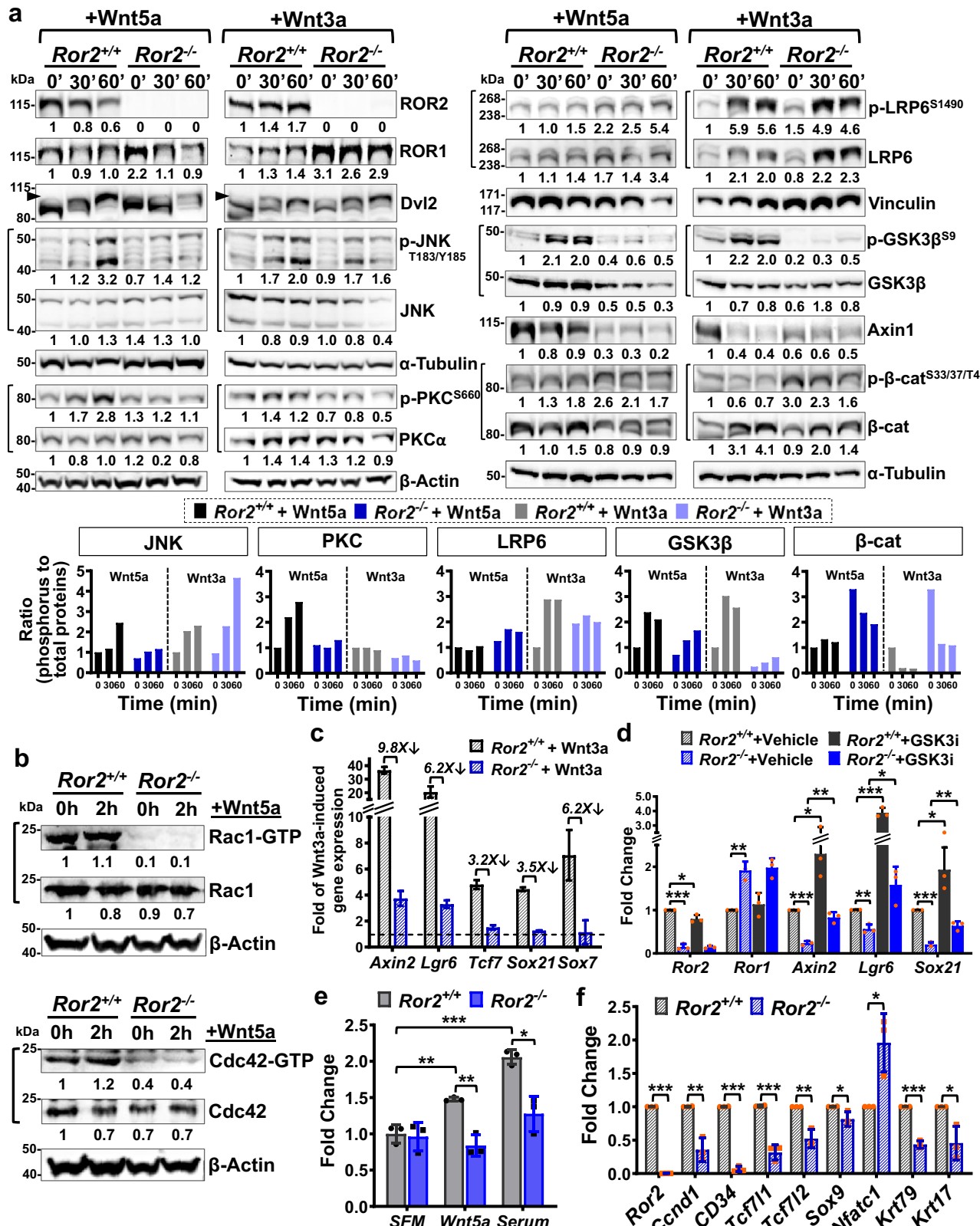

for DNA double-strand breaks (DSBs)[71], and 8-oxoguanine (8-oxoG), a biomarker for measuring the effect of oxidative DNA damage[72]. As shown in Fig. 5c and Supplementary Fig. 5a, three times more of *Ror2*[−/−] HFSCs showing over 6 γH2AX foci than *Ror2*[+/+] HFSCs indicated the accumulation of DNA DSBs in *Ror2*[−/−] HFSCs. We also found a significant percentage of *Ror2*[−/−] HFSCs displaying nuclear staining of 8-oxoG, something not detected in the nuclei of *Ror2*[+/+] HFSCs (Fig. 5d).

The nuclear staining of 8-oxoG in *Ror2*[−/−] HFSCs suggested an increase in the production of intracellular ROS. As judged by the staining with CellROX dye, the ROS level was indeed significantly higher in *Ror2*[−/−] HFSCs (Fig. 5e). Our results together implicate that *Ror2* depletion results in excess ROS production and accumulation of DNA damage, which in turn leads to slow proliferation and cell cycle arrest.

**Fig. 4 | ROR2 not only transduces non-canonical Wnt signaling and regulates HFSC migration, but also modulates canonical Wnt signaling via regulating GSK3β stability. a** *Ror2*⁺/⁺ and *Ror2*⁻/⁻ HFSCs were treated with Wnt5a or Wnt3a for 30 and 60 min before harvested for immunoblotting. Arrowhead denotes the phosphorylated form of Dvl2 protein. Values of fold change in the protein levels are shown under indicated bands. Data were normalized to β-Actin or α-Tubulin and calculated by comparing to *Ror2*⁺/⁺ HFSCs at 0′. Quantification of phosphorylated to total proteins in a time course manner is shown below. **b** *Ror2*⁺/⁺ and *Ror2*⁻/⁻ HFSCs were treated with Wnt5a for 2 h, and GTP-bound Rac1 or Cdc42 were pulled down and identified using a small GTPase activation assay. Quantification of protein levels is shown. Data of total proteins were normalized to β-Actin. **c** ROR2 is required for the transcriptional activation of canonical Wnt signaling. HFSCs were starved overnight and then treated with Wnt3a for 6 h. Folds of Wnt3a induction were calculated by comparing Wnt3a-treated HFSCs to their vehicle-treated

counterparts. Folds of reduction are shown above the bars. Data are reported as average ± s.d. **d** Inhibition of GSK3β activity restores the expression of canonical Wnt target genes in *Ror2*⁻/⁻ HFSCs. Real-time PCR analyses of *Ror2*⁺/⁺ and *Ror2*⁻/⁻ HFSCs treated with DMSO (vehicle) or CHIR-99021 (GSK3i) for 24 h. **e** *Ror2*⁻/⁻ HFSCs are not responsive to Wnt5a-induced cell migration. Migration abilities of *Ror2*⁺/⁺ and *Ror2*⁻/⁻ HFSCs were examined by transwell cell migration assay with Wnt5a or serum as a chemoattractant. **f** Real-time PCR analyses of *Ror2*⁺/⁺ and *Ror2*⁻/⁻ HFSCs for *Ror2*, *Ror1*, HFSC stemness, and HF fate-related genes. Note that except for *Nfatc1* which is related to quiescent state of HFSCs, *Ror2*⁻/⁻ HFSCs expressed lower levels of HFSC stemness and HF fate-related genes. Data in **d**–**f** are reported as average ± s.d.; *n* = 3 independent experiments; \**p* < 0.05, \*\**p* < 0.005, \*\*\**p* < 0.0005. All *p* values were calculated using unpaired two-sided *t*-test. All immunoblotting results are representative of at least two independent experiments. Source data are provided as a Source Data file.

ATM and ATR are central regulators of the DDR, and ATM also functions as a redox sensor to control the level of ROS. ATM deficiency increases the level of ROS and reduces NRF2-dependent antioxidant response[37,42,43]. Excess ROS production and accumulation of DNA damage in *Ror2*⁻/⁻ HFSCs led us to postulate that *Ror2* depletion might impair ATM- and/or ATR-dependent DDR. To test this possibility, we examined ATM activity, as well as the activity of ATR, which could act as a complementary regulator for ATM in the DDR[73]. Immunoblotting analyses showed dramatic reduction in the activity and total protein expression of ATM, and to a lesser extent for ATR, in *Ror2*⁻/⁻ HFSCs (Fig. 5f). The reduced ATM/ATR activity in *Ror2*⁻/⁻ HFSCs was further confirmed by the diminished levels in phosphorylation of ATM/ATR substrates that regulate cell cycle checkpoints and DNA repair[73] (Fig. 5g). More specifically, CHK2 and CHK1, the direct downstream targets of ATM and ATR, respectively, were downregulated in *Ror2*⁻/⁻ HFSCs (Supplementary Fig. 5b).

Due to the elevated ROS level, downregulation of ATM in *Ror2*⁻/⁻ HFSCs might also compromise ROS-activated ATM downstream effectors, AMPK and NRF2. Indeed, phosphorylation of AMPK and NRF2 were both decreased in *Ror2*⁻/⁻ HFSCs as compared to their control cells (Fig. 5h, i). The downregulation of NRF2 was further demonstrated by the remarkable reduction of activated NRF2 in the nucleus of *Ror2*⁻/⁻ HFSCs (Supplementary Fig. 5c). Interestingly, it was demonstrated that inhibition of NRF2 could lead to transcriptional repression of *ATM* and *ATR* genes; thus, dramatic reduction of NRF2 activity might explain why the total protein levels of ATM and ATR were prominently compromised in *Ror2*⁻/⁻ HFSCs.

GSK3β is also shown to inhibit the activity of AMPK[74] and regulate the degradation of NRF2[75]; thus, unleashed GSK3β activity in *Ror2*⁻/⁻ HFSCs could be one of causes leading to their downregulation. In fact, inhibition of GSK3β activity by treatment of GSK3 inhibitor could restore the expression level and activity of NRF2 in *Ror2*⁻/⁻ HFSCs to the comparable levels with those in *Ror2*⁺/⁺ HFSCs, but had no effect on AMPK activity (Fig. 5j). This data suggests that excess GSK3β activity resulted from ROR2 loss partly contributed to imbalanced oxidative response in *Ror2*⁻/⁻ HFSCs. In summary, our results reveal a previously unrecognized role of ROR2 in the regulation of ATM/ATR-dependent DDR in HFSCs. Without ROR2, HFSCs are unable to properly repair DNA damage and balance ROS production/scavenging attributed to deficiencies in activation of ATM, ATR, and their downstream signaling, which eventually leads to slow cell growth, accumulation of unrepaired DNA damage, and the elevation of ROS (Supplementary Fig. 5d).

**β-Catenin-null HFSCs upregulate ROR2 to protect themselves against the loss of stem cell pool and wrong fate differentiation**
Unlike *Ror2* cKO HFSCs, *β-catenin*-null HFSCs are maintained in their niche long-term, despite their inability to be activated during HF cycling[63]. However, the mechanism that prevents *β-catenin*-null HFSC depletion from the stem cell pool remains unclear. Using an inducible

conditional mutant mouse line for *β-catenin* (*K15CrePGR;Ctnnb1*^fl/fl^;*ROSA26*^LSL−YFP^), we depleted *β-catenin* in HFSCs and their progenies at P18−25, and found that *β-catenin* cKO (*β-cat* KO) HFs displayed a delay in anagen entry at the 1st hair cycle and then stayed at the 2nd telogen throughout the rest of the life as evidenced by the delay and inability of hair coat recovery, respectively (Fig. 6a). Notably, arrested *β-cat* cKO HFs maintained intact bulge compartments (Fig. 6a, bottom right), which was also found in the previous study where *β-catenin* was depleted in HFSCs at the 2nd telogen[63]. Interestingly, while analyzing the expression of HFSC stemness genes in *β-cat* Ctrl and cKO HFSCs at the 2nd telogen phase, we found an association of lower expression levels of HFSC stemness genes with the upregulation of *Ror2* expression. At the early telogen, *β-cat* cKO HFSCs displayed similar or slightly elevated levels of HFSC stemness genes; in these HFSCs, the expression level of *Ror2* mRNA was comparable with those in their control HFSCs (Fig. 6b, top). However, in the late telogen *β-cat* cKO HFSCs where the expression of HFSC stemness genes went down, the level of *Ror2* mRNA became elevated (Fig. 6b, middle). The elevation of *Ror2* expression was retained in the *β-cat* cKO HFSCs of aged animals (Fig. 6b, bottom). In agreement, the elevated *Ror2* expression in *β-cat* cKO HFSCs at late telogen and aged HFs was confirmed by whole-mount immunostaining, showing that ROR2 expression was higher in the bulge of *β-cat* cKO HFs than Ctrl HFs at late 2nd telogen (Supplementary Fig. 6a) as well as at aged animals (Supplementary Fig. 6b).

To investigate if upregulated ROR2 in *β-cat* cKO HFSCs plays a role in HFSC maintenance, we generated an inducible conditional double mutant mouse line for *Ror2* and *β-catenin* (*K15CrePGR;Ror2*^fl/fl^;*Ctnnb1*^fl/fl^;*ROSA26*^LSL−YFP^), referred as *Ror2/β-cat* dKO. Depletion of *Ror2* and/or *β-catenin* was initiated by administration of RU486 to mice at P18-25. Similar to *Ror2* cKO and *β-cat* cKO mice, *Ror2/β-cat* dKO mice also showed the delay of anagen entry at the 1st hair cycle (Fig. 6c, top). However, at the onset of the 2nd anagen while *Ror2* cKO HFs continuously showed the delay of anagen entry (Fig. 1e) and *β-cat* cKO HFs were arrested at telogen (Fig. 6a and Supplementary Fig. 7a), *Ror2/β-cat* dKO HFs failed to initiate hair cycle and displayed signs of massive sebocyte differentiation (Fig. 6c and Supplementary Fig. 7b). Quantitative analysis of HFs from Ctrl, *Ror2* cKO, *β-cat* cKO, and *Ror2/β-cat* dKO mice demonstrated that the phenotype displaying enlarged sebaceous glands (SGs) was only observed in *Ror2/β-cat* dKO HFs and that more than a half of *Ror2/β-cat* dKO HFs showed enlarged SGs accompanied with aberrant HF structure or diminished bulge compartments (Fig. 6c, bottom). Immunostaining of whole-mount HFs revealed the gradual loss of CD34⁺ *Ror2/β-cat* dKO HFSCs along with the enlargement of SG (Supplementary Fig. 7c). Lineage tracing with YFP confirmed that these differentiated sebocytes residing in enlarged SGs were generated from Cre-activated YFP+ *Ror2/β-cat* dKO HFSCs (Fig. 6d). Depletion of *Ror2* and *β-catenin* were confirmed in FACS-purified YFP+ *Ror2/β-cat* dKO HFSCs that showed prominent downregulation of HFSC stemness genes (Fig. 6e); this

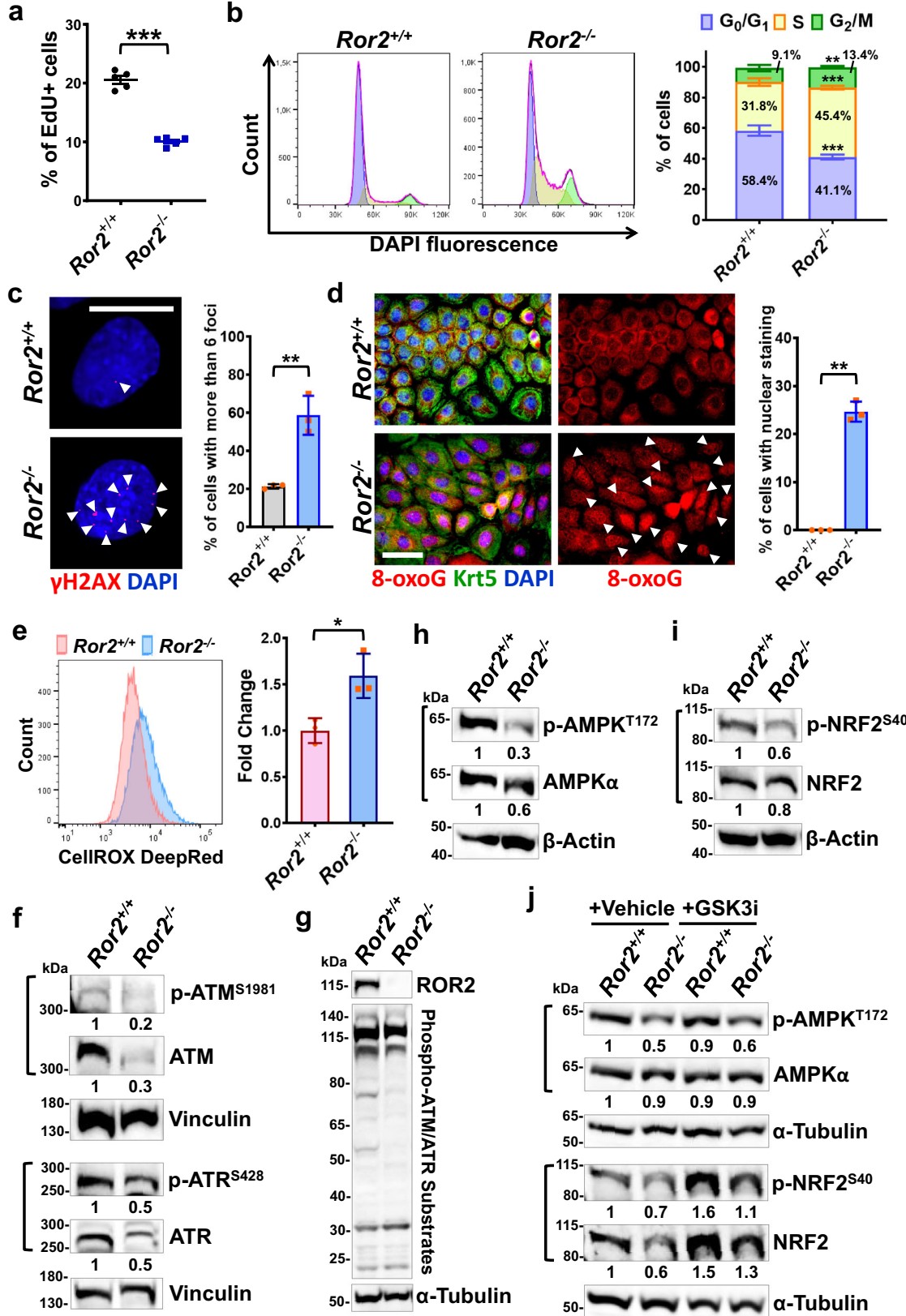

downregulation was even more significant than those in *Ror2* cKO (Fig. 2e, f) or in the late telogen *β-cat* cKO HFSCs (Fig. 6b). In summary, our results unravel a previously unrecognized role of ROR2 in long-term maintenance of HFSCs, which is especially important for those HFSCs that would undergo wrong fate differentiation upon activation.

## PKC inhibition recapitulates *Ror2* depletion in regulation of HFSC proliferation and maintenance

We have showed that ROR2 is required for Wnt-induced activation of JNK and PKC (Fig. 4a). Without additional Wnt stimulation, the overall activities of JNK and PKC were already lower in *Ror2*[−/−] HFSCs that showed significant reduction in p-GSK3β[S9] in the regular growing

**Fig. 5 | *Ror2* depletion impairs ATM/ATR-dependent DNA damage response, which in turn leads to DNA damage accumulation, ROS elevation, and cell cycle arrest. a** *Ror2*$^{-/-}$ HFSCs display reduced cell proliferation. HFSCs were labeled with EdU for 4 h prior to examination. Data are reported as mean ± SEM; *n* = 5 independent experiments; ***p* = 0.00001. **b** *Ror2* depletion causes slower S phase progression and cell cycle arrest at G$_2$/M phase. (Left) DNA content of HFSCs was stained with DAPI and analyzed by flow cytometry. (Right) Bar chart depicts the cell cycle distribution. **c** *Ror2*$^{-/-}$ HFSCs show a significant increase in double-strand breaks. Immunofluorescence staining of *Ror2*$^{+/+}$ and *Ror2*$^{-/-}$ HFSCs for γH2AX (red). Arrowheads indicate γH2AX foci. Quantification of cells showing more than 6 foci is shown at right. **d** Immunofluorescence staining of *Ror2*$^{+/+}$ and *Ror2*$^{-/-}$ HFSCs for Keratin 5 (Krt5; green) and 8-oxoguanine (8-oxoG; red). Quantification of cells showing nuclear 8-oxoG is shown at right. **e** Flow cytometry analysis of the ROS level in *Ror2*$^{+/+}$ and *Ror2*$^{-/-}$ HFSCs using CellROX Deep Red reagent. Values of fold change in the ROS level are shown at right. **f–i** Immunoblotting analyses of *Ror2*$^{+/+}$ and *Ror2*$^{-/-}$ HFSCs for activated and total proteins of ATM and ATR (**f**), AMPK (**h**), NRF2 (**i**), as well as for phosphorylated ATM/ATR substrates and ROR2 (**g**). Vinculin, α-Tubulin, and β-Actin were used as internal controls for indicated experiments. Values of fold change in the protein levels are shown under indicated bands. **j** Inhibition of GSK3β activity restores the expression level and activity of NRF2, but not AMPK, in *Ror2*$^{-/-}$ HFSCs. Immunoblotting analyses with *Ror2*$^{+/+}$ and *Ror2*$^{-/-}$ HFSCs treated with DMSO (vehicle) or CHIR-99021 (GSK3i) for 24 h. Scale bars in **c** and **d** represent 15 and 50 μm, respectively. Data in **b–e** are reported as average ± s.d.; *n* = 3 independent experiments; ***p* < 0.005, ****p* < 0.0005 (**b**), ***p* = 0.0035 (**c**), ***p* = 0.0024 (**d**), **p* = 0.02 (**e**). All *p* values were calculated using unpaired two-sided *t*-test. All immunoblotting results are representative of at least two independent experiments. Source data are provided as a Source Data file.

medium (Supplementary Fig. 8a). Other than cultured HFSCs, we also found that both total and activated forms of JNK and PKC proteins were significantly reduced in depilation-activated *Ror2* cKO HFSCs (Fig. 7a). Immunostaining of whole-mount *Ror2* Ctrl and cKO skin at anagen onset confirmed the reduction of JNK and PKC protein expression and their activities in depilation-activated *Ror2* cKO HFSCs (Supplementary Fig. 8b). Notably, the reduction of PKC expression in *Ror2* cKO HFSCs that showed diminished ROR2 expression persisted in telogen phase (Fig. 7b). To address whether the effects of *Ror2* depletion in HFSC proliferation and maintenance were caused by the downregulation of JNK and/or PKC in *Ror2*$^{-/-}$ HFSCs, we used small molecule inhibitors, SP600125 (JNK inhibitor; JNKi) and GF109203X (PKC inhibitor selective to classical and novel PKC isoforms; PKCi), to suppress the activities of JNK and PKC of cultured HFSCs, respectively, and then examined their effects in HFSC proliferation by performing EdU labeling experiments. As shown, inhibition of PKC, but not JNK, erased the differences in cell proliferation between *Ror2*$^{+/+}$ and *Ror2*$^{-/-}$ HFSCs (Supplementary Fig. 8c), suggesting that ROR2 regulates HFSC proliferation, at least partly, via PKC-dependent signaling.

PKC is shown to phosphorylate and inactivate GSK3β[76,77], we wonder whether PKC inhibition could partly recapitulate *Ror2* loss by alleviating GSK3β inactivation. As shown, PKC inhibition, validated by reduced PKC activity and total protein level upon PKCi treatment, compromised GSK3β inactivation in *Ror2*$^{+/+}$ HFSCs to the similarly low level as those in *Ror2*$^{-/-}$ HFSCs (Fig. 7c), suggesting that reduction of PKC activity upon *Ror2* depletion could be one of causes leading to elevated GSK3β activity. Interestingly, while increasing GSK3β stability, PKC inhibition had only minor effects on β-catenin stability and β-catenin-dependent transcriptional activity judged respectively by the phosphorylation/expression levels of β-catenin and canonical Wnt target gene expression (Fig. 7c and Supplementary Fig. 9a). These data imply that PKC mainly modulates cytoplasmic pool of GSK3β in HFSCs. Moreover, in agreement with the fact that GSK3β negatively modulates NRF2 activity in HFSCs (Fig. 5j), PKC inhibition, which stabilized GSK3β, resulted in a decrease in NRF2 activity while showing no significant effects on AMPK activation (Supplementary Fig. 9b). Taken together, our results implicate that in addition to compromising Wnt-dependent signaling activation, loss of ROR2 also resulted in PKC downregulation, which could contribute to GSK3β stabilization and NRF2 inactivation, thereby leading to imbalanced oxidative response.

We have showed that ROR2 expression was elevated in *β-cat* cKO HFSCs at late telogen and aged HFs (Supplementary Fig. 6); interestingly, we also detected increased PKC expression in these *β-cat* cKO HFSCs of late telogen HFs (Supplementary Fig. 10a) as well as at aged animals (Supplementary Fig. 10b). To determine whether PKC plays a role downstream of ROR2 in *β-cat* cKO HFSCs, we inhibited PKC activity by applying PKCi to the back skin of *β-cat* Ctrl and cKO mice during the 2nd telogen (Fig. 7d, top). Notably, 1 week after application of PKCi, *β-cat* cKO HFSCs had already undergone sebocyte differentiation, and by 3 weeks post-treatment 77% of *β-cat* cKO HFs lost

their bulge compartments accompanied with enlarged SGs (Fig. 7d and Supplementary Fig. 10c). The phenotype of *β-cat* cKO HFs from PKC inhibition recapitulated the effect of *Ror2* depletion in *β-cat* cKO HFSCs, suggesting that PKC acts in the same axis as ROR2 in regulating HFSC maintenance. Furthermore, we found that the effect of PKC inhibition in downregulation of HFSC stemness genes was greater in *β-cat* cKO HFSCs than *β-cat* Ctrl HFSCs (Fig. 7e), suggesting PKC in *β-cat* cKO HFSCs plays a role downstream of ROR2 in HFSC maintenance. Together, our collective findings provide compelling evidence that ROR2-dependent regulation is vital for the proliferation and maintenance of HFSCs, especially when HFSCs are unable to differentiate properly.

## Discussion

Extensive research on how Wnt signaling modulates the behavior of stem cells has focused on β-catenin-dependent canonical Wnt pathway, yet very limited information about β-catenin-independent signaling in the stem cell niche can be found. This conceptual gap has prompted us to investigate the functional significance of a Wnt receptor ROR2 in regulation of stem cells by using the HF as a model system. Using a genetic mouse model, we showed that ROR2 is indispensable for HFSC self-renewal and long-term maintenance. Further analyses with cultured HFSCs demonstrated that ROR2 not only transduces Wnt5a-activated non-canonical Wnt signaling and migration through activating PKC, JNK, and small GTPases, but also is required for Wnt3a-induced expression of canonical Wnt target genes in HFSCs via regulating GSK3β stability. Strikingly, by searching for ROR2-dependent regulation responsible for HFSC maintenance, we uncovered a previously unrecognized function of ROR2 in regulation of ATM/ATR-dependent DDR, which is important for the maintenance of HFSC genomic integrity. Lastly, our analyses with *β-catenin*-null HFSCs further unveiled a compensatory role of ROR2-PKC signaling in protecting HFSCs, especially for those unable to differentiate properly, from the uncontrollable loss. Collectively, our results provide an insight into a signaling network cross-regulated by a Wnt receptor in the context of stem cells (Fig. 7f).

Several studies have supported the notion that ROR2 mediates Wnt5a-dependent antagonism of Wnt/β-catenin signaling[22,30–33]. However, our data indicate that ROR2 is not only responsible for the activation of Wnt5a-dependent signaling, but is also essential for canonical Wnt-activated transcriptional regulation in HFSCs. We showed that canonical Wnt target genes were downregulated in *Ror2*-depleted HFSCs upon depilation-induced HFSC activation and at anagen onset, these processes predominantly governed by Wnt/β-catenin signaling to promote hair cycle entry[61,63]. We also found that *Ror2* depletion diminished Wnt3a-induced transcriptional activation in cultured HFSCs due to unleashed GSK3β activity. Both in vivo and in vitro analyses indicate that ROR2 favors to activate rather than antagonize Wnt/β-catenin signaling in HFSCs via regulating GSK3β stability. Our findings are in agreement with those studies showing that

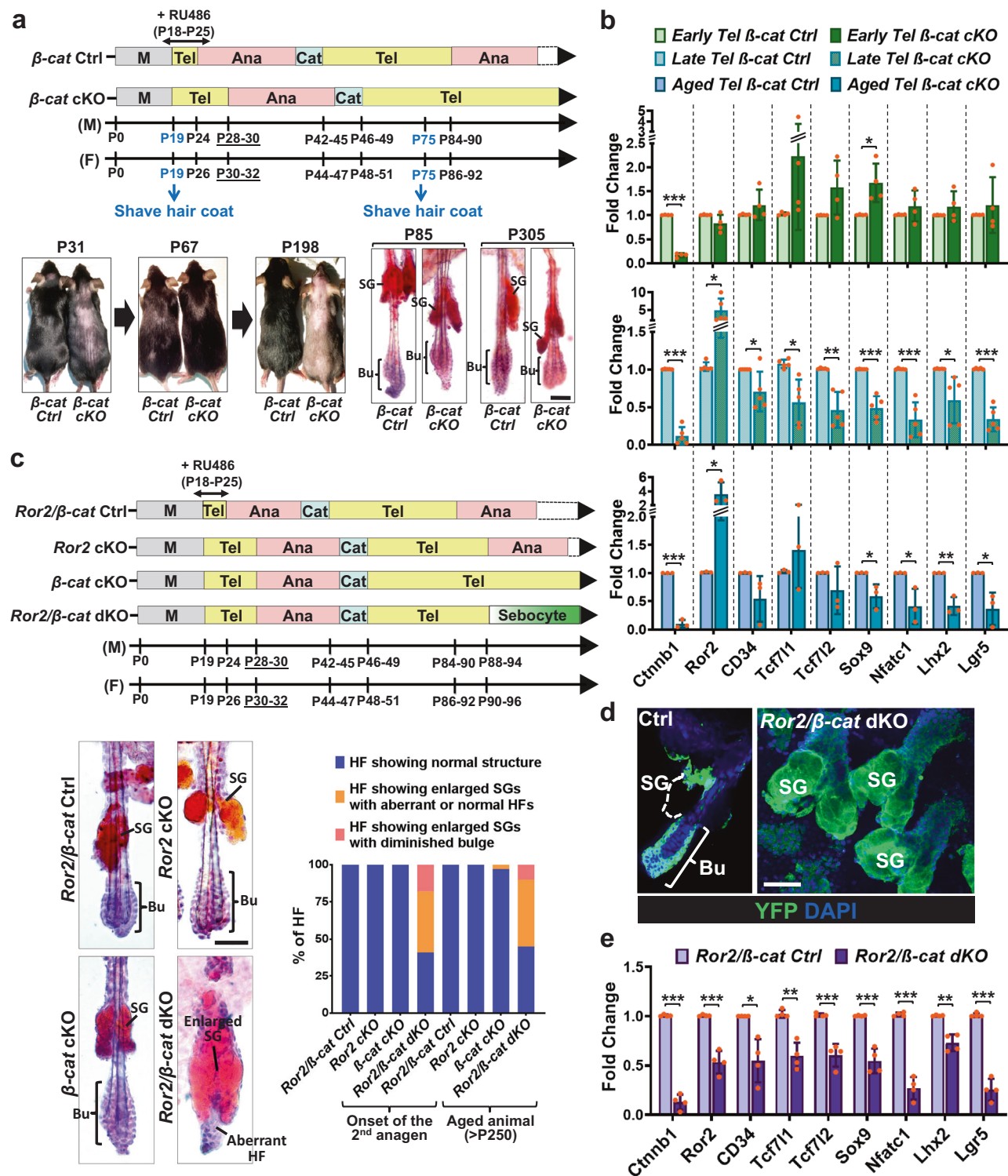

overexpressed ROR2 can potentiate Wnt3a-induced β-catenin signaling activity measured by the Wnt/β-catenin signaling reporter[31,35,36]. Yet, our research model offers a physiological context allowing us to determine the action of endogenous ROR2 in the regulation of Wnt/β-catenin signaling.

It has been shown that HFSCs reside in the niche rich for non-canonical Wnt ligands, e.g., Wnt4 and Wnt11, throughout hair cycle[56,66,78], implicating a possibility of non-canonical Wnt signaling being activated in HFSCs for the maintenance. In our study, we demonstrate that ROR2 is essential to activate non-canonical Wnt-induced signaling in HFSCs, and that a population of HFSCs could not

be maintained without ROR2. Moreover, in *β-cat* cKO HFSCs where canonical Wnt signaling is suppressed, ROR2 is upregulated to protect *β-catenin*-null HFSCs from the loss. All these findings support a model where ROR2-dependent regulation, at least partly activated by non-canonical Wnts, plays a critical role in maintaining HFSCs at the quiescent and/or activated state. By examining *Ror2*-depleted HFSCs in culture, we identified ROR2-dependent regulation in ATM/ATR-mediated DDR, which is crucial for the long-term maintenance of HFSCs. However, it remains unclear if this mechanism is utilized by HFSCs residing in their niche and whether the regulation of DDR by ROR2 depends on Wnt. Thus, future investigation should focus on analyzing

**Fig. 6 | β-catenin-null HFSCs elevate ROR2 expression to protect themselves from the loss of stem cell pool and wrong fate differentiation. a** *β-cat* cKO HFs stay at telogen phase permanently without losing HFSC compartment. (Top) Schematic diagram illustrating hair cycle progression of *β-cat* Ctrl and cKO mice. (Bottom left) *β-Cat* cKO mouse displayed a delay in hair coat recovery at the 1st anagen (P31), and then stay at the 2nd telogen permanently without entering the next hair cycle as evidenced by no hair coat recovery from the shaved skin of the older animal (P198). (Bottom right) Oil Red O staining of sebocytes on whole-mount skins from *β-cat* Ctrl and cKO mice. The bulge compartment of *β-cat* cKO HFs at aged animals is intact and HFSCs are maintained. **b** Real-time PCR analyses of FACS-purified *β-cat* Ctrl and cKO HFSCs from early (P55-60), late (P65-90), and aged (>P250) telogen HFs. Values of *β-cat* cKO HFSCs were normalized to *β-cat* Ctrl HFSC mRNA for each set. **c** *Ror2* depletion in *β-cat* cKO HFs results in the loss of HFSCs and wrong fate differentiation. (Top) Schematic diagram illustrating the

hair cycle progression of *Ror2/β-cat* Ctrl, *Ror2* cKO, *β-cat* cKO, and *Ror2/β-cat* dKO mice. (Bottom left) Oil Red O staining of sebocytes on whole-mount skins from *Ror2/β-cat* Ctrl, *Ror2* cKO, *β-cat* cKO, and *Ror2/β-cat* dKO mice. (Bottom right) Quantification of HFs with normal structure, enlarged sebaceous glands (SGs), or accompanied by diminished bulge is shown. **d** Immunofluorescence images of whole-mount skins from *Ror2/β-cat* Ctrl and dKO mice for YFP. Results are representative of at least three independent experiments. **e** Real-time PCR analyses of *Ror2/β-cat* Ctrl and dKO HFSCs for HFSC stemness genes. Values were normalized to mRNA of *Ror2/β-cat* Ctrl HFSCs. Scale bars in **a**, **c**, **d** represent 50 μm. Data in **b** and **e** are reported as average ± s.d.; *n* = 4 (for Early Tel), 5 (for Late Tel), or 3 (for Aged Tel) biological independent animals (**b**), *n* = 4 biological independent animals (**e**); *$p < 0.05$, **$p < 0.005$, ***$p < 0.0005$. Unpaired two-sided *t*-test. Source data are provided as a Source Data file.

the Wnt dependency in ROR2-mediated regulation and deciphering the mechanism of ROR2 underlying the above-mentioned regulation, e.g., by identifying ROR2-interacting proteins.

Our analyses with the in vivo mouse model show that although a population of *Ror2*-depleted HFSCs got lost after the 1st round of HF regeneration, the recovered *Ror2* cKO HFSCs after the process were able to sustain and replenish HF lineage in the following regeneration. However, FACS-purified *Ror2* cKO HFSCs could not be maintained in culture long-term in the medium promoting proliferation. This discrepancy in HFSC survival in vivo and in culture is likely due to the differences in the frequency of HFSC proliferation. HFSCs residing in their native niche are maintained in a quiescent state until the initiation of hair cycle. Even at anagen onset, only a subset of HFSCs undergo proliferation for self-renewal and regeneration. On the contrary, HFSCs in culture are constantly in proliferation, which makes HFSCs vulnerable to DNA damage. Given the deficiency of *Ror2*-depleted HFSCs in ATM/ATR-dependent DDR, high frequency of DNA replication could lead to rapid accumulation of DNA damage, which subsequently causes cell cycle arrest or death. This explains why *Ror2* cKO HFSCs in culture displayed stronger phenotypes than those residing in their niche.

Our identification on dual roles of ROR2 involved in canonical and non-canonical Wnt signaling crystallized the idea that cells exploit different Wnt signaling components to build a Wnt signaling network in order to maintain the stability of cellular state. The ability of ROR2 to transduce condition-specific signals likely involves the availabilities of co-receptors, interacting proteins, or downstream effectors. Hence, ROR2-dependent signaling may trigger cell type-specific effects and lead to a situation-dependent functional outcome. This last conjecture may provide the required flexibility to the system to accomplish multiple functions. Future studies should delve into the interactions between ROR2-dependent canonical and non-canonical Wnt signaling that contribute to the self-renewal and maintenance of HFSCs.

## Methods

### Mice and treatment
Mice were housed and treated according to the guidelines of the University Animal Ethics Committee, Université Catholique de Louvain. All experimental procedures were conducted in compliance with animal welfare regulations of Belgium. All laboratory mice are housed in 12:12 light:dark light cycles at ambient temperature 20–24 °C and humidity ranges of 45–65%. *K15CrePGR* and *Ror2^fl/fl* mice on a C57BL/6J background were obtained from Jackson laboratory. Inducible knockout mouse lines were generated by crossing *K15CrePGR*[79], *Ror2^fl/fl* and/or *Ctnnb1^fl/fl*[80], *ROSA26^LSL-YFP*[81]. The strategy to generate *Ror2* cKO and their control littermates was by breeding *K15CrePGR+;Ror2^+/fl;Rosa26^LSL-YFP* males with *Ror2^fl/fl; Rosa26^LSL-YFP* females. The control animals for *Ror2* cKO mice were sex-matched *Ror2* heterozygous (*Cre+*) littermates, or wildtype (*Cre-*) littermates only when the littermates of *Ror2* cKO mice did not contain any *Ror2* heterozygous mouse. Cre-

recombinase activity was induced by intraperitoneal injection of RU486 (1 mg/mouse; TCI Europe N.V.) at P18, P21, and P24, and daily topical administration of 4% RU486 in ethanol from P21–P25. To induce synchronized HFSC activation, HF depilation was performed on anesthetized mice at P55. For PKC inhibitor treatment, 100 μg of GF109203X (R&D Systems) in acetone (1 μg/μl) or acetone alone was applied to P55 mouse back skin every other day for 3 days. Experiments were designed based on sex-, age- and strain-matched pairs, mainly littermates, and were repeated on ≥3 pairs of sample sets. Phenotype and obtained results were reproducible in both male and female pairs.

### Fluorescence-activated cell sorting
FACS purification procedures were as described previously[63,78]. Briefly, dorsal skin was harvested, scrapped for fat removal, and incubated in 0.25% trypsin for 30 min (females) or 40 min (males) at 37 °C. Epidermal and HF cells were scrapped off and cell suspension was filtered through 70 and 40 μm cell strainers (Fischer Scientific). Collected cells were suspended in PBS containing 4% FBS and incubated with antibodies before FACS. Stained cells were then sorted using a FACS Aria III (BD Biosciences) with FACS Diva v8.0.1 software. Fixable Viability Dye eFluor 780 was used to exclude death cells. Living HFSCs were then sorted based on surface expression of integrin α6, CD34 and cytosolic YFP. FACS gating scheme is provided in Supplementary Fig. 11. Cells were collected into either TRI reagent (Sigma-Aldrich) for RNA purification, pre-coated 15-ml Flacon tubes for protein purification, or culture medium with feeder cells for colony formation.

### Histology and immunofluorescence
Skin biopsies were collected and fixed in 4% paraformaldehyde at 4 °C for 4 h, and then immersed in 30% sucrose at 4 °C overnight before embedded in OCT compound. For skin whole-mounts, fat from the back skin was removed, and then incubated in 2 mg/ml dispase (Thermo Fisher Scientific) supplemented with 20 mM EDTA at 4 °C overnight. Skin epidermis with attached hair follicles was then peeled off from dermis and fixed in 4% PFA for 30 min at room temperature (RT). Washed whole mount skins were stored in PBS at 4 °C for further use.

For immunofluorescence, 7 μm tissue sections were fixed, washed, and permeabilized in 0.5% Triton-X100 for 20 min. Sections were then incubated in immunofluorescence blocking buffer (0.3% Triton X-100, 1% bovine serum albumin, 1% gelatin, 5% normal goat serum, 5% normal donkey serum in PBS) at RT for 1 h, and then with primary antibodies at 4 °C overnight. When applicable, the MOM Fluorescein Kit (Vector Laboratories) was used to prevent nonspecific binding of primary mouse monoclonal antibodies. Tissue sections were washed and incubated with Alexa Fluor-conjugated secondary antibody (Jackson ImmunoResearch) at RT for 1 h before mounted with mounting medium containing DAPI (Roth). For whole-mount tissues, immunostaining was performed as described previously with overnight incubation of primary antibodies at 4 °C[63]. Images of

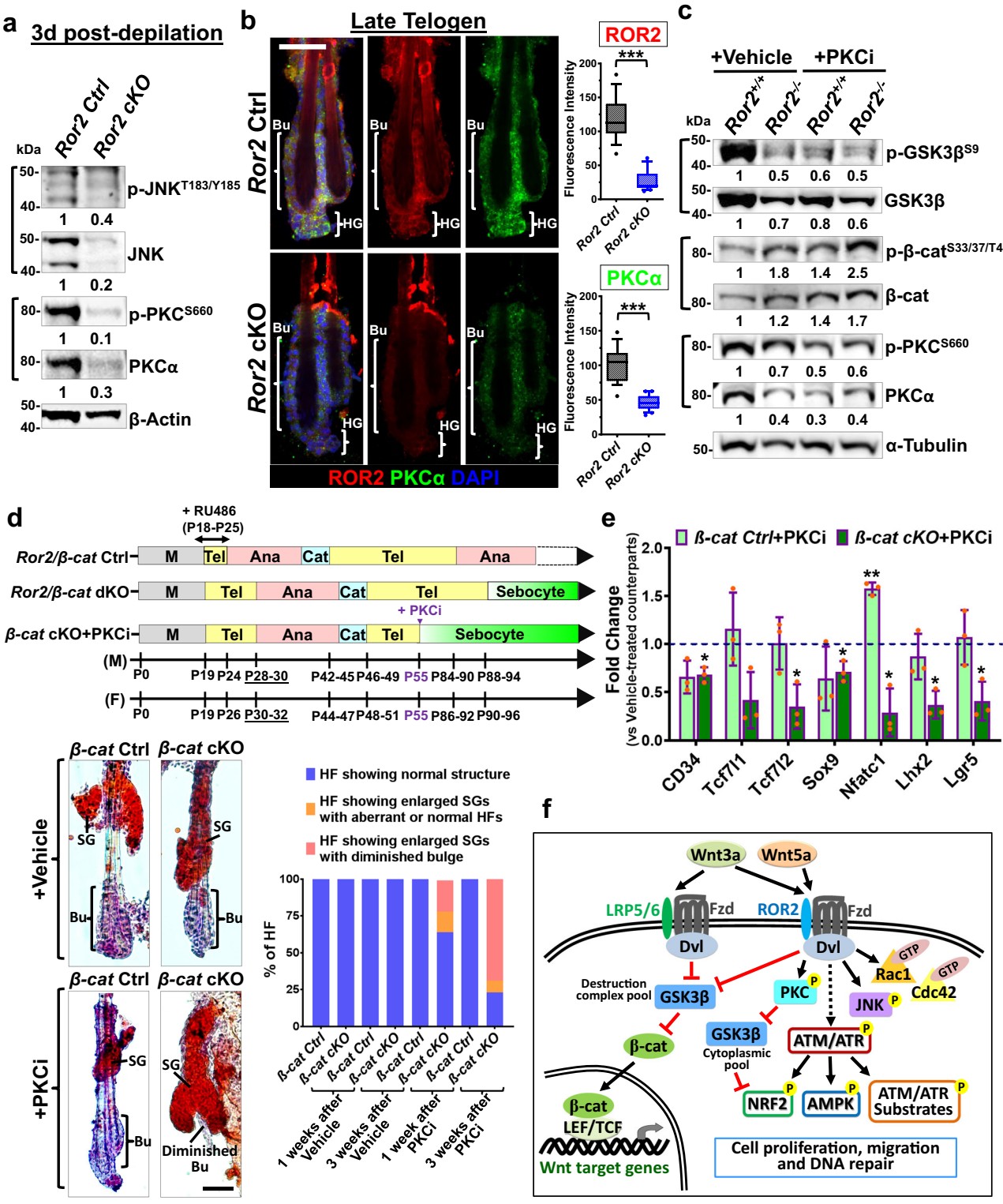

immunofluorescence staining were acquired using Zen software v3.0 (Zeiss) on a Axio Vert.A1 fluorescence microscope (Zeiss) or Zen software v2.6 (Zeiss) on a Axio observer Z1 confocal microscope (Zeiss). Fluorescence intensity was measured using ImageJ.

**EdU labeling assays**
To detect HFSC proliferation in vivo, 1 mg of EdU (5-ethynyl-2'-deoxyuridine; TCI Europe) per mouse was injected intraperitoneally twice at 12 h intervals for 24 h before skin samples were collected. For EdU-labeling assay in vitro, HFSCs were grown on Nunc Lab-Tek II Chamber

Slides (Thermo Fisher Scientific) coated with 10 µg/ml fibronectin (Merck Millipore). Cells were incubated with DMSO (vehicle), 2 µM of SP600125 (JNKi), or 0.5 µM of GF109203X (PKCi) for 24 h and then labeled with 10 µM EdU for 4 h prior to detection. Incorporated EdU was detected by a Click-iT EdU Alexa Fluor 647 Imaging Kit (Thermo Fisher Scientific) based on manufacturer's instructions.

**Colony formation assays, cell culture, and treatment**
For colony formation assays, FACS-purified HFSCs were seeded at 20,000 cells/well (for anagen onset HFs) or 33,000 cells/well (for

**Fig. 7 | PKC inhibition recapitulates *Ror2* depletion in regulation of HFSC maintenance. a** Immunoblotting analyses of FACS-purified HFSCs from 3dpd *Ror2* Ctrl and cKO HFs. **b** Whole-mount immunofluorescence staining of late telogen *Ror2* Ctrl and cKO HFs for ROR2 (red) and PKCα (green). Quantification analyses of fluorescence intensity for ROR2 and PKCα staining are shown at right. Data are reported as the median (the line within the box), the 25th to 75th percentiles (bottom and top lines of the box) and the 10th to 90th percentiles (bottom and top whiskers); $n = 18$ (Ctrl) or 20 (cKO) regions over 7 (Ctrl) or 8 (cKO) independent HFs; ***$p < 0.0001$. Unpaired two-sided *t*-test. **c** Immunoblotting analyses for indicated proteins with *Ror2*[+/+] and *Ror2*[-/-] HFSCs that were treated with DMSO (vehicle) or GF109203X (PKCi) for 24 h. **d** PKC inhibition in *β-cat* cKO HFs leads to the loss of HFSCs and wrong fate differentiation. (Top) Schematic diagram illustrating the hair cycle progression of *Ror2/β-cat* Ctrl, dKO mice, and *β-cat* cKO mice treated with PKC inhibitor at the 2nd telogen phase. (Bottom) Oil Red O staining of sebocytes on 1-week post-treated whole-mount skins (left) and quantification of HFs (right) are shown. **e** Real-time PCR analyses of FACS-purified HFSCs from 24 h acetone (vehicle)- or GF109203X (PKCi)-treated *β-cat* Ctrl or cKO mice. Values of GF109203X-treated *β-cat* Ctrl and cKO HFSCs were normalized to mRNA of vehicle-treated *β-cat* Ctrl and cKO HFSCs, respectively. Data are reported as average ± s.d.; $n = 3$ biological independent animals; *$p < 0.05$, **$p < 0.005$. Unpaired two-sided *t*-test. **f** An illustrating model summarizing our results in this study. In HFSCs, ROR2 not only transduces Wnt5a-induced non-canonical Wnt signaling via activation of PKC, JNK, and small GTPases, but also modulates Wnt3a-induced canonical Wnt signaling by controlling GSK3β stability. In parallel, ROR2 also plays a role in regulation of ATM/ATR-dependent DDR. These ROR2-dependent regulations modulate cell proliferation, motility, and DNA repair, which are essential for the long-term maintenance of HFSCs. All immunoblotting results are representative of at least two independent experiments. Scale bars in **b**, **d** represent 50 µm. Source data are provided as a Source Data file.

telogen-phase HFs) of six-well plate containing mitomycin C-treated J2 3T3 mouse fibroblast feeders (kind gift from E. Fuchs lab upon agreement of H. Green lab). After two weeks of co-culture, colonies were then fixed with 4% of paraformaldehyde and stained with 1% Rhodamine B. Numbers of colonies which sizes ≥ 3 mm$^2$ were counted for analysis. For long-term passaging experiments, three independent groups of 6 colonies per group were picked with clone cylinders (Bel-Art) and passaged onto mitomycin C-treated fibroblast feeders in 24-well plate. HFSCs were trypsinized and passaged at 1:10 dilution onto feeders every 10 days for 10 passages. The expression of *Ror2* from every single colony was validated by RT-qPCR.

FACS-purified HFSCs from the back skin of *Ror2*[fl/fl]/*ROSA26*[LSL-YFP] mice were cultured and expanded on mitomycin C-treated fibroblast feeders in medium containing 0.3 mM calcium as described[82]. After 10 passages, HFSCs were taken off from feeders and ready for lentiviral infection. For the treatment of Wnts, HFSCs were starved with medium containing 0.5% of FBS for 16 h and then treated with vehicle (PBS), carrier-free 100 ng/ml Wnt3a (R&D Systems) or 300 ng/ml Wnt5a (R&D Systems) for 30 or 60 min prior to harvesting for protein purification, or for 6 h for mRNA purification. For the treatment of GSK3 or PKC inhibitor, HFSCs were treated with vehicle (DMSO) or 1 µM of CHIR99021 (R&D Systems) or 1 µM of GF109203X (R&D Systems) for 24 h prior to harvesting for protein extraction.

### Migration assays, cell cycle analysis, and ROS detection

For directional cell migration assay, QCM 24-well colorimetric cell migration assay (Merck) was performed according to the manufacturer protocol. To analyze collective cell migration ability, scratch wound migration assay was performed. Briefly, feeder-free HFSCs were cultured to grow into a confluent monolayer, and then HFSC monolayer in the medium containing 1% of FBS with or without 8 µg/ml of mitomycin C (Merck) was scraped in a straight line using a pipet tip to create a gap. Images of the gap closing were acquired using a phase-contrast microscope at 0, 8, and 24 h post-scratching, and the gaps at each time points were measured using ImageJ. Presented data are the averages of three independent experiments.

For cell cycle analysis, DNA of HFSCs was stained with 10 µg/ml DAPI at 37 °C for 1 h, and DNA content was measured by LSR Fortessa flow cytometer with FACS Diva v8.0.1 software (BD Biosciences) and analyzed by using FlowJo v10.7.2 software. To measure the level of intracellular ROS, HFSCs were stained with CellROX Deep Red reagent (Thermo Fisher Scientific) and analyzed by LSR Fortessa flow cytometer according to the manufacturer's instruction.

### Lentiviral infection

HEK293FT cells (Thermo Fisher Scientific) were CaCl$_2$-transfected with a plasmid mix containing pLKO.1 lentiviral expressing plasmid, PAX2 and MD2 viral packing plasmids. 48 h post-transfection, virus-containing supernatant from transfected HEK293FT cells was collected, filtered and mixed with polybrene (Sigma-Aldrich) and an additional 10% FBS before applying to *Ror2*[fl/fl]/*ROSA26*[LSL-YFP] HFSCs. Viral infection was performed by centrifugation of HFSCs with virus-containing mix at $1100 \times g$ at 35 °C for 1 h. 5 days after infection, HFSCs were purified by FACS based on RFP and/or YFP expression. The following lentiviral expressing plasmids were used: pLKO.1-NLS-iCre-mRFP (kind gift from S. Beronja[83], pLKO.1-MCS-mRFP (derived from pLKO.1-NLS-iCre-mRFP where NLS-iCre was replaced by multiple cloning sites).

### Reverse transcription and quantitative Real-time PCR (RT-qPCR)

Total RNAs from FACS-sorted cells or cultured cells were purified with the Direct-zol RNA MiniPrep kit (Zymo Research) and reverse transcribed with ProtoScript(r) II First Strand cDNA Synthesis Kit (New England Biolabs). cDNAs were amplified with indicated primers in SYBR Green (Bio-Rad) quantitative PCR assay on the Thermoblock 96 Silver Block Real-Time PCR system (Bio-Rad) and cycle numbers were recorded by CFX Manager v3.1 software. The relative expression levels were normalized to PCR amplification of housekeeping gene *Ppib2* or *Rps16*. Primers used for mRNA expression are list in Supplementary Table 1. Error bars for qPCR analyses were calculated from the independent biological replicates ($n ≥ 3$) indicated in each data set.

### Small GTPase activation assay and immunoblotting analysis

HFSCs were starved and then treated with 300 ng/ml of Wnt5a for 2 h prior to harvesting for small GTPase activation assay. For small GTPase activation assay, Rac1/Cdc42 activation assay combo kit (Cell Biolabs) was performed according to the manufacturer protocol. For immunoblotting analysis, proteins were extracted from cells with RIPA buffer (50 mM Tris-HCl, 100 mM NaCl, 1% of NP-40, 0.5% of NaDeoxycholate, 0.1% of SDS, 1 mM EDTA, protease inhibitor, phosphatase inhibitors). Protein lysates were then loaded into a 4–12% Bis-Tris gel (Thermo Fisher Scientific) and transferred onto a nitrocellulose membrane (Thermo Fisher Scientific). Transferred membranes were blocked with 5% BSA in TBST buffer at RT for 1 h, and then incubated with primary antibodies at 4 °C overnight. After incubation, blots were washed and then incubated with the HRP-conjugated secondary antibodies (Jackson ImmunoResearch) for 1 h. The blots were developed using chemiluminescence detection reagent (SuperSignal West substrates, Thermo Fisher Scientific), and signals were then detected using the Fusion SL imaging system (Vilber) with Fusion-Capt Advance Solo 4S v16.16b software.

### Antibody information

The following antibodies (Abs) were used for FACS: Viability dye F780 (1:200, 65-0865), integrin α6 (1:500, PE-conjugated, clone GoH3, 12-0495), and CD34 (1:100, Alexa Fluor 660-conjugated, clone RAM34, 50-0341) from eBiosciences. The following Abs were used for immunoblotting: ROR2 (Ror2) from Developmental Studies Hybridoma

Bank; ROR1 (clone D6T8C, 16540), p-JNK[T183/Y185] (clone 81E11, 4668), JNK (clone 56G8, 9258), p-PKC[S660] (9371), PKCα (2056), Dvl2 (clone 30D2, 3224), p-LRP6[S1490] (2568), LRP6 (clone C5C7, 2560), p-GSK3β[S9] (9336), GSK3β (clone 27C10, 9315), p-β-catenin[S33/37/T41] (9561), Axin1 (clone C76H11, 2087), CK1 (2655), p-ATM[S1981] (clone D6H9, 5883), ATM (clone D2E2, 2873), p-ATR[S428] (2853), ATR (2790), p-CHK2[T68] (2661), CHK2 (2662), p-CHK1[S345] (clone 133D3, 2348), CHK1 (clone 2G1D5, 2360), p-AMPK[T172] (clone 40H9, 2535), AMPKα (2532), α-tubulin (2144), Lamin B1 (clone D9V6H, 13435), Vinculin (clone E1E9V, 13901) and p-ATM/ATR substrates (2851) from Cell Signaling Technology; p-NRF2[S40] (clone EP1809Y, ab76026) and NRF2 (ab137550) from Abcam; β-catenin (clone 15B8, C7207) and β-actin (clone AC-74, A2228) from Sigma-Aldrich; CD34 (Clone RAM34, 13-0341) from eBiosciences; Rac1 (clone 23A8, 05-389), Cdc42 (07-1466) and GAPDH (Clone 6C5, MAB374) from Merck-Millipore; Krt5 (905901) and Krt15 (833901) from Biolegend; p-GSK3β[Y216] (clone 13A, 612313) from BD Biosciences; HRP-conjugated secondary antibodies (715-035-150, 711-035-152, 712-035-150) from Jackson ImmunoResearch. All primary antibodies used for immunoblotting were at 1:1000 dilution, and secondary antibodies at 1:4000. The following Abs were used for immunostaining: ROR2 (1:100, Ror2) from Developmental Studies Hybridoma Bank; ROR1 (1:100, 16540), p-JNK[T183/Y185] (1:100, 4668), JNK (1:100, 9258), p-PKC[S660] (1:100, 9371), PKCα (1:100, 2056), γH2AX (1:200, BET A300-081A-M) from Cell Signaling Technology; GFP (1:1000, ab13970) from Abcam; β-catenin (1:100, C7207) from Sigma-Aldrich; CD34 (1:100, 13-0341) from eBiosciences; 8-oxoG (1:100, clone 483.15, MAB3560) from Merck-Millipore. Fluorescent dye-conjugated secondary antibodies (703-545-155, 715-545-150, 715-585-150, 711-545-152, 711-585-152, 712-545-150, 712-585-150) from Jackson ImmunoResearch.

### Statistics and reproducibility

Animal experiments were performed in equal numbers of male and female mice. Phenotypes are representative of a minimum of three sex-paired littermates ($n \geq 3$). Experiments were not randomized and investigators were not blinded during data collection. All RT-qPCR results are generated from at least three biological independently animals or independent experiments. All results from histological analysis, immunoblotting and immunofluorescence staining are representative of at least two independent experiments. Graphs were presented using GraphPad Prism 8. Box-whisker plots are presented as the median (the line within the box), the 25th to 75th percentiles (bottom and top lines of the box) and the 10th to 90th percentiles (bottom and top whiskers). Dot plots were presented as the mean ± SEM. Other data were reported as average ± s.d.; * $p < 0.05$, ** $p < 0.005$, *** $p < 0.0005$, or specified $p$ values in figure legends. Statistical analyses were determined by the unpaired two-tailed *Student's t-test* for all experiments.

### Reporting summary

Further information on research design is available in the Nature Research Reporting Summary linked to this article.

## Data availability

All of the data supporting this study are included within the Source Data file that is provided with this paper. Source data are provided with this paper.

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

## Acknowledgements

We thank Nicolas Dauguet for FACS sorting (FACS facility at de Duve Institute); Slobodan Beronja, Valeri Vasioukhin, Frederic Lemaigre and Christophe Pierreux for discussions and comments. A.V. and C.M.R.L. are supported by FRIA fellowships (1.E136.15 to A.V. and 1.E041.16 to C.M.R.L.) from the Fonds de la Recherche Scientifique (FNRS) and Patrimoine fellowships from Université catholique de Louvain (UCLouvain). G.C. is an FSR fellow from UCLouvain and Aspirant fellow (1.A551.19) from FNRS. C.K.C. is supported by the Fondation Contre le Cancer and FSR postdoc fellowship from UCLouvain. W.-H.L. is an independent investigator of FNRS. This work was supported by grants (to W.-H.L.) from FNRS (Ulysse-F.6002.14 and PDR-T.0078.16), Fonds Joseph Maisin (2016-2018), and Fondation Contre le Cancer (FAF-F/2016/792).

## Author contributions

A.V. and W.-H.L. conceived the study and design experiments. A.V. performed the majority of the experiments, analyzed data, and prepared figures. C.L. developed lentiviral infection assays and perform critical experiments during the revision; G.C. conducted a part of histology analyses, cell culture, and RT-qPCR experiments; C.K.C. performed a part of FACS experiments, immunoblotting, cell cycle analysis and ROS measurement; W.-H.L. supervised the study, performed some cell culture experiments, analyzed data, prepared figures and wrote the paper. All authors provided intellectual input and approved the final manuscript.

## Competing interests

The authors declare no competing interests.
