## [Peer Review File · Nature Communications]

ROR2 Regulates Self-Renewal and Maintenance of Hair Follicle Stem CellsReviewers' comments:

Reviewer #1 (Remarks to the Author):

The manuscript attempts to show Ror2-PKC signaling (non-canonical Wnt signaling) is required for stemness gene expression, self renewal, proliferation and migration of HFSCs. They also found Ror2-PKC signaling is upregulated in beta-catenin null HFSCs and this leads to increased HFSC stemness gene expression. Surprisingly, Ror2 is required for canonical Wnt signaling although the elevated Ror2 expression is not sufficient to transduce canonical Wnt signaling. In addition, Ror2 depletion or PKC inhibition in beta-catenin null HFSCs showed differentiation of HFSCs into sebaceous glands. I have several concerns and questions to be addressed before recommending this study for publication in this journal.

1. In Fig. 1d, secondary hair germ seems to be more sensitive to the depletion of Ror2 than bulge stem cells because of their smaller size. Is there a difference in expression level of Ror2 between bulge stem cells and hair germ cells? or is it because HG is closer to dermal papilla which is a source of non-canonical Wnt ligands?
2. In Fig. 2a, Ror2-null HFSCs from telogen which are quiescent showed decreased colony formation efficiency. However, Ror2 is barely expressed in qHFSC based on Fig. 1a and b. Thus, the data supporting the role for Ror2 in quiescent HFSCs need additional confirmation. (E.g. Do telogen hair follicles show any defect such as smaller size? Do HFSC markers decrease in Ror2-null telogen HFSC?)
3. In Fig. 3e, HFSC signature genes decreased in cultured Ror2^{-/-} HFSCs. Have they checked if those gene expression in the Ror2^{-/-} HFSCs are restored in long term culture? In Fig. 5a, HFSC signature genes and Ror2 expression increased in b-cat cKO HFSCs following a long term culture. Does this mean canonical Wnt also can compensate for the loss of Ror2 in a long term culture.
4. In Fig. 4a and b, increased Ror2 expression is correlated increased PKC activity without Wnt5a (0' lane) in b-cat null and Ror2 O/E HFSCs compared to their controls. Please, explain how Ror2 transduces signaling without the ligand.
5. In Fig. 4a, Wnt5a treatment decreased PKC activity in b-cat null HFSCs suggesting downregulation of Ror2-PKC signaling. This is contrast to increased HFSC stemness gene expression.

6. In Fig. 4e, Ror2 null HFSCs decreased Axin2 expression in presence of canonical Wnt3a suggesting Ror2 is required for canonical Wnt signaling. However, underlying mechanism is missing.

7. In Fig. 5a and b, HFSC signature genes and Ror2 expression in long term cultured (over P20) b-catenin null HFSCs. Lien et al (Nature cell biology, 2014) showed b-catenin null HFSCs can self-renew and maintained at earlier passages (by P9) suggesting HFSC signature gene expression may not decrease in b-catenin null HFSCs even in short term culture. Have they checked if b-catenin null HFSCs express those signature genes at earlier passages? If so, is the expression still correlated with increased Ror2 expression?

8. The authors speculated “depletion of Ror2 in beta-catenin KO HFSCs might alter the expression of HFSC stemness genes, which in turn primed b-catenin-null HFSCs for differentiation into a wrong fate.” However, Ror2-null HFSCs also showed decreased HFSC stemness gene expression in Fig. 1e and 3e but they do not differentiate into sebocytes in Fig. 6a. Thus, there should be other factors involved in their fate change with reduced HFSC stemness gene expression. However, the authors need to provide an insight into the mechanism underlying differentiation of HFSCs into sebocytes in absence of Wnt signaling.

9. In a lane for Ror2 expression of qHFSCs and aHFSCs in Fig. 1a and in a lane for p-PKC of beta-catenin KO Scr compared to other lanes in Fig. 4c, background intensity looks different.

10. In immunoblotting (Fig. 1a and b, Fig. 1d, Fig. 3a and b, Fig. 4a, b and c), Ror2 expression level varies in HFSCs although b-actin level is considered. Is this because there is a difference in Ror2 expression between freshly isolated HFSCs and cultured HFSCs?

11. In Fig. 1b, there seems to be two bands of upper band of JNK which is not seen in Fig. 1a and C.

12. In Fig. 4b, there is two bands in Ror2 expression which is not seen in other immunoblotting of Ror2. Did other immunoblotting show upper band only?

Reviewer #2 (Remarks to the Author):

Beta-catenin dependent WNT signaling has a well-established role in hair cycling, but mechanisms by which beta-catenin independent WNT signaling contribute to hair cycling remain understudied. In this manuscript, Veltri et al. aim to fill this gap in knowledge and make several new observations regarding the role of ROR2 in hair follicle stem cell (HFSC) maintenance, namely that ROR2 expression levels appear to be increased in both activated vs. quiescent HFSC as well as in beta-catenin knockout HFSC. Veltri et al. use a combination of biochemical, molecular, and cell behavioral assays to suggest that ROR2 signaling regulates a variety of HFSC activities, including self-renewal, differentiation, migration and proliferation, which ultimately influence hair cycling. While the data are potentially interesting, the quality of several experiments and their interpretation need improvement/clarification, as commented below.

Major points:

Figure 1: The quantification of ROR2 expression and the activation of downstream effectors is crucial to the conclusions made by the authors. However, the overall quality of the Western blots is rather poor. Some of the signals appear to be near background (this is especially apparent when looking at the full scans in the supplementary materials). Optimization of the Western conditions is therefore needed to prove blot quality. It is also important that the signal intensity be quantified from multiple experimental replicates and analyzed for statistical significance. Based on the data shown in Figure 1c, the authors concluded that ROR2-associated signaling is profoundly reduced in FACS-purified ROR2 CKO HFSCs. However, the Western blots actually show that while the p-JNK and p-PKAlpha signals decrease in ROR2 CKO HFSCs, the total JNK and PCKalpha protein levels are also similarly decreased; it is therefore inconclusive whether ROR2-JNK or ROR-PCK signaling are actually reduced. The authors should also clearly describe what the ROR2 CTRL animals are. The scheme shown in Figure S1a is not sufficient. What exactly were the genotypes of the parent mice used to generate the ROR2 CTRL and CKO progeny? Were the ROR2 CTRL mice littermates of the CKO animals? What were their genotypes? In order to ascertain that Cre-driver allele itself does not contribute to the observed phenotype, the CTRL mice should carry the Cre allele as well (e.g. K15CrePGR; ROR2f/+;ROSA26LSL-YFP).

Figure S1: In my opinion, the data presented here are crucial to the main conclusions of the paper and should be shown in the main figures. Figure S1d is confusing. The authors used an anti-ROR2 to perform the staining. However, instead of showing that expression of the ROR2 protein is indeed abrogated in the ROR2 CKO sample, they used the "ROR2" autofluorescence signals to mark hair shafts and sebaceous glands.

Figure 3: The quality of the Western blots shown in this figure is better. However, some issues concerning data interpretation need to be addressed. Based on the changes in Rho GTPase activities

show in Figure 3b, the author tested whether migration is affected (Figure 3c). However, it appears that it is the basal activity of Rac1 and Cdc42 that is altered in ROR2 null cells, and Wnt5a stimulation has not further effects in this context. However, in the migration assay, the knocking out ROR2 alone does not affect migration, and the migration phenotype is only apparent in ROR2 nulls cells after Wnt5a stimulation. This discrepancy in regulation between the Rho GTPase and migration assays needed to be addressed. In Figure 3d, while the effects of GF109203X could be interpreted as PKC acting downstream of ROR2, as the authors did, it could also be explained by PKC acting in an independent pathway to regulate HFSC proliferation. In fact, the observation that GF109203X treatment produces a stronger effect than vehicle-treated ROR2 null cells is more consistent with the latter possibility.

Figure 4: This and the next two figures test the possibility that upregulated ROR2 signaling might compensate for the loss of beta-catenin-dependent canonical Wnt signaling. The part of the story seems somewhat disconnected from the first part of the study. While the upregulation of ROR2 expression in beta-catenin KO HFSCs is convincing and potentially interesting, the experiments do not offer deeper mechanistic insights into how ROR2, a noncanonical receptor, could function to mediate WNT3A-induced canonical Wnt signaling. Their observation that ROR2 upregulation is insufficient to functionally compensate for the loss of beta-catenin also diminishes the significance of the experiment. Lastly, this entire second part of the study is conducted in the context of beta-catenin KO, which does not further corroborate how ROR2 signaling functions to control HFSC biology under normal physiological context, as shown in the first part of the study.

Figure 6: The original data used to construct the scheme shown in Figure 6a should be presented. In Figure 6d and 6e, data for beta-catenin single CKO mutant should be included as an additional control for comparison. In their previous study (Ref 45), the authors observed that conditionally knocking out beta-catenin alone was sufficient to induce inappropriate sebocyte differentiation. Why do they not observe this effect in the present study (Figure 6a)?

Minor points:

- Several typos within text (e.g., “confirm our previous discoverY” rather than discoveries, “native niche by generating inducible... mice” rather than “a inducible... mice”, etc.)
- Figures:
 - o 2b, error bars for ROR2 control appear to be exactly the same for each passage number; please double check.
 - o 4a, the pJNK bands were partially cut off, which makes the quantification less reliable.
 - o 4d, 5a, 5c, shades of green are too similar to easily distinguish one condition from the other
 - o 4e, Y-axis legend is confusing and could be changed to “Fold of induction of Wnt3a genes” or something similar to make interpretation of results easier to understand

- Other:

- o At least two publications are listed twice within the references (specifically 37/52 and 51/58)

Reviewer #3 (Remarks to the Author):

Extensive studies have been performed in exploring the Wnt/b-catenin signaling pathway in activation of hair follicle stem cells (HFSCs) during hair regeneration and tissue homeostasis, but little is known about the function of β -catenin-independent Wnt signaling in HFSCs. In the manuscript titled “Ror2-dependent Signaling Regulates Self-Renewal and Maintains Stem Cell Identity in Hair Follicle”, Veltri et al. investigated the role of non-canonical signaling pathway receptor ROR2 in regulating the activation of HFSCs during hair regeneration and the maintenance of HFSCs identity using in vivo and ex vivo experiments. First, using a ROR2 conditional knockout mice model, they show that ROR2 is required for HFSCs activation during hair regeneration. The expression of HFSC signature genes is decreased in the bulge cells of ROR2 knockout mice, as well as in the cultured ROR2-depleted HFSCs. Without ROR2, the cultured HFSCs cannot self-renew and maintain their identity during long time culture. In addition, they generated the ROR2-deficient HFSCs, through which they show that ROR2 transduces Wnt5a-dependent signaling and regulates HFSC migration and proliferation. Since previous study shows that b-catenin-deficient HFSCs can maintain their stemness, authors found that ROR2 is highly expressed in b-catenin-deficient HFSCs and is required for b-catenin-deficient HFSCs to maintain their identity through PKC signaling, which has been shown to maintain HFSC identity (Niessen et al., 2013, J Cell Biol). ROR2 removal or PKC inhibition in b-catenin-deficient HFSCs not only leads to the reduced stemness of HFSCs but also turns HFSCs into a sebaceous gland fate. In general, the approaches used in this study can be updated but the results described in this study are novel, and they provide new insight into the mechanisms by which non-canonical signaling pathway in regulating stem cell self-renewal and identity maintenance.

Major points:

1. HFSC activation and identity maintenance seem to be in the opposite directions for stem cell fate. Many previous studies show that loss of stemness-related genes in HFSCs leads to activation of HFSCs. This study shows that ROR2 is both required for HFSCs activation and stemness maintenance. Loss of ROR2 in HFSCs leads to decreased stemness-related genes but delayed anagen entry of the hair follicles. Thus, it is unclear how ROR2 balances these two functions in HFSCs in the present study. Moreover, if ROR2 functions to maintain the identity of HFSCs, ROR2 should also have a higher expression in telogen, whereas it is not the case in the present study?
2. Most of the studies are based on cultured HFSCs, which do not have the stem cell niche environment, therefore, is different from the in vivo case. This is one of the major concerns of this study.

3. Authors show decreased proliferation and decreased expression of HFSCs signature genes in bulge stem cells of the K15Cre-ROR2 mice, whereas the hair follicles just show delayed rather than abolished regeneration, indicating other compensative mechanisms for hair regeneration. This is not clear in this study.
4. What is the phenotype of the mice with long term deletion of ROR2 in HFSCs? This should be characterized.
5. Author claimed that Ror2 is diminished in the YFP+ bulge and hair germ (HG) of Ror2 cKO HFSCs (Fig. S1d), which is unconvincing. In addition, please show ROR2 and PKC expression during telogen and anagen initiation (multiple stages) using immunofluorescent staining.
6. Fig. 1f, you hypothesize that some of the activated Ror2 cKO HFSCs lost HFSC markers, e.g. CD34, used in the FACS analysis, and conclude that Ror2 can modulate HFSC identity and maintenance by regulating the expression of HFSC markers. Please make sure if this is true. First, it looks that Ror2 cKO has a smaller bulge from Fig. 1d. Second, please do more morphological and molecular characterization in early anagen bulge of Ror2 cKO through staining.
7. Authors largely used the b-catenin-deficient mouse model, whereas it is unclear why Ror2 is upregulated in b-catenin knockout mice.
8. Using a similar Lgr5-CreERT2;aPKCfl/fl;Rosa26RLacZ transgenic mouse model, previous study showed that loss of aPKC in HFSCs results in increased proliferation and loss of quiescence in the bulge, turning HFSCs into a sebaceous gland and epidermis cell fate (Niessen et al., 2013, J Cell Biol). In the present study, authors also show that PKC functions downstream of ROR2 to regulate HFSCs activity. Please explain the inconsistency between this study and the previous study.
9. Moreover, please explain why HFSCs only differentiate into sebaceous gland in Ror2 and β -cat double cKO HFSCs, rather than in ROR2-deficient mice, in which PKC is downregulated.
10. Fig. 6e, authors show that additional ablation of Ror2 in β -cat cKO HFSCs (Ror2/ β -cat dKO) resulted in downregulation of HFSC signature genes, which might have tipped the balance of arrested β -cat cKO HFSCs towards differentiation. Whereas authors show that both Ror2 and b-catenin knockout mice display downregulation of HFSC signature genes without differentiation into SG. So other key genes should be identified.

Minor points:

1. Fig 1a-c is unconvincing, as b-actin is higher in aHFSC group compared to the qHFSC group. Also in many cases, the expression of b-Actin and GAPDH is different between different groups. In addition, many bands are barely legible throughout the study. For example, the blot for Figs. 1a & b etc. should be replaced.
2. Results between Fig. S1b and Fig. S1e should be consistent. How do you define anagen entry, through morphology, or through HFSCs (e.g., second hair germ cells) proliferation during hair regeneration?
3. Page 9 and Fig. S3a, fluorescent images should be shown to display the Cre activity which demonstrates the ROR2 knockdown efficiency in cultured HFSCs.

4. Fig. 3f, Krt1 and Krt10 are epidermal differentiation markers. Please make sure if they are expressed in HFSCs, or this is due to impurity of the sorted HFSCs. In addition, lower expression of these markers should mean lower differentiation state of the HFSCs in ROR2-deficient cells, thus better maintenance of HFSCs identity? But this is not the case in this study.

5. Page 11, authors state that the upregulation of Ror2 was also observed in arrested β -cat cKO HFSCs residing in their native niche during prolonged telogen. Please show an image using immunofluorescent staining.

6. Page 14, 1 week after application of PKC inhibitor, β -cat cKO HFSCs had already undergone sebocyte differentiation. Please show the data using staining.

7. Authors state that ROR2 knockout mice have a lower ability to migrate. In this case, how do the HFSCs in Ror2/ β -cat dKO change the cell fate and migrate into SG to differentiate?

Responses to Reviewers on manuscript NCOMMS-19-20272A-Z

Overall

We appreciated all three reviewers' comments as each of them was constructive in highlighting the strengths of our findings but also in identifying shortcomings in our initial submission which by rectifying have greatly improved our manuscript. All of our reviewers appreciated that we provided interesting data and new insight into the mechanisms by which ROR2-mediated signaling in regulating hair follicle stem cell (HFSC) self-renewal and maintenance. However, it was also clear from several of their comments that our manuscript fell short in some places in providing clear explanation of our experiments and datasets. Therefore, we clarified the confusing points and provided a better description of our data with additional experiments and supporting data as delineated below.

All of our three reviewers made helpful comments. They all raised the concerns on: (1) the expression level and pattern of ROR2 in anagen and telogen HFs; (2) the quality of our immunoblotting and immunostaining results; (3) how ROR2 balances HFSC activation and maintenance these two functions; (4) the significance of ROR2 upregulation in long-term cultured β -cat cKO HFSCs; and (5) the underlying mechanisms differentiation of β -cat cKO HFSCs into sebocytes in absence of *Ror2*. All reviewers had valuable suggestions for points that necessitated clarification and/or expanding upon several of our experiments. We've successfully addressed each of the reviewers' suggestions, and we provide new substantiating data and high-quality images to support our findings. The new data are now in Figs. 1a-b, 2c-f, 2h, 4e, 5a-j, 6b, 7b-c, S1a-f, S3a-b, S4b, S5a-c, S6a-b, S7a-c, S8a-b and S9a-c. To summarize:

(1) The expression level and pattern of ROR2 in anagen and telogen HFs:

Using whole-mount immunofluorescence staining of HFs at anagen onset and telogen for ROR2, we showed that ROR2 is expressed in HFSCs and neighboring cells in the bulge, but relatively low in the secondary hair germ. By performing immunoblotting with 4 independent experiments in which 4 different sets of animals were used, we demonstrated that ROR2 expression in the bulge is higher in HFs at anagen onset than those in telogen. The elevated ROR2 expression in anagen HFs was also detected by whole-mount immunostaining. These datasets are presented in Figs. 1a-b, S1a-c.

(2) Quality of the immunoblotting and immunostaining results:

The quality of immunostaining and immunoblotting in the current manuscript was greatly improved, and all immunoblotting results presented in the manuscript were repeatedly shown from several independent biological replicates ($n \geq 3$). As for immunostaining, we perform immunofluorescence staining with whole-mount skin tissues, which give better staining signals without interference by signals from the dermis. These datasets are presented in Figs. 1a-b, S1a-c.

(3) How ROR2 balances HFSC activation and maintenance these two functions:

In the current manuscript, we found that ROR2, as a Wnt receptor, not only transduces non-canonical Wnt signaling, but is also required for canonical Wnt-induced transcriptional activation (Figs. 2c-d, 4c). Since canonical Wnt signaling is necessary for HFSC activation and anagen entry, downregulation of canonical Wnt signaling in *Ror2*-deficient HFSCs would lead to a delay in HFSC activation. In addition to HFSC activation, we also demonstrated that ROR2 plays an essential role in regulation of ATM/ATR-dependent DNA damage response (Fig. 5), which is indispensable for long-term maintenance of HFSCs. During the normal homeostasis or culture condition, HFSCs undergo self-renewal and tissue regeneration which make them vulnerable to DNA damage caused by external and/or internal factors. When *Ror2* was depleted in HFSCs, ATM/ATR-dependent DDR was compromised in *Ror2*-deficient HFSCs, which led to accumulation of DNA damage and excess ROS production. This deficiency eventually resulted in slow growth of HFSCs and inability in HFSC

long-term maintenance. Therefore, ROR2 plays the dual role in regulation of HFSC activation and maintenance through its independent functions in controlling activation of canonical Wnt signaling and ATM/ATR-dependent DDR, respectively.

(4) The significance of ROR2 upregulation in long-term cultured β -cat cKO HFSCs:

In our previous manuscript, long-term cultured β -cat Ctrl and cKO HFSCs were representatives of different individual colonies which were generated from FACS-purified β -cat Ctrl or cKO HFSCs. However, they gave significant variation in the gene expression within the group of β -cat Ctrl or cKO HFSCs. Hence, the results obtained from pairwise analysis of independent HFSC clones (Figs. 4, 5, S4, and S5 in the previous manuscript) mislead the data interpretation. Therefore, all results generated from long-term cultured β -cat Ctrl and cKO HFSCs are removed. In the current manuscript, we focused only on the findings and data generated from the *in vivo* mouse model, including the mouse phenotypes and freshly purified β -cat Ctrl and cKO HFSCs. By performing analyses with FACS-purified β -cat cKO HFSCs at different stages of hair cycle, we found that *Ror2* expression became upregulated in β -cat cKO HFSCs at late telogen and aged HF where the expression of HFSC signature genes went down (Fig. 6b). This analysis was performed with HFSCs purified from several independent pairs of animals. The upregulated ROR2 in late telogen and aged β -cat cKO HFSCs was also detected by the whole-mount immunostaining (Fig. S6). Thus, using the *in vivo* model is a better option to analyze the role of ROR2 and ROR2-dependent signaling in β -cat cKO HFSCs.

(5) The underlying mechanisms differentiation of β -cat cKO HFSCs into sebocytes in absence of *Ror2*:

As described above, we showed that ROR2 upregulation is associated with downregulation of HFSC stemness genes in the late telogen and aged β -cat cKO HFSCs. Thus, we speculated that upregulated ROR2 is required for the long-term maintenance of β -cat cKO HFSCs where HFSC stemness gene expression were lower. Indeed, depletion of *Ror2* caused β -cat cKO HFSCs leaving quiescence and undergoing sebocyte differentiation (Figs. 6c-d), indicating that upregulated ROR2 keeps β -cat cKO HFSCs in quiescence, which protects HFSCs against activation and subsequent differentiation. It was already demonstrated by several studies that the HF fate was determined by β -catenin-dependent signaling (Gat *et al.*, *Cell*, 1998; Huelsken *et al.*, *Cell*, 2001; Lo Celso *et al.*, *Development*, 2004; Lien *et al.*, *Nat Cell Biol*, 2014). Depletion of β -catenin in the developing epidermis (K14-Cre driven) led to HF degeneration and sebocyte formation (Huelsken *et al.*, *Cell*, 2001). However, when β -catenin was depleted specifically in HFSCs (K15-Cre driven), β -cat cKO HFSCs remained quiescent throughout the rest of the life without undergoing differentiation unless HFSCs were activated by depilation (Lien *et al.*, *Nat Cell Biol*, 2014; our manuscript Fig. 6a). In general, depilation induces a wound response of HFSCs, which lowers down the expression of HFSC stemness genes and then initiates HFSC proliferation and subsequent differentiation. Upon depilation, β -cat cKO HFSCs were able to proliferate, but differentiated into sebocyte fate instead of HF fate (Lien *et al.*, *Nat Cell Biol*, 2014). Thus, the wrong fate determination of *Ror2*/ β -cat dKO HFSCs was caused by the synergistic effects of β -catenin loss and *Ror2* depletion. In other words, *Ror2* depletion challenges the quiescent state of β -cat cKO HFSCs by lowering down the threshold of β -cat cKO HFSCs for activation, and activated β -cat cKO HFSCs then undergo differentiation into sebocyte fate. This effect is similar to the phenomenon of depilation on β -cat cKO HFSCs (Lien *et al.*, *Nat Cell Biol*, 2014).

Below, we describe in greater detail the changes made in response to each reviewer's point. We'd like to take this opportunity to thank the reviewers for their very helpful comments, which have been valuable in strengthening the impact of our study and raising its level to be of interest to future *Nature Communications* readers.

Reviewer #1:

The manuscript attempts to show Ror2-PKC signaling (non-canonical Wnt signaling) is required for stemness gene expression, self-renewal, proliferation and migration of HFSCs. They also found Ror2-PKC signaling is upregulated in beta-catenin null HFSCs and this leads to increased HFSC stemness gene expression. Surprisingly, Ror2 is required for canonical Wnt signaling although the elevated Ror2 expression is not sufficient to transduce canonical Wnt signaling. In addition, Ror2 depletion or PKC inhibition in beta-catenin null HFSCs showed differentiation of HFSCs into sebaceous glands. I have several concerns and questions to be addressed before recommending this study for publication in this journal.

1. In Fig. 1d, secondary hair germ seems to be more sensitive to the depletion of Ror2 than bulge stem cells because of their smaller size. Is there a difference in expression level of Ror2 between bulge stem cells and hair germ cells? or is it because HG is closer to dermal papilla which is a source of non-canonical Wnt ligands?

We thank the reviewer for this question. As shown in Fig. 1a of our current manuscript, ROR2 was expressed at a lower level at HG as compared to bulge HFSCs. This was observed from both anagen and telogen HF. The quantification for anagen HF was provided at fig.1A below. Thus, we do not think that HG is more sensitive than bulge to Ror2 depletion.

In Fig. 1d (now in Fig. 2a), we show that upon depilation fewer YFP+ Ror2 cKO HFSCs underwent proliferation measured by the EdU incorporation. In fact, there were also fewer YFP+ HG cells incorporated by EdU at Ror2 cKO HF (see fig.1B at right). Due to the proximity of HG to dermal papilla, progenies at HG are the first group of cells responding to the signals and getting activated at the onset of anagen. Activated HG cells first expand the population, which increases the size of the HG. As fewer Ror2 cKO HG cells underwent proliferation, it is expected to see the HG of Ror2 cKO HF smaller in size as compared to Ror2 Ctrl HF.

2. In Fig. 2a, Ror2-null HFSCs from telogen which are quiescent showed decreased colony formation efficiency. However, Ror2 is barely expressed in qHFSC based on Fig. 1a and b. Thus, the data supporting the role for Ror2 in quiescent HFSCs need additional confirmation. (E.g. Do telogen hair follicles show any defect such as smaller size? Do HFSC markers decrease in Ror2-null telogen HFSC?)

We thank the reviewer for bringing up this concern. Indeed, the expression level of ROR2 is lower at telogen HFSCs *in vivo*. In the current manuscript, we included 4 independent datasets for the quantification of relative levels of ROR2 protein in FACS-purified qHFSCs and aHFSCs (now in Figs. 1b, S1c). However, ROR2 was not completely absent in qHFSCs of telogen HF as demonstrated by the whole-mount immunostaining for ROR2 shown in Fig. 1a (quantification in Fig. S1b) of our current manuscript, as well as by the longer exposure of the western blot (fig.2 at right).

The colony formation assay shown in Fig. 2a (now in Fig. 3a) was performed with quiescent HFSCs that were FACS-purified from telogen HFs and then cultured in the medium promoting proliferation, which makes the microenvironment of these purified “quiescent” HFSCs resembling the onset of anagen. Thus, reduced colony formation of the “quiescent” HFSCs was due to the deficiency in self-renewal/ proliferation. To examine the maintenance ability of HFSCs, following the colony formation assay we performed the long-term culture passaging experiment shown in Fig. 2b (now in Fig. 3b). To avoid the confusion, we clarified the condition of these experiments in page 9-10 of the current manuscript.

Moreover, to analyze HFSCs *in vivo*, we conducted flow cytometry analysis to measure the recovery of YFP+ HFSCs after one round of HF regeneration that was either induced by depilation (now in Fig. 2g) or after hair cycling (now in Fig. 2h). As shown, the pool of YFP+ *Ror2* cKO HFSCs that recovered from depilation-induced HF regeneration or hair cycling was reduced as compared to control HFSCs. These results indicate that ROR2 is required for the maintenance of a HFSC population. In supporting this finding, we also showed that the expression levels of HFSC stemness genes in depilation-activated *Ror2* cKO HFSCs (now in Fig. 2e) or in quiescent *Ror2* cKO HFSCs of the late telogen HFs (now in Fig. 2f) were significantly downregulated. These data show the necessity of ROR2 in maintaining HFSC identity regardless the status of HFSCs.

3. In Fig. 3e, HFSC signature genes decreased in cultured *Ror2*^{-/-} HFSCs. Have they checked if those gene expression in the *Ror2*^{-/-} HFSCs are restored in long term culture? In Fig. 5a, HFSC signature genes and *Ror2* expression increased in *b-cat* cKO HFSCs following a long term culture. Does this mean canonical Wnt also can compensate for the loss of *Ror2* in a long term culture.

In fact, the expression of HFSC signature genes remained downregulated in the long-term cultured *Ror2*^{-/-} HFSCs (>25 passages, see fig.3 at right). In the current manuscript, we showed that ROR2 not only transduces non-canonical Wnt signaling, but is also required for canonical Wnt-activated transcriptional activity (now in Fig. 4). Given that ROR2 is essential to transduce both canonical and non-canonical Wnt signaling, activation of canonical Wnt signaling alone is not sufficient to compensate for the loss of ROR2-dependent regulation, especially on non-canonical Wnt pathways.

The *in vivo* mouse model is a better option to analyze the role of ROR2 in *β-cat* cKO HFSCs. In our previous manuscript, long-term cultured *β-cat* Ctrl and cKO HFSCs were representatives of different individual colonies which were generated from FACS-purified *β-cat* Ctrl or cKO HFSCs. However, they gave significant variation in the gene expression within the group of *β-cat* Ctrl or cKO HFSCs. Therefore, all results generated from cultured *β-cat* Ctrl and cKO HFSCs (Fig. 4, 5, S4, and S5 in the previous manuscript) are removed. In the current manuscript, we focused only on the findings and data generated from the *in vivo* mouse model, including the mouse phenotypes and freshly purified *β-cat* Ctrl and cKO HFSCs via FACS (Figs. 6, 7, S6 and S9 in the current manuscript).

Due to the above-mentioned concern, we removed the entire datasets showing the differential expression of HFSC signature genes from the long-term cultured *β-cat* cKO HFSCs in Fig. 4 and 5 of the previous manuscript. In the current manuscript, we focused on *in vivo β-cat* cKO HFSCs by performing analyses with FACS-purified *β-cat* cKO HFSCs at different stages of hair cycle (now in Fig. 6b). We showed that *β-cat* cKO HFSCs at early telogen did not show changes in the expression of HFSC signature genes as well as *Ror2*; however, *Ror2* expression became upregulated in *β-cat*

cKO HFSCs at late telogen and aged HFs where the expression of HFSC signature genes went down. This analysis was performed with HFSCs purified from several independent pairs of animals. The upregulated ROR2 in late telogen and aged β -cat cKO HFSCs was also detected by the whole-mount immunostaining (now in Fig. S6). Thus, using the *in vivo* model is a better option to analyze the role of ROR2 and ROR2-dependent signaling in β -cat cKO HFSCs.

4. In Fig. 4a and b, increased Ror2 expression is correlated increased PKC activity without Wnt5a (0' lane) in b-cat null and Ror2 O/E HFSCs compared to their controls. Please, explain how Ror2 transduces signaling without the ligand.

AND

5. In Fig. 4a, Wnt5a treatment decreased PKC activity in b-cat null HFSCs suggesting downregulation of Ror2-PKC signaling. This is contrast to increased HFSC stemness gene expression.

We thank the reviewer for reviewer's questions (#4 & #5) regarding the datasets presented in the previous version of our manuscript. Due to the concern described in Q#3 above, we removed the entire datasets generated from the long-term cultured β -cat cKO HFSCs in Fig. 4 and 5 of the previous manuscript. We again apologize for these inadequate datasets.

6. In Fig. 4e, Ror2 null HFSCs decreased Axin2 expression in presence of canonical Wnt3a suggesting Ror2 is required for canonical Wnt signaling. However, underlying mechanism is missing.

In the current manuscript, we confirmed the finding showing that ROR2 is required for Wnt3a-induced expression of canonical Wnt target genes (now in Fig. 4c). This requirement is also found in depilation-activated HFSCs as well as in HFSCs at anagen onset, where canonical Wnt target genes were downregulated upon *Ror2* depletion (now in Figs. 2c, 2d).

The underlying mechanism on how ROR2 activates canonical Wnt signaling is not clear yet. However, according to the previous study (Li *et al.*, *BMC Mol Biol*, 2008), overexpressed ROR2 could cooperate with Fzd2 to mediate Wnt3a-stimulated canonical Wnt signaling. The other speculated mechanism could be via activation of JNK downstream, AP-1 proteins. It has been reported that c-Jun, a member of AP-1 complex, interacts with TCF4/ β -catenin complex to regulate expression of target genes in a JNK-dependent manner. In the absence of *c-Jun*, β -catenin signaling is incapable of inducing intestinal tumor development (Nateri *et al.*, *Nature*, 2005). This study indicates that c-Jun activity (Ser63) is required for c-Jun-TCF4 interaction, which is essential for β -catenin-dependent signaling. In the current manuscript, we showed that Wnt3a stimulation could also induce JNK activation (now in Fig. 4a), which implies the activation of AP-1 proteins. Indeed, as shown in fig. 4 at top, the activity of c-Jun was also upregulated in HFSCs upon Wnt3a stimulation, but this activation was compromised in *Ror2*^{-/-} HFSCs, similar to the pattern of its upstream JNK protein. Thus, we speculate the reduction in c-Jun activity might be one of causes leading to the downregulation of β -catenin-dependent Wnt signaling in the absence of ROR2.

7. In Fig. 5a and b, HFSC signature genes and Ror2 expression in long term cultured (over P20) b-catenin null HFSCs. Lien et al (Nature cell biology, 2014) showed b-catenin null HFSCs can self-renew and maintained at earlier passages (by P9) suggesting HFSC signature gene expression

may not decrease in b-catenin null HFSCs even in short term culture. Have they checked if b-catenin null HFSCs express those signature genes at earlier passages? If so, is the expression still correlated with increased Ror2 expression?

We appreciate the reviewer's comment on cultured β -cat Ctrl and cKO HFSCs by comparing with the previously published work to help us clarify the confusion. In the previous work (Lien et al., *Nat Cell Biol* 2014), authors depleted β -catenin in HFSCs of HF at the 2nd telogen, and then analyzed FACS-purified β -cat Ctrl and cKO HFSCs from 10 days post-depleted HF. They showed that HFSC signature genes remained expressed in FACS-purified β -cat cKO HFSCs by performing the immunostaining with the whole-mount tissues for HFSC markers (Fig. 1 in the reference) as well as qPCR analysis with FACS-purified HFSCs (Fig. S3c in the reference). However, the expression levels of these genes had never been shown in cultured β -cat Ctrl and cKO HFSCs. By looking back the qPCR results from several clones of cultured β -cat Ctrl and cKO HFSCs at early (P1), late passages (P12) established at that time, the variation in the expression levels of these genes was already detected within the group of β -cat Ctrl or cKO HFSCs (see fig. 5 below). This variation was further enhanced in the long-term cultured β -cat Ctrl and cKO HFSCs (P25). Hence, we concluded that the results obtained from pairwise analysis of independent HFSC clones could cause the confusion and mislead the data interpretation. Therefore, as described in Q#3, we decided to remove the datasets generated from those HFSC clones (Figs. 4 & 5 in the previous manuscript) and focused on β -cat cKO HFSCs *in vivo* (Fig. 6 in the current manuscript).

To exclude the differences between colonies, HFSCs depleted for β -catenin should be generated from Cre-transduced HFSCs carrying conditional β -cat alleles, similar to what we did to generate *Ror2*^{-/-} HFSCs. In this way, β -cat^{+/+} and β -cat^{-/-} HFSCs contain the same genetic background, except for β -catenin. Thus, the changes in gene expression will be more specific for the depletion of β -catenin. Nevertheless, our analysis with *in vivo* β -cat cKO HFSCs addressed the reviewer's question that the downregulation of HFSC signature genes in β -cat cKO HFSCs correlated with elevated levels of *Ror2* expression (now in Figs. 6b, S6).

8. The authors speculated “depletion of *Ror2* in beta-catenin KO HFSCs might alter the expression of HFSC stemness genes, which in turn primed b-catenin-null HFSCs for differentiation into a wrong fate.” However, *Ror2*-null HFSCs also showed decreased HFSC stemness gene expression in Fig. 1e and 3e but they do not differentiate into sebocytes in Fig. 6a. Thus, there should be other factors involved in their fate change with reduced HFSC stemness gene expression. However, the authors need to provide an insight into the mechanism underlying differentiation of HFSCs into sebocytes in absence of Wnt signaling.

We appreciate this reviewer for bringing up this unclear interpretation. In the current manuscript, we discovered that the upregulation of ROR2 was associated with downregulation of HFSC stemness genes in the late telogen and aged β -cat cKO HFSCs (now in Figs. 6b, S6). Thus, we speculated that upregulated ROR2 is required for the long-term maintenance of β -cat cKO HFSCs

where HFSC stemness gene expression were lower. Indeed, depletion of *Ror2* caused β -cat cKO HFSCs leaving quiescence and undergoing sebocyte differentiation (now in Figs. 6c, 6d), indicating that upregulated ROR2 keeps β -cat cKO HFSCs in quiescence, which protects HFSCs against activation and subsequent differentiation.

It was already demonstrated by several studies that the HF fate was determined by β -catenin-dependent signaling (Gat *et al.*, *Cell*, 1998; Huelsken *et al.*, *Cell*, 2001; Lo Celso *et al.*, *Development*, 2004; Lien *et al.*, *Nat Cell Biol*, 2014). Depletion of β -catenin in the developing epidermis (K14-Cre driven) led to HF degeneration and sebocyte formation (Huelsken *et al.*, *Cell*, 2001). However, when β -catenin was depleted specifically in HFSCs (K15-Cre driven), β -cat cKO HFSCs remained quiescent throughout the rest of the life without undergoing differentiation (Lien *et al.*, *Nat Cell Biol*, 2014; our manuscript Fig. 6a) unless HFSCs were activated by depilation (Lien *et al.*, *Nat Cell Biol*, 2014). In general, depilation induces a wound response of HFSCs, which lowers down the expression of HFSC stemness genes and then initiates HFSC proliferation and subsequent differentiation. Upon depilation, β -cat cKO HFSCs were able to proliferate, but differentiated into sebocyte fate instead of HF fate (Lien *et al.*, *Nat Cell Biol*, 2014). Thus, the wrong fate determination of *Ror2*/ β -cat dKO HFSCs was caused by the synergistic effects of β -catenin loss and *Ror2* depletion. In other words, *Ror2* depletion challenges the quiescent state of β -cat cKO HFSCs by lowering down the threshold of β -cat cKO HFSCs for activation, and activated β -cat cKO HFSCs then undergo differentiation into sebocyte fate. This effect is similar to the phenomenon of depilation on β -cat cKO HFSCs (Lien *et al.*, *Nat Cell Biol*, 2014).

9. In a lane for *Ror2* expression of qHFSCs and aHFSCs in Fig. 1a and in a lane for p-PKC of beta-catenin KO Scr compared to other lanes in Fig. 4c, background intensity looks different.

We thank the reviewer for pointing this out. In the current manuscript, the ROR2 expression of qHFSCs and aHFSCs were analyzed and quantified from 4 independent experiments in which 4 different sets of animals were used (now in Figs. 1b, S1c). In addition to immunoblotting, whole-mount immunofluorescence staining of anagen and telogen HFs for ROR2 also supports this finding in which ROR2 expression level is higher in the bulge of anagen HFs (now in Figs. 1a, S1b). The datasets about cultured β -cat cKO HFSCs in Fig. 4 were removed due to the above-mentioned reason. All of western blotting results were re-analyzed throughout the manuscript. The full blots of WB were presented in Fig. S10.

10. In immunoblotting (Fig. 1a and b, Fig. 1d, Fig. 3a and b, Fig. 4a, b and c), *Ror2* expression level varies in HFSCs although b-actin level is considered. Is this because there is a difference in *Ror2* expression between freshly isolated HFSCs and cultured HFSCs?

Indeed, ROR2 expression levels were higher in cultured HFSCs than FACS-purified HFSCs mainly due to the limitation in the amount of HFSCs purified by FACS. It is also partly due to the status of HFSCs as HFSCs growing in the culture medium are more proliferative than FACS-purified quiescent HFSCs.

11. In Fig. 1b, there seems to be two bands of upper band of JNK which is not seen in Fig. 1a and C.

Due to limited cell numbers that we could purify from one animal using FACS, one biological replicate is only sufficient for one western blot that was analyzed for various proteins. Thus, the upper band of JNK in Fig. 1b was a leftover from the previous blotting against ROR2. However, this set of western blotting results was removed from the current manuscript.

12. In Fig. 4b, there is two bands in Ror2 expression which is not seen in other immunoblotting of Ror2. Did other immunoblotting show upper band only?

Due to the reason mentioned above, the datasets about cultured β -cat cKO HFSCs in Fig. 4 were removed. Regarding original Fig. 4b, two bands of ROR2 (labeled as stars) were overexpressed ROR2, not endogenous ROR2 (labeled as an arrow). Our speculation is that two bands of overexpressed ROR2 were likely caused by the post-translational modifications.

Reviewer #2:

Beta-catenin dependent WNT signaling has a well-established role in hair cycling, but mechanisms by which beta-catenin independent WNT signaling contribute to hair cycling remain understudied. In this manuscript, Veltri et al. aim to fill this gap in knowledge and make several new observations regarding the role of ROR2 in hair follicle stem cell (HFSC) maintenance, namely that ROR2 expression levels appear to be increased in both activated vs. quiescent HFSC as well as in beta-catenin knockout HFSC. Veltri et al. use a combination of biochemical, molecular, and cell behavioral assays to suggest that ROR2 signaling regulates a variety of HFSC activities, including self-renewal, differentiation, migration and proliferation, which ultimately influence hair cycling. While the data are potentially interesting, the quality of several experiments and their interpretation need improvement/clarification, as commented below.

Major points:

1. Figure 1: The quantification of ROR2 expression and the activation of downstream effectors is crucial to the conclusions made by the authors. However, the overall quality of the Western blots is rather poor. Some of the signals appear to be near background (this is especially apparent when looking at the full scans in the supplementary materials). Optimization of the Western conditions is therefore needed to prove blot quality. It is also important that the signal intensity be quantified from multiple experimental replicates and analyzed for statistical significance.

We thank the reviewer for bringing up this concern. In the current manuscript, the quality of western blots was greatly improved, and all WB results presented in the manuscript were repeatedly shown from several independent biological replicates ($n \geq 3$). For examples, the expression levels of ROR2 in FACS-purified HFSCs were quantified from 4 independent experiments in which 4 different sets of animals were used (now in Figs. 1b, S1c).

2. Based on the data shown in Figure 1c, the authors concluded that ROR2-associated signaling is profoundly reduced in FACS-purified ROR2 CKO HFSCs. However, the Western blots actually show that while the p-JNK and p-PKCalpha signals decrease in ROR2 CKO HFSCs, the total JNK and PCKalpha protein levels are also similarly decreased; it is therefore inconclusive whether ROR2-JNK or ROR-PCK signaling are actually reduced.

We thank the reviewer for this comment. Indeed, the overall signals of total JNK and PKC proteins were significantly reduced while p-JNK and p-PKC signals decreased in FACS-purified 3dpd *Ror2* cKO HFSCs (now in Fig. 7a). The reduction in phosphorylated forms and total JNK and PKC proteins was also detected in anagen *Ror2* cKO HFSCs by whole-mount immunofluorescence staining (now in Fig. S8b). Given that HFSCs are usually at transition from the quiescence state to an activated state at anagen onset or upon depilation, reduction in total protein levels suggests the insufficient induction of protein expression. Thus, we believe that the overall signaling activities of JNK and PKC were compromised in *Ror2* cKO HFSCs *in vivo*. Besides that, we also found the activities of JNK and PKC was reduced in *Ror2*^{-/-} HFSCs either stimulated by Wnt ligands (now in

Fig. 4a) or cultured in the medium promoting proliferation (now in Fig. S8a). Taken together, we could conclude that the signaling activities mediated by JNK and PKC at least partly, if not all, depends on ROR2.

3. The authors should also clearly describe what the ROR2 CTRL animals are. The scheme shown in Figure S1a is not sufficient. What exactly were the genotypes of the parent mice used to generate the ROR2 CTRL and CKO progeny? Were the ROR2 CTRL mice littermates of the CKO animals? What were their genotypes? In order to ascertain that Cre-driver allele itself does not contribute to the observed phenotype, the CTRL mice should carry the Cre allele as well (e.g. K15CrePGR; ROR2f/+;ROSA26LSL-YFP).

We thank the reviewer for this comment. Our breeding strategy to generate *Ror2* cKO and their control littermates was *K15CrePGR+;Ror2^{+fl};Rosa26^{LSL-YFP}* (males) X *Ror2^{fl/fl};Rosa26^{LSL-YFP}* (females) to prevent any issues caused by the germline transmission. Our *Ror2* Ctrl animals for *Ror2* cKO mice were sex-matched *Ror2* heterozygous (*Cre*+) littermates, or wildtype (*Cre*-) littermates only when the littermates of *Ror2* cKO mice did not contain any *Ror2* heterozygous mouse. We are aware of the potential contribution of Cre allele. Before performing any experiments, we have carefully observed the phenotypes of mice (e.g. hair coat recovery and hair cycle entry) and analyzed the gene expression levels of HFSCs purified from *Ror2* heterozygous and wildtype mice in the same litter. As we did not find the differences between them, we decided to use sex-matched *Ror2* wildtype mice as the control if we did not have any *Ror2* heterozygous mouse in the same litter of *Ror2* cKO mice. The description about the breeding strategy and used control animals are now included in the “Methods” section of our current manuscript on page 21.

4. Figure S1: In my opinion, the data presented here are crucial to the main conclusions of the paper and should be shown in the main figures. Figure S1d is confusing. The authors used an anti-ROR2 to perform the staining. However, instead of showing that expression of the ROR2 protein is indeed abrogated in the ROR2 CKO sample, they used the “ROR2” autofluorescence signals to mark hair shafts and sebaceous glands.

We thank the reviewer for this suggestion. We have moved the datasets in Fig. S1 to the main figure (now in Fig. 1). The immunostaining for ROR2 in Fig. S1d was improved by performing immunostaining with the whole-mount skin tissues (now in Fig. S1e). As proven in Fig. S1e, the expression of ROR2 protein was indeed diminished in the bulge of *Ror2* cKO HF.

5. Figure 3: The quality of the Western blots shown in this figure is better. However, some issues concerning data interpretation need to be addressed. Based on the changes in Rho GTPase activities show in Figure 3b, the author tested whether migration is affected (Figure 3c). However, it appears that it is the basal activity of Rac1 and Cdc42 that is altered in ROR2 null cells, and Wnt5a stimulation has not further effects in this context. However, in the migration assay, the knocking out ROR2 alone does not affect migration, and the migration phenotype is only apparent in ROR2 nulls cells after Wnt5a stimulation. This discrepancy in regulation between the Rho GTPase and migration assays needed to be addressed.

We appreciate the reviewer for this comment. In Fig. 3b (now in Fig.4b), we could appreciate a little effect of Wnt5a stimulation for Rac1 and Cdc42 activation if we considered the total levels of them (for Rac1: Rac1-GTP/Rac1 = 1.1/0.8 = 1.4-fold increase; for Cdc42: Cdc42-GTP/Cdc42 = 1.2/0.7 = 1.7-fold increase). However, we agreed with the reviewer that the effect of *Ror2* depletion was mainly on the basal activities of Rac1 and Cdc42. Thus, we modified the interpretation of the result for Fig. 3b (now in Fig.4b) to “regardless of Wnt5a stimulation, activities of Rac1 and Cdc42 were prominently reduced in *Ror2*^{-/-} HFSCs” on page 11.

In the transwell cell migration assay, serum-free medium (SFM) was used in both upper and lower chambers as our control, which confirmed no migration activity of HFSCs until Wnt5a or Serum was added into lower chamber as a chemoattractant. Therefore, in the presence of Wnt5a or Serum, the cell migration of the *Ror2*^{+/+} HFSCs was increased up to 1.5 or 2 folds, respectively, but these migratory effects were abolished in *Ror2*^{-/-} HFSCs (now in Fig. 4d). This result indicates that *Ror2*^{-/-} HFSCs displayed impaired chemotaxis cell migration. In order to examine the cell movement of *Ror2*^{-/-} HFSCs, we performed scratch wound migration assay, which measures the collective cell migration ability with no need of any stimuli. Given the low activities of Rac1 and Cdc42 in *Ror2*^{-/-} HFSCs, the cell motility was indeed compromised in *Ror2*^{-/-} HFSCs (now in Figs. 4e, S4b).

6. In Figure 3d, while the effects of GF109203X could be interpreted as PKC acting downstream of ROR2, as the authors did, it could also be explained by PKC acting in an independent pathway to regulate HFSC proliferation. In fact, the observation that GF109203X treatment produces a stronger effect than vehicle-treated ROR2 null cells is more consistent with the latter possibility.

We appreciate the reviewer's comment. Indeed, in Fig 3d (now in Fig. S8c), the effect of PKC inhibitor on HFSC proliferation was slightly stronger than vehicle-treated *Ror2*^{-/-} HFSCs (16.7% v.s. 11.4%; about 30% reduction). However, we could not fully rule out the possibility of PKC acting in ROR2-independent pathway in regulating cell proliferation. Thus, we modified our statement to "ROR2 regulates HFSC proliferation, at least partly, via PKC-dependent signaling" on page 16.

However, as we found that PKC expression was downregulated in *Ror2* cKO HFSCs *in vivo* (now in Figs. 7a, 7b, S8b) and in cultured *Ror2*^{-/-} HFSCs (now in Fig. S8a), we believe that PKC downregulation at least partly contributed to the phenotype that we observed in HFSCs lacking *Ror2*. Besides that, we also found that PKC inhibition downregulated AMPK to a similarly low level in between *Ror2*^{+/+} and *Ror2*^{-/-} HFSCs (now in Fig. 7c), and that PKC inhibition recapitulated the effect of *Ror2* depletion in β -cat cKO HFSCs (now in Fig. 7d). All of these findings suggest PKC acts in the same axis as ROR2 in regulating HFSCs.

7. Figure 4: This and the next two figures test the possibility that upregulated ROR2 signaling might compensate for the loss of beta-catenin-dependent canonical Wnt signaling. The part of the story seems somewhat disconnected from the first part of the study. While the upregulation of ROR2 expression in beta-catenin KO HFSCs is convincing and potentially interesting, the experiments do not offer deeper mechanistic insights into how ROR2, a noncanonical receptor, could function to mediate WNT3A-induced canonical Wnt signaling. Their observation that ROR2 upregulation is insufficient to functionally compensate for the loss of beta-catenin also diminishes the significance of the experiment. Lastly, this entire second part of the study is conducted in the context of beta-catenin KO, which does not further corroborate how ROR2 signaling functions to control HFSC biology under normal physiological context, as shown in the first part of the study.

We appreciate the reviewer for this comment. We understand the reviewer's concerns, so that we further investigated how ROR2 functions to regulate HFSC activation, self-renewal/proliferation and maintenance. In the current manuscript, we showed that in HFSCs ROR2 not only transduced Wnt5a-dependent non-canonical Wnt signaling (now in Fig. 4), but it is also required for Wnt3a-induced transcriptional activation of canonical Wnt target genes, which is likely responsible for HFSC activation (now in Figs. 2c, 2d, 4c). As explained in the question #6 asked by the reviewer #1, the underlying mechanism on how ROR2 activates canonical Wnt signaling is not yet clear. However, according to the previous studies, ROR2 could cooperate with Fzd2 to mediate Wnt3a-stimulated canonical Wnt signaling (Li *et al.*, *BMC Mol Biol*, 2008), or modulate β -catenin-dependent transcriptional regulation via activation of JNK downstream protein, c-Jun (Nateri *et al.*, *Nature*, 2005).

Furthermore, to corroborate how ROR2 regulates HFSC self-renewal/proliferation and maintenance, we analyzed *Ror2*-depleted HFSCs in culture and identified a novel role of ROR2 in the regulation of ATM/ATR-dependent DNA damage response, which is crucial for the long-term maintenance of HFSCs. We found that *Ror2*^{-/-} HFSCs were unable to properly repair DNA damage and control ROS production, which in turn led to slow proliferation and cell cycle arrest (now in Fig. 5). Given that a population of *Ror2* cKO HFSCs got lost during HF regeneration (now in Figs. 2g, 2h), we speculate that this ROR2-dependent regulation might be utilized by HFSCs residing in their niche; however, it requires further investigation to confirm this speculation. We also discussed this possibility in the “Discussion” section of our current manuscript on page 19.

Due to the reason described above (Please see comments on reviewer #1, questions #3 & #7), Fig. 4 & 5 about long-term cultured β -cat cKO HFSCs were removed in our current manuscript. Instead, we turned to study β -cat cKO HFSCs *in vivo* by performing analyses with FACS-purified β -cat cKO HFSCs at different stages of hair cycle (now in Fig. 6b). As the expression levels of *Ror2* mRNA and protein were significantly elevated at late telogen and aged β -cat cKO HFSCs (now in Figs. 6b, S6), we believe that ROR2 plays a critical role in the maintenance of β -cat cKO HFSCs, which strongly correlates with its role in the normal physiological context. Indeed, depletion of *Ror2* caused β -cat cKO HFSCs leaving quiescence and undergoing wrong fate differentiation (now in Figs. 6c, 6d), which confirms the role of ROR2 in regulation of HFSC long-term maintenance. Thus, *in vivo* model of β -cat cKO HFSCs in the current manuscript (now in Figs. 6, 7d-e, S6) is more appropriate to explore the functional role of ROR2.

8. Figure 6: The original data used to construct the scheme shown in Figure 6a should be presented. In Figure 6d and 6e, data for beta-catenin single CKO mutant should be included as an additional control for comparison. In their previous study (Ref 45), the authors observed that conditionally knocking out beta-catenin alone was sufficient to induce inappropriate sebocyte differentiation. Why do they not observe this effect in the present study (Figure 6a)?

We thank the reviewer for the suggestions. In the current manuscript, we added in the original data generated from β -cat single cKO mice (now in Fig. 6a). In agreement with the scheme shown in Fig. 6a, β -cat cKO HFs displayed a delay in anagen entry at the 1st hair cycle and then stayed at the 2nd telogen throughout the rest of the life as evidenced by the delay and inability of hair coat recovery, respectively (now in bottom left of Fig. 6a,). Histological analysis of whole-mount β -cat cKO skin tissues from aged animals demonstrates that the arrested β -cat cKO HFs maintained intact bulge compartments (Now in bottom right of Fig. 6a, S6b). This phenotype was also found in the previous study (Lien *et al.*, *Nat Cell Biol*, 2014) where β -catenin was depleted in HFSCs at the 2nd telogen, and β -catenin-depleted HFs also remained in telogen throughout the rest of the life (Fig. 1 in the reference) unless HFSCs were activated by depilation. Only depilation-activated β -cat cKO HFSCs underwent sebocyte differentiation (Fig. 4 in the reference). Thus, knocking out β -catenin alone in HFSCs is not sufficient to induce wrong fate differentiation. Together, our findings agree with the previously published study and further provide a mechanism on how β -catenin-depleted HFSCs could be retained in the bulge without loss of stem cell pool.

Minor points:

1. Several typos within text (e.g., “confirm our previous discoverY” rather than discoveries, “native niche by generating inducible... mice” rather than “a inducible... mice”, etc.)

We thank the reviewer for the corrections. The texts of our current manuscript were carefully edited and the indicated typos were corrected accordingly.

2. Figures:

o 2b, error bars for ROR2 control appear to be exactly the same for each passage number; please double check.

The graph that contains the same error bars in some passage numbers was correct. The graph was made based on 3 independent groups (6 colonies per group) for each genotype. The viabilities of colonies were judged based on the growth of YFP+ HFSCs after each passaging (now in Fig. 3b). As the growth of colonies was maintained within the group and each group remained in the same numbers of viable colonies over the passaging, the error bars would be exactly the same for these passages.

o 4a, the pJNK bands were partially cut off, which makes the quantification less reliable.

o 4d, 5a, 5c, shades of green are too similar to easily distinguish one condition from the other.

o 4e, Y-axis legend is confusing and could be changed to “Fold of induction of Wnt3a genes” or something similar to make interpretation of results easier to understand.

These figures were removed in our current manuscript due to the reason mentioned above (Please see comments on reviewer #1, questions #3 & #7). Regarding Wnt3a-induced gene expression (now in Fig. 4c), we changed the Y-axis legend to “Fold of Wnt3a-induced gene expression”.

3. Other: At least two publications are listed twice within the references (specifically 37/52 and 51/58).

We thank the reviewer for identifying the redundant citation. We have corrected it and carefully gone through the list of our references to avoid this type of mistakes.

Reviewer #3:

Extensive studies have been performed in exploring the Wnt/b-catenin signaling pathway in activation of hair follicle stem cells (HFSCs) during hair regeneration and tissue homeostasis, but little is known about the function of β -catenin-independent Wnt signaling in HFSCs. In the manuscript titled “Ror2-dependent Signaling Regulates Self-Renewal and Maintains Stem Cell Identity in Hair Follicle”, Veltri et al. investigated the role of non-canonical signaling pathway receptor ROR2 in regulating the activation of HFSCs during hair regeneration and the maintenance of HFSCs identity using in vivo and ex vivo experiments. First, using a ROR2 conditional knockout mice model, they show that ROR2 is required for HFSCs activation during hair regeneration. The expression of HFSC signature genes is decreased in the bulge cells of ROR2 knockout mice, as well as in the cultured ROR2-depleted HFSCs. Without ROR2, the cultured HFSCs cannot self-renew and maintain their identity during long time culture. In addition, they generated the ROR2-deficient HFSCs, through which they show that ROR2 transduces Wnt5a-dependent signaling and regulates HFSC migration and proliferation. Since previous study shows that b-catenin-deficient HFSCs can maintain their stemness, authors found that ROR2 is highly expressed in b-catenin-deficient HFSCs and is required for b-catenin-deficient HFSCs to maintain their identity through PKC signaling, which has been shown to maintain HFSC identity (Niessen et al., 2013, J Cell Biol). ROR2 removal or PKC inhibition in b-catenin-deficient HFSCs not only leads to the reduced stemness of HFSCs but also turns HFSCs into a sebaceous gland fate. In general, the approaches used in this study can be updated but the results described in this study are novel, and they provide new insight into the mechanisms by which non-canonical signaling pathway in regulating stem cell self-renewal and identity maintenance.

We appreciate the reviewer recognizing the novel findings of our study. We have updated our manuscript and added in new datasets about the mechanism of ROR2 in regulation of HFSC maintenance (now in Fig. 5). The answers for the questions raised by the reviewer are described below.

Major points:

1. HFSC activation and identity maintenance seem to be in the opposite directions for stem cell fate. Many previous studies show that loss of stemness-related genes in HFSCs leads to activation of HFSCs. This study shows that ROR2 is both required for HFSCs activation and stemness maintenance. Loss of ROR2 in HFSCs leads to decreased stemness-related genes but delayed anagen entry of the hair follicles. Thus,

(1) it is unclear how ROR2 balances these two functions in HFSCs in the present study.

We thank the reviewer for bringing up this question which helped us to dissect carefully the role of ROR2 in HFSC activation and maintenance. In fact, ROR2 not only transduces non-canonical Wnt signaling, but is also required for canonical Wnt-induced transcriptional activation. In the current manuscript, we showed that the expression of genes that respond to β -catenin-dependent Wnt signaling were significantly downregulated in depilation-activated *Ror2* cKO HFSCs (now in Fig. 2c) or *Ror2* cKO HFSCs at anagen onset (now in Fig. 2d), suggesting the activation of Wnt/ β -catenin signaling that facilitates HFSC activation is dependent on ROR2. Moreover, cultured *Ror2*^{-/-} HFSCs displayed significant reduction in Wnt3a-induced expression of canonical Wnt target genes (now in Fig. 4c), indicating that ROR2 is required to transduce canonical Wnt signaling. Thus, as a Wnt receptor ROR2 is essential to transduce canonical Wnt signaling which is necessary for HFSC activation and anagen entry.

In addition to HFSC activation, in the current manuscript we demonstrated that ROR2 plays an essential role in regulation of ATM/ATR-dependent DNA damage response (now in Fig. 5), which is indispensable for long-term maintenance of HFSCs. During the normal homeostasis or culture condition, HFSCs undergo self-renewal and tissue regeneration which make them vulnerable to DNA damage caused by external and/or internal factors. When *Ror2* was depleted in HFSCs, ATM/ATR-dependent DDR was compromised in *Ror2*-deficient HFSCs, which led to accumulation of DNA damage and excess ROS production. This deficiency eventually resulted in slow growth of HFSCs and inability in HFSC long-term maintenance (now in Fig. 5). The HFSC maintenance defect was reflected by the low expression of HFSC stemness genes (now in Figs. 2e, 2f, 4f).

Therefore, ROR2 could play the dual role in regulation of HFSC activation and maintenance due to its independent functions in controlling activation of canonical Wnt signaling and ATM/ATR-dependent DDR, respectively.

(2) Moreover, if ROR2 functions to maintain the identity of HFSCs, ROR2 should also have a higher expression in telogen, whereas it is not the case in the present study?

The expression levels of ROR2 should be regulated by the transcription factors and co-factors that bind to the promoter of ROR2 or by the post-translational modification machinery that regulates protein stability. We do not know what is the upstream mechanism regulates ROR2 expression, which cause the differential expression of ROR2 in telogen and anagen HFSCs, but we did find that depletion of *Ror2* caused the reduction in expression of HFSC stemness genes in both activated and late telogen HFSCs (now in Fig. 2e, 2f), and FACS-purified HFSCs could not be maintained for a long time in culture (now in Fig. 3). As described above in (1), the downregulation of HFSC stemness genes could be the consequence of the HFSC maintenance defect, which

potentially could be caused by deficient DDR in HFSCs depleted for *Ror2*. Even if ROR2 expression in telogen HFSCs is lower (but not absent, please refer to response to reviewer#1, question #2), its role in guarding HFSCs from the loss remains functional significant.

2. Most of the studies are based on cultured HFSCs, which do not have the stem cell niche environment, therefore, is different from the *in vivo* case. This is one of the major concerns of this study.

We agree with the reviewer. Thus, in the current manuscript, we have used the *Ror2* cKO animal model to analyze the effect of *Ror2* loss in HFSC activation and maintenance (now in Figs. 1 & 2), and the *ex vivo* model (now in Fig. 3) to examine FACS-purified *Ror2* cKO HFSCs in cell growth and maintenance in culture condition. In order to dissect the mechanism of ROR2 in regulation of HFSC activation and maintenance, we had to generate *Ror2*^{-/-} HFSCs and then analyzed these cells in various aspects (now in Figs. 4 & 5). Regarding the role of ROR2 in β -cat cKO HFSCs, instead of analyzing cultured β -cat cKO HFSCs (in the previous manuscript), we focused on *in vivo* β -cat cKO HFSCs that reside in their native niche and performed analyses with FACS-purified β -cat cKO HFSCs at different stages of the hair cycle (now in Fig. 6), as well as with β -cat cKO animals depleted for *Ror2* or treated with PKC inhibitor (now in Figs. 6 & 7). Apart from these, in order to demonstrate the physiological relevance of the key proteins, ROR2 and PKC, in HFSC maintenance, we have performed several whole-mount immunostainings with skin tissues from animal models. Altogether, our current manuscript has been greatly improved in this regard.

3. Authors show decreased proliferation and decreased expression of HFSCs signature genes in bulge stem cells of the K15Cre-ROR2 mice, whereas the hair follicles just show delayed rather than abolished regeneration, indicating other compensative mechanisms for hair regeneration. This is not clear in this study.

We thank the reviewer for bringing up this question, which helped us to provide better interpretation of our data. As described above in the response to question #1, delayed HFSC activation/hair regeneration of *Ror2* cKO HFSCs was caused by downregulation of canonical Wnt target genes at anagen onset (now in Figs. 2c, 2d). Given that ROR2 is one of Wnt receptors, but not the only, when *Ror2* is depleted canonical Wnt signaling could still be transduced by other Wnt receptors, e.g. Lrp proteins. Thus, depletion of *Ror2* only resulted in delayed, but not abolished, hair regeneration due to the compensation of other Wnt receptors. The explanation for this phenotype is now clearly described in our current manuscript on page 8.

4. What is the phenotype of the mice with long term deletion of ROR2 in HFSCs? This should be characterized.

We appreciate the reviewer for this important question. Based on our observation, *Ror2* cKO mice continuously displayed a delay in anagen entry up to 3-4 rounds of hair cycle (note that hair cycle entry is not synchronized in aged animals). To analyze HFSCs *in vivo*, we conducted flow cytometry analysis to measure the recovery of YFP+ HFSCs after one round of HF regeneration. As shown, the pool of YFP+ *Ror2* cKO HFSCs that recovered from the 1st depilation-induced HF regeneration (now in Fig. 2g) or the 1st round of hair cycling (now in Fig. 2h) was reduced as compared to control HFSCs. These results indicate that ROR2 is required for the maintenance of a HFSC population. However, we noticed that if any *Ror2* cKO HFSCs could survive through the 1st depilation, they were able to sustain the subsequent HF regeneration upon repetitive depilation (Fig. S3a). This result suggests that loss of ROR2 could be compensated by other factors in those *Ror2* cKO HFSCs that survived through the process of regeneration. Due to this potential compensatory mechanism, we could also find the presence of YFP+ *Ror2* cKO HFSCs in some aged HFSCs that show normal HF morphology (Fig. S3b). Thus, we could only conclude that *Ror2* depletion has a

primary effect on HF regeneration, particularly in self-renewal/proliferation; in the severe case, *Ror2* cKO HFSCs die and get lost in the pool; in the mild case, they bypass the challenge and get recovered from the process, which allows them maintained for long term.

The phenotype of mice with long-term deletion of *Ror2* is relatively mild as compared to cultured *Ror2* cKO HFSCs that could not be maintained long-term in the medium promoting proliferation (now in Fig. 3). The discrepancy in the phenotype severity of *Ror2* cKO HFSCs *in vivo* and in culture is likely due to the differences in the frequency of HFSC proliferation. Compared to cultured HFSCs which are constantly in proliferation, HFSCs residing in their native niche are maintained in a quiescent state until the initiation of hair cycle. Even at anagen onset, only a subset of HFSCs undergo proliferation for self-renewal and regeneration. Constant proliferation in cultured HFSCs means frequent DNA replication, which concomitantly increases the frequency of DNA damage repair. As we found that *Ror2*-depleted HFSCs show defects in ATM/ATR-dependent DDR, constant proliferation without *Ror2* could lead HFSCs to rapid accumulation of DNA damage and subsequent cell cycle arrest or death. This is one of reasons why long-term cultured *Ror2* cKO HFSCs show stronger phenotype than those *in vivo*. We had included this comprehensive explanation in the “Discussion” section of our current manuscript on page 19-20.

5. Author claimed that *Ror2* is diminished in the YFP+ bulge and hair germ (HG) of *Ror2* cKO HFs (Fig. S1d), which is unconvincing. In addition, please show ROR2 and PKC expression during telogen and anagen initiation (multiple stages) using immunofluorescent staining.

We appreciate the reviewer’s comment. In the current manuscript, we showed the results of immunofluorescent staining with the whole-mount skin tissues, which clearly display ROR2 staining in the cell membrane of bulge HFSCs (now in Figs. 1a, S1a), and these staining were diminished in *Ror2* cKO HFs (now in Fig. S1e).

As show in Figs. 1a and S1b, ROR2 was expressed at a higher level in the bulge of HFs at anagen onset as compared to the bulge of telogen HFs. The ROR2 immunostaining results agree with the results of immunoblotting with FACS-purified HFSCs from telogen and anagen HFs (now in Fig. 1b, S1c). As suggested by the reviewer, we also performed whole-mount immunostaining for PKC and found that the bulge of HFs at anagen onset displayed a higher expression level of PKC than those at telogen HFs (see fig. 6 at right, quantification is shown).

Regarding the expression levels of ROR2 and PKC in *Ror2* Ctrl and cKO HFs, the results of whole-mount immunostaining were shown in Fig. S1e and Figs. 7b, S8b, respectively. As demonstrated, ROR2 and PKC expression levels were significantly reduced in the bulge of *Ror2* cKO HFs at anagen onset (now in Figs. S1e, S8b) as well as in telogen (now in Fig. 7b). In addition to immunostaining, we also performed western blotting analyses, and showed that ROR2 protein was diminished in FACS-purified *Ror2* cKO HFSCs (now in Fig. S1d) and that both PKC activity and protein level went down in depilation-activated *Ror2* cKO HFSCs (now in Fig. 7a). Thus, we conclude that ROR2 and PKC expression are upregulated in anagen HFSCs as compared to telogen HFSCs, and their expression are significantly diminished or reduced upon *Ror2* depletion.

6. Fig. 1f, you hypothesize that some of the activated *Ror2* cKO HFSCs lost HFSC markers, e.g. CD34, used in the FACS analysis, and conclude that *Ror2* can modulate HFSC identity and

maintenance by regulating the expression of HFSC markers. Please make sure if this is true. First, it looks that *Ror2* cKO has a smaller bulge from Fig. 1d. Second, please do more morphological and molecular characterization in early anagen bulge of *Ror2* cKO through staining.

We thank the reviewer for bringing up this concern. Indeed, we could not conclude that “ROR2 modulates HFSC identity and maintenance by regulating the expression of HFSC markers” given that not only *Ror2* cKO HFSCs (now in Figs. 2e, 2f) but also late telogen β -cat cKO HFSCs (now in Fig. 6b) display lower expression of HFSC signature genes. However, the phenotypes of these mutant animals are different. *Ror2* cKO animals displayed delayed HFSC activation and deficiency in long-term maintenance of cultured HFSCs (now in Figs. 1-5), whereas β -cat cKO animal showed arrested hair cycle at telogen phase throughout the rest of life (now in Fig. 6a).

In the current manuscript, we performed several immunostainings with whole-mount skin tissues for ROR2, PKC and JNK to characterize *Ror2* Ctrl and cKO HFSCs. As shown, all of these staining were significantly compromised in *Ror2* cKO HFSCs (now in Fig. S1e, 7b, S8b), but we did not find any differences in bulge sizes between *Ror2* Ctrl and cKO HFSCs. The images of Fig.1d (now in Fig. 2a) were from depilated HFSCs, which HF structure is altered due to the removal of hair shafts. However, by conducting flow cytometry analysis to measure the recovery of YFP+ HFSCs after one round of HF regeneration, we found that the recovery of YFP+ *Ror2* cKO HFSCs was reduced as compared to control HFSCs (now in Figs. 2g, 2h), indicating that a population of *Ror2* cKO HFSCs was lost likely due to their deficiency in self-renewal/proliferation.

Moreover, we also showed that FACS-purified *Ror2* cKO HFSCs could not be long-term maintained in culture (now in Fig. 3) and that cultured *Ror2*^{-/-} HFSCs displayed impaired ATM/ATR-dependent DNA damage response, which led to HFSC slow growth and inability in long-term maintenance (now in Fig. 5). Thus, the downregulation of HFSC stemness genes in *Ror2* cKO HFSCs could be the outcome of the maintenance defect. Therefore, according to the datasets from our *in vivo* and *in vitro* models shown in our current manuscript, we revised the manuscript accordingly and made a general conclusion as “ROR2 plays a critical role in HFSC self-renewal and maintenance of a HFSC population”.

7. Authors largely used the b-catenin-deficient mouse model, whereas it is unclear why *Ror2* is upregulated in b-catenin knockout mice.

We appreciate the reviewer for this comment. In the current manuscript, we discovered that the upregulation of ROR2 was associated with downregulation of HFSC stemness genes in the late telogen and aged β -cat cKO HFSCs (now in Figs. 6b, S6). Thus, we speculated that upregulated ROR2 is required to maintain β -cat cKO HFSCs in quiescence, which could prevent them from activation and subsequent wrong fate differentiation. To demonstrate this possibility, we depleted *Ror2* in β -cat cKO HFSCs by generating *Ror2* and β -cat double mutant mice. Our results showed that depletion of *Ror2* in β -cat cKO HFSCs caused HFSCs leaving quiescence and undergoing sebocyte differentiation (now in Figs. 6c, 6d), which supports our speculation.

8. Using a similar Lgr5-CreERT2;aPKC^{fl/fl};Rosa26RLacZ transgenic mouse model, previous study showed that loss of aPKC in HFSCs results in increased proliferation and loss of quiescence in the bulge, turning HFSCs into a sebaceous gland and epidermis cell fate (Niessen et al., 2013, J Cell Biol). In the present study, authors also show that PKC functions downstream of ROR2 to regulate HFSCs activity. Please explain the inconsistency between this study and the previous study.

We thank the reviewer for pointing out the differences between two studies, which helped us to clarify our data. First of all, the PKC activity detected in our study was based on the phosphorylation status of PKC proteins at the carboxy-terminal hydrophobic site Ser660, which phosphorylation is

only present in classical or novel PKC isoforms, not in atypical PKC (aPKC) proteins (see more antibody information in the website of Cell Signaling Technology, #9371). To avoid the confusion, in the current manuscript we clearly stated that the analyzed PKC activity is specific to classical and novel PKC proteins described on page 10.

Second, in our current manuscript, PKC inhibition was performed by the treatment of a PKC inhibitor, GF109203X, which is a potent and selective inhibitor to classical PKC proteins and to novel PKC proteins when used in a higher concentration, but has low or no effect to aPKC proteins (https://www.rndsystems.com/products/gf-109203x_0741). Thus, the effects of PKC inhibition on cultured HFSCs and the mouse model in our study (now in Figs. 7d-e, S8c) are attributed to the suppression of classical and novel PKC isoforms; this statement is clarified on page 16 of our current manuscript.

Third, as described in question #1, the delayed activation of *Ror2* cKO HFSCs was due to downregulation of β -catenin-dependent Wnt signaling (now in Figs. 2c, 2d) due to the role of ROR2 in transduction of both canonical and non-canonical Wnt signaling. Whether downregulation of PKC in *Ror2* cKO HFSCs contributes to this phenotype was not examined.

Due to the differences on targeted PKC isoforms between our study and the aPKC KO paper (Niessen *et al.*, *J Cell Biol*, 2013), we could expect the differences in mouse phenotypes and the effects on HFSCs. Nevertheless, both studies highlight the importance of PKC proteins for HFSC proliferation and maintenance.

9. Moreover, please explain why HFSCs only differentiate into sebaceous gland in *Ror2* and β -cat double cKO HFSCs, rather than in ROR2-deficient mice, in which PKC is downregulated.

We appreciate this reviewer bringing up this unclear interpretation. As explained in the question #8 from reviewer#1, previous studies have demonstrated that HF fate was determined by β -catenin-dependent signaling as *β -catenin* depletion in the developing epidermis led to HF degeneration and sebocyte formation (Gat *et al.*, *Cell*, 1998; Huelsken *et al.*, *Cell*, 2001; Lo Celso *et al.*, *Development*, 2004; Lien *et al.*, *Nat Cell Biol*, 2014). However, when *β -catenin* was depleted specifically in HFSCs, *β -cat* cKO HFSCs remained quiescent throughout the rest of the life without undergoing differentiation unless HFSCs were activated by depilation. Depilation-activated *β -cat* cKO HFSCs were able to proliferate, but differentiated into sebocyte fate instead of HF fate (Lien *et al.*, *Nat Cell Biol*, 2014). This finding suggests the additional factors that lower down the requirement for *β -cat* cKO HFSCs to get activated is the cause of *β -cat* cKO HFSC differentiation, which preferentially turns into sebocyte fate.

In the current manuscript, using qPCR analyses we discovered that *Ror2* mRNA was upregulated in the late telogen and aged *β -cat* cKO HFSCs showing downregulation of HFSC stemness genes (now in Fig. 6b). By immunostaining we also showed the elevated expression of ROR2 protein in the bulge of *β -cat* cKO HF at late telogen or aged animals (now in Fig. S6). Thus, we speculated that upregulated ROR2 is required for the long-term maintenance of *β -cat* cKO HFSCs where HFSC stemness gene expression were lower. In fact, by depleting *Ror2* in *β -cat* cKO HFSCs, we found that *Ror2* depletion challenges the quiescent state of *β -cat* cKO HFSCs by lowering down the threshold of *β -cat* cKO HFSCs for activation, and activated *β -cat* cKO HFSCs then undergo differentiation into sebocyte fate. Thus, the cause of the wrong fate determination of *Ror2*/ *β -cat* dKO HFSCs was the synergistic effects of *β -catenin* loss and *Ror2* depletion.

Furthermore, given that PKC activity and expression depend on the presence of ROR2 (now in Figs. 7b, S8b) and PKC expression was elevated in *β -cat* cKO HFSCs (now in Fig. S9), we examined if PKC inhibition could mimic *Ror2* depletion. Indeed, PKC inhibition could also cause β -

cat cKO HFSCs undergoing differentiation (now in Fig. 7d), indicating PKC acts in the same axis as ROR2 in regulating HFSC maintenance. Together, our results suggest that depletion of *Ror2* or PKC inhibition lowered down the requirement of β -cat cKO HFSCs for activation, which in turn causes activated β -cat cKO HFSCs undergoing differentiation into sebocyte fate.

10. Fig. 6e, authors show that additional ablation of *Ror2* in β -cat cKO HFSCs (*Ror2*/ β -cat dKO) resulted in downregulation of HFSC signature genes, which might have tipped the balance of arrested β -cat cKO HFSCs towards differentiation. Whereas authors show that both *Ror2* and β -catenin knockout mice display downregulation of HFSC signature genes without differentiation into SG. So other key genes should be identified.

We appreciate this reviewer for bringing up this question. As explained in question #9, the wrong fate determination of *Ror2*/ β -cat dKO HFSCs was caused by the synergistic effects of β -catenin loss and *Ror2* depletion. In other words, *Ror2* depletion challenges the quiescent state of β -cat cKO HFSCs by lowering down the threshold of β -cat cKO HFSCs for activation, and activated β -cat cKO HFSCs then undergo differentiation into sebocyte fate. This effect is similar to the phenomenon of depilation on β -cat cKO HFSCs (Lien *et al.*, *Nat Cell Biol*, 2014).

Indeed, both *Ror2* and β -cat single cKO HFSCs displayed a certain level of HFSC signature gene downregulation (now in Figs. 2e, 2f, 6b). However, the downregulation levels of these genes were even more significant and consistent among animals in *Ror2*/ β -cat dKO HFSCs (now in Fig. 6e). Given that we do not have strong evidence supporting “lowering down HFSC signature gene by *Ror2* depletion is the cause inducing β -cat cKO HFSC differentiation”, we removed any statements which might cause misleading, and we revised our manuscript according to the datasets shown in Figs. 6 & 7, and concluded that “we identify a compensatory role of ROR2-PKC signaling in protecting β -catenin-null HFSCs from the loss of stem cell pool”.

Regarding how *Ror2* depletion caused β -cat cKO HFSCs undergoing differentiation, we speculate as follows. In the current manuscript, we showed that depletion of *Ror2* in cultured HFSCs resulted in accumulation of unrepaired DNA damage and impairment in ATM/ATR-dependent DDR (now in Fig. 5). Along the same line, we can speculate that depletion of *Ror2* in β -cat cKO HFSCs might cause the similar defects, which in turn trigger a wound response of β -cat cKO HFSCs and the subsequent HFSC activation. Although we are not very clearly yet if *Ror2* depletion in HFSCs *in vivo* acts similarly as those occurring in cultured *Ror2*^{-/-} HFSCs, we believe that loss of *Ror2* in β -cat cKO HFSCs would have disrupted the balance of quiescent β -cat cKO HFSCs, which led to β -cat cKO HFSC activation for repair.

Minor points:

1. Fig 1a-c is unconvincing, as b-actin is higher in aHFSC group compared to the qHFSC group. Also in many cases, the expression of b-Actin and GAPDH is different between different groups. In addition, many bands are barely legible throughout the study. For example, the blot for Figs. 1a & b etc. should be replaced.

We agree with the reviewer about the poor quality of the western blots in the previous version of our manuscript. In the current manuscript, the quality of western blots was greatly improved, and all WB results presented were repeatedly shown from several independent biological replicates (n \geq 3). The uncropped images of blots were provided in Fig. S10. The expression levels of ROR2 in FACS-purified qHFSCs and aHFSCs were quantified from 4 independent experiments in which 4 different sets of animals were used (now in Fig. 1b and S1c). As for the blots of HFSCs from the post-depilated HF (Fig. 1b in the previous version), it has been challenging to find an appropriate

loading control for this experiment. The results of this experiment were inconclusive, so that these blots were removed from the current manuscript.

2. Results between Fig. S1b and Fig. S1e should be consistent. How do you define anagen entry, through morphology, or through HFSCs (e.g., second hair germ cells) proliferation during hair regeneration?

We thank the reviewer for this comment. Indeed, there was disagreement between Fig. S1b and S1e of the previous manuscript due to the differences in the hair cycle between male and female animals. In the current manuscript, we labeled each diagram showing the postnatal days of every hair cycle phases according to the sex of the animals (now in Fig. 1c, 6a, 6c, 7d, S2a). The anagen entry was defined based on the morphology of HFs (e.g. the expansion of the second HG) as well as the proliferative state of HG cells (by performing EdU labeling experiments).

3. Page 9 and Fig. S3a, fluorescent images should be shown to display the Cre activity which demonstrates the ROR2 knockdown efficiency in cultured HFSCs.

We appreciate the reviewer for this suggestion. As described, the viral infection efficiency and Cre activity were reported by the expression of RFP and YFP, respectively. To demonstrate these efficiencies, the fluorescence images of infected HFSCs before FACS purification are now shown in Fig. S4a.

4. Fig. 3f, *Krt1* and *Krt10* are epidermal differentiation markers. Please make sure if they are expressed in HFSCs, or this is due to impurity of the sorted HFSCs. In addition, lower expression of these markers should mean lower differentiation state of the HFSCs in ROR2-deficient cells, thus better maintenance of HFSCs identity? But this is not the case in this study.

We agree with the reviewer that the expression levels of *Krt1* and *Krt10* are indeed very low in HFSCs, which are not the appropriate epidermal markers for HFSCs. Thus, we removed them from the study and focused on epidermal/HF fate markers, *Krt79* and *Krt17*, which are highly expressed in HFSCs (now in Fig. 4f).

5. Page 11, authors state that the upregulation of *Ror2* was also observed in arrested β -cat cKO HFSCs residing in their native niche during prolonged telogen. Please show an image using immunofluorescent staining.

In the current manuscript, using qPCR analyses we discovered that *Ror2* mRNA was upregulated in the late telogen and aged β -cat cKO HFSCs showing downregulation of HFSC stemness genes (now in Fig. 6b). By performing immunostaining of whole-mount skin tissues, we also showed that late telogen and aged β -cat cKO HFs expressed higher levels of ROR2 in the bulge (now in Fig. S6). Depletion of *Ror2* caused β -cat cKO HFSCs leaving quiescence and undergoing sebocyte differentiation (now in Figs. 6c, 6d), indicating that upregulated ROR2 keeps β -cat cKO HFSCs in quiescence, which protects HFSCs against activation and subsequent differentiation.

6. Page 14, 1 week after application of PKC inhibitor, β -cat cKO HFSCs had already undergone sebocyte differentiation. Please show the data using staining.

We thank the reviewer for this question. We did show the phenotype of PKC inhibitor-treated β -cat cKO HFs by performing the Oil Red O staining on whole-mount skin 1 week (now in Fig. 7d, bottom left) and 3 weeks (now in Fig. S9c) post-treatment, and we described the details in the figure legends. As shown, β -cat cKO HFs treated with PKC inhibitor (+PKCi) had undergone sebocyte

differentiation. In some rare cases, we could also find PKCi-treated β -cat cKO HFs showing thickened upper follicle along with enlarged SG (now in Fig. S9c).

7. Authors state that ROR2 knockout mice have a lower ability to migrate. In this case, how do the HFSCs in *Ror2*/ β -cat dKO change the cell fate and migrate into SG to differentiate?

We thank the reviewer for bringing up this interesting question. We indeed found that cultured *Ror2* cKO HFSCs displayed deficiencies in Wnt5a-induced cell migration and collective cell migration (now in Figs. 4d, 4e, S4b), suggesting that it would take *Ror2* cKO HFSCs longer time to migrate, but their motility is not completely abolished in the presence of other migration stimuli (+ Serum in Fig. 4d). As HFSCs *in vivo* are exposed to various mitogens in their niche, we would expect a certain level of migration ability of *Ror2*/ β -cat dKO HFSCs. Migrating toward to the SG from the bulge is a shorter path than migrating downward to generate an anagen HF. Thus, it would not be impossible for *Ror2*/ β -cat dKO HFSCs to migrate into SG to differentiate.

REVIEWER COMMENTS

Reviewer #1 (Remarks to the Author):

The authors responded comments with additional data. Figs are beautiful. I do not have additional comments.

Reviewer #3 (Remarks to the Author):

The authors have fully addressed my questions. Thus, I recommend publication of this manuscript in Nature Communications.

Reviewer #4 (Remarks to the Author):

The study of Lien et al focused on the novel role of ROR2 in hair follicle Stem Cell Self-Renewal and Maintenance. The authors performed quite a diverse arrays of experiments and also put significant effort in improving the ms following first round reviewers comments.

Although, the authors performed many experiments using in vivo models to delineate the role of ROR2 in the biology of HF's stem cell niche, some caveats need to be addressed. The molecular mechanisms of ROR2 signaling in the context of stemness gene expression, self-renewal, proliferation and migration of HFSCs should be better delineated and placed into a proper context within the Wnt pathway. ROR2 is known to belong to the non-canonical Wnt signaling, therefore its functional role in the canonical, β -catenin dependent Wnt signaling should be more carefully described.

Few essential points:

1. The biological role of Wnt ligand binding and activation of ROR2 signaling. What is the level of Wnt5a and Wnt3a ligands, which are described by the authors as putative Wnt-binding ligands of ROR2 during HF's development. Does the loss of ROR2 correlated with loss of Wnt ligands (Wnt5a and/or Wnt3a). This would help to clarify whether Wnt ligand binding is actually required for the biological effect of ROR2 or whether ROR2 alone is sufficient to sustain the HF stemness niche. It would also help to delineate the fate of canonical vs non-canonical Wnt pathway during HF's development, as Wnt5a is conventionally a non-canonical Wnt ligand, whereas Wnt3a is a canonical Wnt ligand. As Wnts are secreted ligands that act in autocrine or paracrine fashion, it would be informative to know whether HFSCs are Wnt-positive or other cells, like the neighboring cells in the bulge. It would be helpful for instance, to check whether

primary HFSCs cultured with fibroblast feeder cells exhibited enhanced proliferation and colony formation in the presence of Wnt ligand compared to Wnt-free media.

2. The status of ROR1, the other ROR receptor, should be better addressed. It is known that ROR1/2 homo- and hetero-dimerize and in many instances, loss of one receptor leads to upregulation of the other one that compensates for the signaling. In WB, authors used a ROR1 antibody from Abcam that is discontinued and had poor overall reviews. Very good ROR1 antibodies are currently available from CSTechnologies, please use commercially up to date reagents. Of note, the overall quality of WB is quite poor, and the signal looks almost as if it is a background band. In Figure 1d looking at the ROR1 mRNA expression, from 5 data points, 2 are very high, pointing to an upregulation and 3 are quite low, giving an average that was interpreted as no changes in ROR1 levels. This is not accurate. Also, in Figure 4f, authors could not get statistical significance for ROR1 levels either. The overall level of ROR1 should be investigated more carefully, in WB and IF, using up to date functional reagents.

3. Whenever possible, could the authors use a ROR2 antibody that is commercially available, for instance from CST, to validate some of the results from their WB. That is because the WB signal looks rather weak, almost equal to background in many of their WB. This would substantiate their findings. For instance, in their Figure S1c and S1d, there seems to be a strong band at 80kDa, which would correspond to non-glycosylated ROR2 isoform.

4. The role of ROR2 in b-catenin-dependent activation should be carefully discussed and delineated. As authors pointed out, studies showing that ROR2 “might” activate b-catenin signaling via binding to Wnt3a are not very solid, as these experiments were done in cancer cells using over-expression systems (ref. 29 and 33 in ms). For instance, ref. 33, this is a very old study, yet pointed out that ROR2 inhibited Wnt3a function in b-catenin reporter activity levels. It is already known that Wnt3a could bind to and transduce signals via ROR1/2 during development and act in a b-catenin independent manner. For instance, Wnt3a, which is a representative of canonical Wnt, can induce neural differentiation in a b-catenin-independent manner by activating JNK and ATF-2 possibly through a receptor complex, comprising of Ror1, Fzd4/5, and LRP6 (Bengoa-Vergniory et al., 2016). Authors should not base their conclusion entirely on ROR2 expression when talking about Wnt3a/b-catenin signaling, since they have not investigated the status of other Wnt- receptors (Fzd, LRP5/6, etc) that could be equally relevant in this context.

5. The conc. of Wnt3a and Wnt5a used for stimulations in Figure 4 are quite different, 100ng/ml and 300ng/ml, respectively, as described in Mat & Met. Can the authors explain why they used Wnt5a 3X more than Wnt3a? It is known that Wnts are very promiscuous ligands and prone to unspecific binding/responses when used at very high conc. In their Figure 4A, seems that ROR1 levels is down in HFSC Ror2^{-/-} w Wnt5a, but is slightly upregulated in HFSC Ror2^{-/-} w Wnt3a. Is this a consistent observation or just a WB artifact? Since the authors postulated that in HFSCs ROR2KO have downregulated b-catenin signaling, then it is a very straight Fw experiment to look at the b-catenin levels in a type of experiment as done in Figure 4a, and directly observe how p/b-catenin is modulated, along with Axin and p/GSK3beta levels. This would be even more informative as whether Wnt3a or Wnt5a could induce (or not) stronger/weaker b-catenin responses in this context.

6. In the context of the role of ROR2 in the activation of ATM/ATR-dependent DNA damage response in HFSCs, have the authors looked at the possible link between cyclin D1 (which they showed is downregulated in ROR2 cKO cells in Figure 3d), GSK3beta and ATM/ATR? GSK3beta is a kinase that phosphorylates both, cyclin D1 and b-catenin, in a context-dependent manner, and this could link better b-catenin activation and ATM/ATR responses.

7. In the last part of the ms, the authors looked at the b-catenin-null HFSCs and investigated the role of ROR2 in this context. ROR2 was upregulated in b-catenin cKO HFSCs at late telogen and in aged HF. This is actually a very interesting observation, and as the previous rev. 2 pointed out, the authors could have offered more mechanistic insights into this aspect. The authors should have (at least in part), explain how does Wnt-ROR2 pathway sustain stemness niche in the normal HF development, and then in b-catenin-null HFSCs. The last cartoon picture (7e) depicts ROR2 as a very powerful receptor involved in almost every intracellular pathway, which is probably not the case, as many of these pathways are indirectly activated by upstream/downstream signaling molecules.

8. The amount of PKC inhibitor GF109203X used was 5uM and as authors indicated, for 24h. This is quite high conc. prone to unspecific responses, especially for an ATP-competitive type of inhibitor. Usually 1uM max should be used, therefore is highly probable that off-target inhibition took place and many pathways were then indirectly affected/inhibited.

Minor points:

- more updated references should be used
- WB should be improved significantly, as many blots have signal almost at background level
- discussion should also focus more on how ROR2 could act specifically within canonical and non-canonical signaling, and not just a general hypothesis.

Responses to Reviewers on manuscript NCOMMS-19-20272A-Z

Overall

We appreciated two of reviewers who accept our revised manuscript and the 4th reviewers' comments as this reviewer raised important questions about ROR2-dependent regulation within the Wnt pathways.

These comments are constructive and helpful. The major concerns are: (1) The expression of Wnt3a and Wnt5a in HFSC niche, and their relation to ROR2; (2) The expression of other Wnt co-receptors, e.g. ROR1 and LRP5/6, and their level changes upon *Ror2* depletion; (3) The ROR1 and ROR2 antibody choices; (4) The concentration of Wnt5a and small molecule inhibitors used in our experiments. We've successfully addressed each suggestion, and we provide new data sets and better quality of western blots to support our findings. To summarize:

(1) The expression of Wnt3a and Wnt5a in HFSC niche, and their relation to ROR2:

According to the published microarray data (**fig.1**; Greco *et al.*, *Cell Stem Cell* 2009; Lien *et al.*, *Cell Stem Cell* 2011; Hsu *et al.*, *Cell* 2011) and *in situ* hybridization analysis (**fig2**; Lim *et al.*, *PNAS* 2016), several canonical and non-canonical Wnt ligands are secreted by either HFSCs themselves, the neighboring K6+ bulge cells, or the underlying dermal papillae (DP) cells during telogen and at anagen onset. While Wnt3a is fairly expressed by HFSCs and K6+ bulge cells throughout, Wnt5a is only expressed by DP cells at late telogen/anagen onset. However, due to the presence of Wnt antagonists, SFRP1 and DKK3, in HFSC niches, Wnt signaling activity remains low during telogen until the accumulation of Wnt ligands is high enough to tip the balance toward Wnt signaling activation at telogen end/anagen onset, which is the time point when Wnt/ β -catenin signaling starts getting activated in HFSCs (Lien *et al.*, *Cell Stem Cell* 2011) and when Wnt5a expression is significantly upregulated in DP cells (Greco *et al.*, *Cell Stem Cell* 2009).

Although animal studies show that loss of ROR2 is correlated with loss of Wnt5a as the expression pattern of Wnt5a is highly similar to that of ROR2 in the developing embryo and the morphological phenotypes of *Ror2*^{-/-} and *Wnt5a*^{-/-} mice are also similar (Yamaguchi *et al.*, *Development* 1999; Takeuchi *et al.*, *Genes Cells* 2000; Oishi *et al.*, *Genes Cells* 2003), we showed that in HFSCs ROR2 expression is higher in HFSCs at anagen onset than those in telogen (Figs. 1a-b, S1a-c). Elevation in ROR2 expression level at anagen onset tends to suggest that ROR2 is upregulated to transduce Wnt signaling upon HFSC activation. Indeed, *Ror2*-depleted HFSCs show delayed activation and anagen entry (Figs. 1e-f, S1f), likely caused by downregulation of canonical Wnt signaling (Figs. 2c-d, 4a, 4c-d) and impaired HFSC migration that resulted from the deficiency in non-canonical Wnt activity (Figs. 4b, 4e, S4d). Thus, during HFSC activation, ROR2 acts not only to respond to non-canonical Wnt signaling which is significantly induced by elevated Wnt5a from DP cells, but also to correspond to canonical Wnt signaling which is prominently activated in HFSCs at anagen onset.

(2) The expression of other Wnt co-receptors, LRP5/6 and ROR1, and their level changes upon *Ror2* depletion:

In the current manuscript, we had carefully checked up the status of canonical Wnt signaling upon *Ror2* depletion and performed several western blotting to analyze the status of proteins involved in Wnt/ β -catenin signaling. As shown in Fig. 4a, without Wnt stimulation, even though *Ror2*^{-/-} HFSCs exhibit slightly increased p-LRP6^{S1490} (LRP6 activation), they showed significant reduction in p-GSK3 β ^{S9} (GSK3 β inactivation) that subsequently caused a prominent increase in p- β -catenin^{S33/37/T41} (β -catenin degradation). Even upon Wnt3a stimulation when LRP6 is fully activated, inactivated β -catenin remained high in *Ror2*^{-/-} HFSCs due to the increase in GSK3 β stability (Fig. 4a), thereby suppressing the expression of Wnt target genes in Wnt3a-treated *Ror2*^{-/-} HFSCs (Fig. 4c). Thus, upregulated LRP6 in *Ror2*^{-/-} HFSCs is not sufficient to make compensation for ROR2 loss. However, the expression of Wnt target genes could be restored by GSK3 inhibition (Fig. 4d), indicating that ROR2 could modulate Wnt/ β -catenin signaling via regulating GSK3 β stability.

Moreover, while we did not detect significant changes in ROR1 expression between *Ror2* Ctrl and cKO HFSCs *in vivo* (Figs. 1d, S1d), we noticed that in cultured HFSCs, ROR1 expression was

getting upregulated at mRNA and protein levels after a few passages of *Ror2*^{-/-} HFSC culture (Figs. 4a, 4d). However, as Wnt3a-induced β -catenin transcriptional activity and Wnt5a-induced small GTPase activation and cell migration remain significantly downregulated in *Ror2*^{-/-} HFSCs (Fig. 4), we believe that the upregulated ROR1 in cultured *Ror2*^{-/-} HFSCs was likely insufficient to compensate for Wnt-ROR2-mediated signaling regulation. This confirms the previous studies demonstrating that ROR2 and ROR1 exhibit partially redundant but also highly specific functions (Stricker et al., *Curr Top Dev Biol* 2017).

(3) The ROR1 and ROR2 antibody choices:

To verify the specificity of ROR1 (Abcam) and ROR2 (DSHB) antibodies that we used in the manuscript and compare them over ROR1 (CST) and ROR2 (CST) antibodies suggested by the reviewer, we checked on antibody recognition ability on the correlated protein sizes and their specificity among protein members using (I) HEK293T cells that transiently express V5-tagged ROR1 or ROR2, and (II) protein extracts from *Ror2*^{+/+} and *Ror2*^{-/-} HFSCs. As shown in **fig. 4** and described in Q#2 below, both ROR1 antibodies could specifically recognize ROR1 protein, but show different preferences in recognizing native non-glycosylated or fully glycosylated ROR1 protein. Nevertheless, we agree with the reviewer that we should select the antibodies that are commercially available at the present time. Thus, all data presented in the revised manuscript for ROR1 expression (WB and immunostaining in Figs. 4a, S1e) were detected using ROR1 antibody from CST. Using the similar strategy, we also demonstrated that both ROR2 antibodies successfully recognize ROR2 protein at the correlated size without non-specifically picking up ROR1 protein (shown in **fig. 5**, described in Q#3 below). Given that ROR2 antibody from DSHB is also commercially available, we keep the data generated from this antibody throughout the manuscript.

(4) The concentration of Wnt5a and small molecule inhibitors used in our experiments:

The concentration of Wnt5a (300 ng/ml) selected for our experiments was determined based on the sensitivity of HFSCs to Wnt5a stimulation, verified based on the downstream signaling activity and the antagonistic ability to Wnt3a-induced gene expression. The downstream signaling activity of Wnt5a was judged by Dvl2 phosphorylation and Wnt5a-induced ROR2 internalization (O'Connell *et al.*, *Oncogene* 2010; Hanaki et al, 2012). It has been shown that Wnt5a binds to its receptors, Fzd and ROR1/2, and induces the internalization of receptors in a clathrin-mediated manner (Sata *et al.*, *EMBO J* 2010; Kikuchi *et al.*, *Acta Physiol* 2012). As shown in **fig. 6**, 300 ng/ml, but not 100 ng/ml, of Wnt5a could induce full Dvl2 phosphorylation and trigger ROR2 internalization at 60 minutes. Moreover, 300 ng/ml Wnt5a could bring down Wnt3a-induced gene expression (*Axin2* and *Lgr6*) up to 50% in control HFSCs, but not *Ror2*-depleted HFSCs (**fig. 7**), indicating that 300 ng/ml Wnt5a is sufficient to antagonize canonical Wnt activation in a ROR2-dependent manner, but not too high to completely erase Wnt/ β -catenin transcriptional activity.

As for small molecule inhibitors (CHIR99021-GSK3 inhibitor and GF109203X-PKC inhibitor), we took the reviewer's suggestions and used 1 μ M concentration to avoid off-target effects, especially on the signaling of our concerns. As shown in the current manuscript, treatment with 1 μ M GSK3 inhibitor increased the total pool of β -catenin (Fig. S4c) and elevated the expression of Wnt target genes (Fig. 4d), as well as increased stability of NRF2 (Fig. 5j) in HFSCs, indicating the efficiency of GSK3 inhibitor. Similarly, treatment with 1 μ M PKC inhibitor reduced PKC phosphorylation and total protein levels, which in turn suppressed phosphorylation of GSK3 β at Ser9 in control HFSCs to the similarly low level as those in *Ror2*^{-/-} HFSCs (Fig. 7c). Our experiments validate the efficacy of 1 μ M small molecule inhibitors on blocking protein activities and downstream signaling.

Below, we describe in greater detail the changes made in response to reviewer's point. We'd like to take this opportunity to thank the reviewers for their very helpful comments, which have been valuable in strengthening the impact of our study and raising its level to be of interest to future *Nature Communications* readers.

Reviewer #1:

The authors responded comments with additional data. Figs are beautiful. I do not have additional comments.

We thank this reviewer for the constructive comments, and we are pleased to know that our additional data sets were sufficient to address all comments raised by this reviewer.

Reviewer #2:

The authors have fully addressed my questions. Thus, I recommend publication of this manuscript in Nature Communications.

We are pleased to know that we addressed all questions asked by this reviewer, and we thank this reviewer for the recommendation.

Reviewer #4:

The study of Lien et al focused on the novel role of ROR2 in hair follicle Stem Cell Self-Renewal and Maintenance. The authors performed quite a diverse array of experiments and also put significant effort in improving the ms following first round reviewers comments. Although, the authors performed many experiments using in vivo models to delineate the role of ROR2 in the biology of HF's stem cell niche, some caveats need to be addressed. The molecular mechanisms of ROR2 signaling in the context of stemness gene expression, self-renewal, proliferation and migration of HFSCs should be better delineated and placed into a proper context within the Wnt pathway. ROR2 is known to belong to the non-canonical Wnt signaling, therefore its functional role in the canonical, b-catenin dependent Wnt signaling should be more carefully described.

Major points:

1. The biological role of Wnt ligand binding and activation of ROR2 signaling. What is the level of Wnt5a and Wnt3a ligands, which are described by the authors as putative Wnt-binding ligands of ROR2 during HF's development. Does the loss of ROR2 correlated with loss of Wnt ligands (Wnt5a and/or Wnt3a). This would help to clarify whether Wnt ligand binding is actually required for the biological effect of ROR2 or whether ROR2 alone is sufficient to sustain the HF stemness niche. It would also help to delineate the fate of canonical vs non-canonical Wnt pathway during HF's development, as Wnt5a is conventionally a non-canonical Wnt ligand, whereas Wnt3a is a canonical Wnt ligand. As Wnts are secreted ligands that act in autocrine or paracrine fashion, it would be informative to know whether HFSCs are Wnt-positive or other cells, like the neighboring cells in the bulge. It would be helpful for instance, to check whether primary HFSCs cultured with fibroblast feeder cells exhibited enhanced proliferation and colony formation in the presence of Wnt ligand compared to Wnt-free media.

We thank the reviewer for these constructive comments, which offer us an opportunity to well explain the nature of Wnt signaling in hair follicle homeostasis and HFSC activation. Below we address to reviewer's questions point-by-point:

(1) The expression patterns of Wnt ligands and receptors in HFSC niche:

According to the published microarray data (Greco *et al.*, *Cell Stem Cell* 2009; Lien *et al.*, *Cell Stem Cell* 2011; Hsu *et al.*, *Cell* 2011), **Wnt receptors**, including *Fzd1-3, 5-8*, and co-receptors, *Lrp5* and *Lrp6*, *Ror2*, *Ror1*, are present in HFSCs, and several **Wnt ligands**, including canonical and non-

canonical Wnt ligands, are secreted by either HFSCs themselves, the neighboring K6+ bulge cells, or the underlying dermal papillae (DP) cells during telogen phase (Tel) and at anagen onset (Ana) (**fig. 1**). Furthermore, the presence of Wnt ligands in bulge at early telogen (P43) and late telogen/anagen onset (P69) were also confirmed by *in situ* hybridization analysis for mRNA expression of Wnt genes (**fig. 2**; adapted from Lim *et al.*, *PNAS* 2016, Figs. 3, S3). Notably, while canonical Wnt ligands, e.g. Wnt3 or Wnt3a, are fairly expressed in HFSC niche, Wnt5a is not expressed by HFSCs or K6+ bulge cells, but its expression is greatly induced in DP cells at late telogen/anagen onset (**fig. 1**, red arrow), suggesting the activation of Wnt5a-induced signaling in HFSCs likely occurs in a paracrine fashion. Together, these combined data demonstrate that HFSCs are residing in the Wnt-rich microenvironment; however, due to the presence of **Wnt antagonists**, SFRP1 and DKK3, in HFSC niches (secreted by HFSCs and K6+ bulge cells), Wnt signaling activity remains low during telogen until the accumulation of Wnt ligands is high enough to tip the balance toward Wnt signaling activation at telogen end/anagen onset, which is the time point when canonical Wnt signaling starts getting activated in HFSCs (Lien *et al.*, *Cell Stem Cell* 2011), and when Wnt5a expression is significantly upregulated in DP cells (Greco *et al.*, *Cell Stem Cell* 2009).

(2) The relation of Wnt(s) to ROR2:

ROR2 was initially recognized as a Wnt receptor due to its ability to interact with Wnt ligands (Xu and Nusse, *Curr Biol* 1998; Masiakowski and Yancopoulos, *Curr Biol* 1998). Animal studies show that the expression pattern of Wnt5a is highly similar to that of ROR2 in the developing embryo, and that the morphological phenotypes of *Ror2*^{-/-} and *Wnt5a*^{-/-} mice are also similar; both mutant newborns exhibit dwarfism, facial abnormalities, short limbs and tails, and respiratory dysfunction, and died shortly after birth (Yamaguchi *et al.*, *Development* 1999; Takeuchi *et al.*, *Genes Cells* 2000; Oishi *et al.*, *Genes Cells* 2003). Moreover, Wnt5a and ROR2 function together to regulate cell migration (Nishita *et al.*, *J Cell Biol* 2006; Yamamoto *et al.*, *Genes Cells* 2007; Nomachi *et al.*, *J Biol Chem* 2008). Thus, loss of ROR2 is believed to be correlated with loss of Wnt5a in general. However, in HFSCs, we showed that ROR2 expression is higher in HFSCs at anagen onset than those in telogen (Figs. 1a-b, S1a-c). Elevation in ROR2 expression level at anagen onset tends to suggest that ROR2 is upregulated to transduce Wnt

signaling upon HFSC activation. Indeed, *Ror2*-depleted HFSCs show delayed activation and anagen entry (Figs. 1e-f, S1f), likely caused by downregulation of canonical Wnt signaling (Figs. 2c-d, 4a, 4c-d) and impaired HFSC migration that resulted from the deficiency in non-canonical Wnt activity (Figs. 4b, 4e, S4d). Thus, during HFSC activation, ROR2 acts not only to respond to non-canonical Wnt signaling which is significantly induced by elevated Wnt5a from DP cells, but also to correspond to canonical Wnt signaling which is prominently activated in HFSCs at anagen onset.

(3) The effect of Wnt(s) on HFSC proliferation:

Primary HFSCs were initially co-cultured with fibroblast feeder cells that secrete several growth factors required for HFSC growth, including Wnt ligands. When HFSCs were freshly FACS-purified from mouse skin, their self-renewal/proliferation ability could be analyzed by colony formation assay in which feeder cells are provided. Without feeder cells and regular culture medium (containing 15% FBS), primary culture of HFSCs could not be established. However, once HFSCs adapted to culture condition (>10 passages), feeder cells could be gradually removed from HFSC culture. *Ror2*^{+/+} and *Ror2*^{-/-} HFSCs that we established and used for analyses were later passages of HFSCs growing without feeder cells. To analyze the effect of Wnt ligands on HFSC proliferation, we performed a cell growth assay (MTT) with off-feeder HFSCs growing in starvation medium (containing 1% FBS; please note: HFSCs could not survive in medium without any FBS), or starvation medium supplemented with 100 ng/ml Wnt3a or 300 ng/ml Wnt5a for 2 days. As shown in **fig. 3**, *Ror2*^{+/+} HFSCs cultured in the medium containing Wnt3a show higher proliferative rate than those without Wnt (Mock) or with Wnt5a. Interestingly, Wnt3a-enhanced HFSC proliferation was abolished in HFSCs depleted for *Ror2*, indicating the dependency of ROR2 in Wnt3a-induced HFSC proliferation. This result supports the previous study showing that activation of canonical Wnt signaling in HFSCs at anagen onset is required for HFSC self-renewal/proliferation (Lien *et al.*, *Cell Stem Cell* 2011), and our findings that loss of *Ror2*, which led to downregulation of canonical Wnt signaling, compromised HFSC self-renewal/proliferation ability (Figs. 2-4). Although we cannot rule out the possibility that ROR2 could also act independently of Wnt for other cellular functions, e.g. cell-cell adhesion (our unpublished data), ROR2 is definitely essential for HFSC activation and self-renewal by modulating Wnt/ β -catenin-dependent transcriptional activation via regulating GSK3 β stability (Figs. 2, 4).

fig. 3 HFSC proliferation in the presence of Wnt ligands.

2. The status of ROR1, the other ROR receptor, should be better addressed. It is known that ROR1/2 homo- and hetero-dimerize and in many instances, loss of one receptor leads to upregulation of the other one that compensates for the signaling. In WB, authors used a ROR1 antibody from Abcam that is discontinued and had poor overall reviews. Very good ROR1 antibodies are currently available from CSTechnologies, please use commercially up to date reagents. Of note, the overall quality of WB is quite poor, and the signal looks almost as if it is a background band. In Figure 1d looking at the ROR1 mRNA expression, from 5 data points, 2 are very high, pointing to an upregulation and 3 are quite low, giving an average that was interpreted as no changes in ROR1 levels. This is not accurate. Also, in Figure 4f, authors could not get statistical significance for ROR1 levels either. The overall level of ROR1 should be investigated more carefully, in WB and IF, using up to date functional reagents.

We appreciated the reviewer's comments, which allow us to double check on protein expression using the suggested antibodies. Below we address the reviewer's questions point-by-point:

(1) ROR1 antibody specificity:

In order to verify the specificity of antibodies against ROR1 from Abcam (we used to detect ROR1 in the previous manuscript) and from Cell Signaling Technologies (CST, suggested by the reviewer), we used HEK293T cells that transiently express V5-tagged ROR1 or ROR2 individually to verify their

specificity and the correlated protein sizes. As shown in **fig. 4A**, both ROR1 antibodies (Abcam and CST) could recognize overexpressed V5-ROR1 in two different sizes; the lower band (~105-110 kDa) considered to represent the native non-glycosylated ROR1, and the upper band (~130 kDa) supposed to represent the fully glycosylated monomer (Daneshmanesh et al., *Int J Cancer* 2008; Kaucká et al., *Acta Physiol* 2011). Interestingly, the ROR1 (Abcam) antibody preferentially recognizes the endogenous ROR1 in glycosylated form (upper band in V5-ROR2 lane at the left panel, **fig. 4A**), whereas ROR1 (CST) antibody preferentially recognizes the native non-glycosylated ROR1 protein (lower band in V5-ROR2 lane at the right panel, **fig. 4A**).

This particularity could also be seen in WB in which protein extracts from *Ror2*^{+/+} and *Ror2*^{-/-} HFSCs were analyzed for ROR1 expression (**fig. 4B**). In **fig. 4B**, the size of ROR1 bands is slightly higher in the blot detected by ROR1 (Abcam) antibody than those in the blot detected by ROR1 (CST) antibody. As a result, these data indicate that both ROR1 antibodies could specifically recognize ROR1 protein, but show different preferences in recognizing native non-glycosylated or fully glycosylated ROR1 protein. Nevertheless, we agree with the reviewer that we should select the antibodies that are commercially available at the present time. Thus, all data presented in the revised manuscript for ROR1 expression (WB and immunostaining in Figs. 4a, S1e) were detected using ROR1 antibody from CST.

(2) Quality of WB overall:

We agree with the reviewer about the poor quality of our WB in the previous manuscript. Indeed, several of WBs were not in the best quality; however, for those samples from FACS-sorted mouse cells, it has been challenging to obtain more cells for WB analysis. To compensate this limitation, we also provided additional data sets, including RT-qPCR for mRNA expression and immunostaining for protein expression, to double confirm the WB results. For example, WB data in **Fig. 1b** is also confirmed by immunostaining shown in **Figs. 1a** and **S1b**; the WB result at **Fig. S1d** is also validated with RT-qPCR and the immunostaining data shown in **Fig. 1d** and **S1e**. Nevertheless, we took the reviewer's comments into account and re-performed WB analyses with independent sets of cultured HFSCs that were transduced with lentiviral Cre. The new sets of data shown in **Fig. 4a** not only significantly improved the quality (can be appreciated from the full-scan images in **Fig. S11**), but also confirmed what we had presented in the previous manuscript, as well as provided additional information regarding the status of proteins involved in canonical Wnt signaling.

(3) *Ror1* mRNA expression (**Figure 1d & Fig. 4f**):

We understand the concern of the reviewer. To verify the expression level of ROR1 in *Ror2*-depleted cells *in vivo*, we performed RT-qPCR for *Ror1* mRNA expression with an additional set of HFSCs at late telogen (Late Tel, **Fig. 1d**). In addition, we also performed immunofluorescence staining with the whole-mount skin for ROR1 protein expression (now in **Fig. S1d**). Both RT-PCR analysis and the quantification of ROR1 staining indicate no significant change in ROR1 expression between *Ror2* Ctrl and cKO HFSCs *in vivo*. However, we noticed that in cultured HFSCs, ROR1 expression was getting upregulated at mRNA and protein levels after a few passages of *Ror2*^{-/-} HFSC culture (**Figs. 4a, 4d**). Interestingly, the upregulated ROR1 seemingly responded to Wnt5a in the same trend as ROR2. As seen in **Fig. 4a**, the ROR2 bands were shifted up and decreased in Wnt5a-treated *Ror2*^{+/+} HFSCs, and these changes were due to Wnt5a-induced phosphorylation and internalization (please also see **Q#5** for the detail; Yamamoto et al., *Genes Cells* 2007; Liu et al., *J Cell Biochem* 2008; O'Connell et al., *Oncogene* 2010; Hanaki et al., 2012; Sata et al., *EMBO J* 2010; Kikuchi et al., *Acta Physiol* 2012). Similar changes were also found in the upregulated ROR1 in *Ror2*^{-/-} HFSCs, where the upregulated ROR1

bands were shifted up and decreased in Wnt5a-treated *Ror2*^{-/-} HFSCs. ROR1 upregulation in *Ror2*-depleted HFSCs was thought to compensate for ROR2 loss; however, even if any compensation would have been made, the upregulated ROR1 in cultured *Ror2*^{-/-} HFSCs was seemingly insufficient to compensate for Wnt-ROR2-mediated signaling regulation as Wnt3a-induced β -catenin transcriptional activity and Wnt5a-induced small GTPase activation and cell migration remain significantly downregulated in *Ror2*^{-/-} HFSCs (Fig. 4). This confirms the previous studies demonstrating that ROR2 and ROR1 exhibit partially redundant but also highly specific functions (Stricker et al., *Curr Top Dev Biol* 2017).

3. Whenever possible, could the authors use a ROR2 antibody that is commercially available, for instance from CST, to validate some of the results from their WB. That is because the WB signal looks rather weak, almost equal to background in many of their WB. This would substantiate their findings. For instance, in their Figure S1c and S1d, there seems to be a strong band at 80kDa, which would correspond to non-glycosylated ROR2 isoform.

We thank the reviewer for this comment. To validate ROR2 antibodies from Developmental Studies Hybridoma Bank (DSHB) that we used to detect ROR2 in our manuscript and compare it to the one from CST that is suggested by the reviewer, we performed the similar experiments as described in Q#2. In doing so, we transiently expressed V5-tagged ROR1 or ROR2 in HEK293T cells and tested on antibody efficiency and correlated protein sizes. As shown in **fig. 5A**, both ROR2 antibodies could recognize overexpressed V5-ROR2 protein (the thick bands in V5-ROR2 lane) as well as the endogenous ROR2 (the thin bands in V5-ROR1 lane) in the same size (~120 kDa). To examine the antibody specificity, we used them on the WB containing protein extracts from *Ror2*^{+/+} and *Ror2*^{-/-} HFSCs. As shown in **fig. 5B**, both ROR2 antibodies successfully recognize ROR2 protein at the correlated size without non-specifically picking up ROR1 protein as we did not detect any band present in the lane of *Ror2*^{-/-} HFSCs. Given that ROR2 antibody from DSHB is also commercially available, we keep all the data generated from this antibody throughout the manuscript.

The WBs shown in Figs. S1c and S1d were performed with protein extracts purified from FACS-sorted HFSCs, from which we could only extract a little amount of proteins. In this case, using more cells to get a better signal would be very challenging as described in Q#2. Nevertheless, we had performed 4 independent sets of WBs (quantification data is presented in Fig. 1b), and we presented two of these blots shown in Fig. 1b and S1c. In addition, we also performed immunostaining analysis for ROR2 shown in Figs. 1a, S1b and S1e, as well as RT-qPCR analysis for *Ror2* mRNA expression shown in Fig. 1d to double confirm our findings. All the data sets demonstrate the expression levels of ROR2 is higher in activated HFSCs at anagen onset (Figs. 1a, 1b, S1b, S1c) and confirm the efficiency of *Ror2* depletion (Figs. 1d, S1d, S1e). As for a band at 80 kDa in Figs. S1d, we believe it is a non-specific band as we did not detect this band in other WBs for ROR2 (e.g. Figs. 1 b, 3d, 4a, 5g, and fig 4B shown above). Non-glycosylated ROR2 should be in the size near 105 kDa (943 a.a.), but not lower than this size unless the band is a degraded form.

4. The role of ROR2 in b-catenin-dependent activation should be carefully discussed and delineated. As authors pointed out, studies showing that ROR2 “might” activate b-catenin signaling via binding to Wnt3a are not very solid, as these experiments were done in cancer cells using over-expression systems (ref. 29 and 33 in ms). For instance, ref. 33, this is a very old study, yet pointed out that ROR2

inhibited Wnt3a function in b-catenin reporter activity levels. It is already known that Wnt3a could bind to and transduce signals via ROR1/2 during development and act in a b-catenin independent manner. For instance, Wnt3a, which is a representative of canonical Wnt, can induce neural differentiation in a b-catenin-independent manner by activating JNK and ATF-2 possibly through a receptor complex, comprising of Ror1, Fzd4/5, and LRP6 (Bengoa-Vergniory et al., 2016). Authors should not base their conclusion entirely on ROR2 expression when talking about Wnt3a/b-catenin signaling, since they have not investigated the status of other Wnt- receptors (Fzd, LRP5/6, etc) that could be equally relevant in this context.

We appreciate the reviewer's comment on ROR2-mediated regulation in Wnt3a-induced Wnt/ β -catenin signaling. In the current manuscript, we had performed several WBs to analyze the status of proteins involved in canonical Wnt signaling activity, including Lrp6, Axin1, GSK3 β and β -catenin (right panel, Fig. 4a). As shown, *Ror2*^{-/-} HFSCs exhibited increased p-LRP6^{S1490} (LRP6 activation), but showed lower levels of Axin1 and significant reduction in p-GSK3 β ^{S9} (GSK3 β inactivation), which in turn caused an increase in p- β -catenin^{S33/37/T41} (β -catenin degradation). The consequence of increased β -catenin degradation was confirmed by significant reduction in Wnt3a-induced expression of canonical Wnt target genes in *Ror2*^{-/-} HFSCs (Fig. 4c). Inhibition of GSK3 β restored the expression of Wnt target genes in *Ror2*^{-/-} HFSCs (Fig. 4d), further demonstrating that ROR2 modulates Wnt3a-induced canonical Wnt signaling via regulating GSK3 β stability.

As described in Q#1, HFSCs express several Wnt receptors/co-receptors and reside in Wnt-rich microenvironment (**fig. 1**) together with high levels of Wnt antagonists (SFRP1 and DKK3). The balance between the presence of Wnt antagonists and the accumulation of Wnt ligands in HFSC niche determines the time point for Wnt/ β -catenin signaling activation, which is demonstrated essential for HFSC activation and self-renewal (Lien et al., Cell stem cell 2011; Choi et al., Cell Stem Cell 2013). Thus, the compromised Wnt/ β -catenin signaling activity is likely the cause leading to delayed HFSC activation in *Ror2* cKO HFSCs and reduced cell proliferation in *Ror2*^{-/-} HFSCs. Even if other Wnt co-receptors (e.g. ROR1, LRP6) are upregulated in *Ror2*^{-/-} HFSCs (Fig. 4a), they are insufficient to compensate for the loss of ROR2 in regulating Wnt-activated signaling in *Ror2*-depleted HFSCs.

5. The conc. of Wnt3a and Wnt5a used for stimulations in Figure 4 are quite different, 100ng/ml and 300ng/ml, respectively, as described in Mat &Met. Can the authors explain why they used Wnt5a 3X more than Wnt3a? Is known that Wnts are very promiscuous ligands and prone to unspecific binding/responses when used at very high conc. In their Figure 4A, seems that ROR1 levels is down in HFSC *Ror2*^{-/-} w Wnt5a, but is slightly upregulated in HFSC *Ror2*^{-/-} w Wnt3a. Is this a consistent observation or just a WB artifact? Since the authors postulated that in HFSCs ROR2KO have downregulated b-catenin signaling, then it is a very straight Fw experiment to look at the b-catenin levels in a type of experiment as done in Figure 4a, and directly observe how p/b-catenin is modulated, along with Axin and p/GSK3beta levels. This would be even more informative as whether Wnt3a or Wnt5a could induce (or not) stronger/weaker b-catenin responses in this context.

We thank the reviewer for these critical questions. We address these questions point-by-point as described below:

(1) The concentration of Wnt ligands:

The concentration of Wnt ligands selected for experiments should be determined based on the cell types (cell sensitivity to Wnt stimulation) and the efficiency of signaling induction by recombinant Wnt ligands (ability to trigger downstream signaling or cellular effects). The reasons of using 300 ng/ml of Wnt5a, instead of 100 ng/ml, for our experiments were:

(I) The signaling induction efficiency of Wnt5a was judged by Dvl2 phosphorylation and Wnt5a-induced ROR2 internalization (O'Connell *et al.*, *Oncogene* 2010; Hanaki *et al.*, 2012). It has been shown that Wnt5a binds to its receptors, Fzd and ROR1/2, and induces the internalization of receptors in a clathrin-mediated manner (Sata *et al.*, *EMBO J* 2010; Kikuchi *et al.*, *Acta Physiol* 2012). As shown in **fig. 6**, HFSCs treated with 100 ng/ml Wnt5a (left panel) show inefficient activation in Dvl2 phosphorylation in control HFSCs (remain as low as those in *Ror2*^{-/-} HFSCs) and display no ROR2 internalization (no decrease in ROR2 level over the course of time). However, treatment of 300 ng/ml Wnt5a induced full Dvl2 phosphorylation at 60 minutes and sufficiently triggered Wnt5a-induced ROR2 internalization (*Ror2*^{+/+}, right panel, fig. 6). Thus, we believe that 300 ng/ml, but not 100 ng/ml, of recombinant Wnt5a is a better selection to induce downstream activities of HFSCs.

(II) According to the manufacture instruction (R&D), the activity of recombinant Wnt5a is measured by its ability to inhibit Wnt3a-induced alkaline phosphatase production of mouse preosteoblast cells, for which the median effective dose (ED₅₀) is 100-500 ng/ml in the presence of 5 ng/ml of Wnt3a (Wnt5a concentration is 20-100 times higher than Wnt3a). Since the optimal concentration for different cell types and applications would vary, we tested the cultured HFSCs for Wnt5a-mediated antagonization of Wnt3a-induced β-catenin transcriptional activity by measuring the gene expression level upon treatment of 100 ng/ml Wnt3a in absence or presence of 300 ng/ml Wnt5a. As shown in **fig. 7**, 300 ng/ml Wnt5a could bring down Wnt3a-induced gene expression (*Axin2* and *Lgr6*) up to 50% in control HFSCs, but not *Ror2*-depleted HFSCs, indicating that 300 ng/ml Wnt5a is sufficient to antagonize canonical Wnt activation in a ROR2-dependent manner, but not too high to completely erase Wnt/β-catenin transcriptional activity. Thus, we believe that using 300 ng/ml of Wnt5a in our experimental setup is an appropriate concentration, which is high enough to trigger full signaling activation, but not too high to jeopardize the cellular effects.

(2) Expression levels of ROR1 in the presence of Wnt ligands (Fig. 4a):

As shown in Figs. 4a and 4d, ROR1 mRNA and protein expression was upregulated in cultured *Ror2*^{-/-} HFSCs; however, the upregulated ROR1 in cultured *Ror2*^{-/-} HFSCs was seemingly insufficient to compensate for ROR2 loss as the deficiencies in Wnt signaling activity and cellular functions remain observed (please also see Q#2 for the detail). It has been demonstrated that Wnt5a-binding induces phosphorylation of ROR2 (Yamamoto *et al.*, *Genes Cells* 2007; Liu *et al.*, *J Cell Biochem* 2008) and ROR2 internalization in a clathrin-mediated manner (Sata *et al.*, *EMBO J* 2010; Kikuchi *et al.*, *Acta Physiol* 2012). This phenomenon was also observed in Wnt5a-treated *Ror2*^{+/+} HFSCs showing up-shifted ROR2 band and the reduction in ROR2 protein levels over the time course of Wnt5a treatment (left panel, Fig. 4a). Interestingly, we also observed these changes in the upregulated ROR1 in *Ror2*^{-/-} HFSCs. As shown in Fig. 4a, the upregulated ROR1 bands were up-shifted and decreased in Wnt5a-treated *Ror2*^{-/-} HFSCs. These modifications have never been detected for either ROR2 or ROR1 upon Wnt3a stimulation; instead, Wnt3a treatment slightly increased expression of ROR2 and ROR1. Since

we have constantly observed these alterations for both ROR2 and ROR1, these results are consistent observations, not WB artifacts.

(3) How Axin1, p-GSK3 β and p- β -catenin are modulated in *Ror2*^{-/-} HFSCs upon Wnt treatment:

Taking reviewer's suggestions into an account, we had performed several WBs to analyze the status of proteins involved in canonical Wnt signaling activity, including Lrp6, Axin1, GSK3 β and β -catenin (right panel, Fig. 4a). As shown, without Wnt stimulation, *Ror2*^{-/-} HFSCs exhibited increased p-LRP6^{S1490} (LRP6 activation), but showed lower levels of Axin1 and significant reduction in p-GSK3 β ^{S9} (GSK3 β inactivation), which in turn caused an increase in p- β -catenin^{S33/37/T41} (β -catenin degradation). These results suggest that β -catenin-dependent signaling would be compromised due to unleashed GSK3 β activity in *Ror2*-depleted HFSCs. Indeed, even upon Wnt3a stimulation when LRP6 is fully activated, inactivated β -catenin remained high in *Ror2*^{-/-} HFSCs due to the increase in GSK3 β stability (Fig. 4a). The suppression of Wnt/ β -catenin signaling in *Ror2*-depleted HFSCs could be evidenced by lower expression of Wnt target genes in Wnt3a-treated *Ror2*^{-/-} HFSCs (Fig. 4c), and this suppression could be restored by GSK3 inhibition (Fig. 4d). Similar expression patterns of GSK3 β and β -catenin were also detected in Wnt5a-treated *Ror2*^{-/-} HFSCs where LRP6 and Axin1 showed moderate changes, suggesting the universal role of ROR2 in suppressing GSK3 β . Altogether, our data provide the mechanism of ROR2 in modulating Wnt/ β -catenin signaling via regulating GSK3 β stability.

6. In the context of the role of ROR2 in the activation of ATM/ATR-dependent DNA damage response in HFSCs, have the authors looked at the possible link between cyclin D1 (which they showed is downregulated in ROR2 cKO cells in Figure 3d), GSK3beta and ATM/ATR? GSK3beta is a kinase that phosphorylates both, cyclin D1 and b-catenin, in a context-dependent manner, and this could link better b-catenin activation and ATM/ATR responses.

We greatly appreciated the reviewer for this suggestion. Indeed, GSK3 β plays a critical role in ROR2-mediated regulation in Wnt/ β -catenin signaling activation and cellular redox balance. In the current manuscript, we showed that upon *Ror2* loss GSK3 β stability was significantly elevated, which in turn suppressed Wnt/ β -catenin transcriptional activity (Figs. 2c, 2d, 4a, 4c, 4d), cyclin D1 expression (Figs. 3d, 4f), and NRF2 activity (Fig. 5j), and subsequently leading to downregulation of canonical Wnt signaling activity and imbalanced oxidative response. Although unleashed GSK3 β activity is unlikely to be the only cause responsible for the deficiency of ATM/ATR-dependent DNA damage response in *Ror2*^{-/-} HFSCs, constantly imbalanced oxidative response surely could result in accumulation of DNA damage, which would increase the burden of DNA repair machinery and halt cell proliferation. While we do not have a direct evidence on how ROR2 directly regulates ATM/ATR-dependent DDR, deregulated GSK3 β activity in *Ror2*^{-/-} HFSCs could be the primary cause that attributed to the phenotypes that we detected in *Ror2*-depleted HFSCs.

7. In the last part of the ms, the authors looked at the b-catenin-null HFSCs and investigated the role of ROR2 in this context. ROR2 was upregulated in b-catenin cKO HFSCs at late telogen and in aged HF. This is actually a very interesting observation, and as the previous rev. 2 pointed out, the authors could have offered more mechanistic insights into this aspect. The authors should have (at least in part), explain how does Wnt-ROR2 pathway sustain stemness niche in the normal HF development, and then in b-catenin-null HFSCs. The last cartoon picture (7e) depicts ROR2 as a very powerful receptor involved in almost every intracellular pathway, which is probably not the case, as many of these pathways are indirectly activated by upstream/downstream signaling molecules.

We thank the reviewer for raising up this question. In our manuscript, we show that without ROR2, HFSCs are unable to be maintained for long-term in culture where HFSCs are constantly in proliferation, which makes HFSCs vulnerable to DNA damages (Fig. 3). In contrast, although a population of *Ror2*-depleted HFSCs *in vivo* got lost after one round of HF regeneration, the recovered *Ror2* cKO HFSCs were able to sustain and replenish HF lineage in the following regeneration (Figs. 2g, S3a). This discrepancy in HFSC survival is due to the differences in the frequency of HFSC proliferation (described

in the Discussion section). Thus, the quiescence of HFSCs, in a way, protects them from exhaustion as long as they are maintained in their native niche. This is definitely a case for β -cat cKO HFSCs, which remain quiescent and maintained in their niche throughout the rest of life unless they are enforced to get activated (e.g. by depilation).

Given that both *Ror2* depletion and PKC inhibition resulted in an increase in GSK3 β stability (Figs. 4a, 7c) and caused β -cat cKO HFSCs undergoing wrong fate differentiation, we speculate that upregulated ROR2 and PKC in β -cat cKO HFSCs, which potentially could inhibit GSK3 β activity, could maintain β -cat cKO HFSCs by preventing them undergoing activation and differentiation. GSK3 inhibition or depletion has been shown to enhance embryonic stem cell self-renewal and pluripotency via activating Wnt/ β -catenin signaling and promoting translation of pluripotency-associated transcription factors (Sato *et al.*, *Nat Med*, 2004; Ying *et al.*, *Nature*, 2008; Doble *et al.*, *Dev Cell*, 2007; Sanchez-Ripoll *et al.*, *PLoS One*, 2013). However, the mechanism by which GSK3 inhibition or loss-of-function enhances pluripotency remains not completely resolved. Although we are not certain why ROR2 and PKC expression were upregulated in β -cat cKO HFSCs, we demonstrated that depletion of *Ror2* or inhibition of PKC caused β -cat cKO HFSCs leaving quiescence and undergoing sebocyte differentiation (Figs. 6c, 7d). Thus, we could speculate that *Ror2* depletion or PKC inhibition might challenge the quiescent state of β -cat cKO HFSCs by lowering down the threshold of β -cat cKO HFSCs for activation, and activated β -cat cKO HFSCs then undergo differentiation into sebocyte fate, similar to the phenomenon of depilation on β -cat cKO HFSCs (Lien *et al.*, *Nat Cell Biol*, 2014). Together, these results suggest that upregulated ROR2-PKC signaling keeps β -cat cKO HFSCs in quiescence, which protects HFSCs against activation and subsequent differentiation, potentially through inhibiting GSK3 β .

As for the cartoon picture shown in Fig. 7f, we modified it accordingly based on the updated findings in the current manuscript. We pointed out ROR2 direct and indirect effectors that were examined in our study, and highlighted at the bottom of the picture the involved functions that were also determined in the manuscript. Given that the purpose of providing this cartoon picture is to summarize our findings, we believe that our picture is clear and giving a general view on ROR2-dependent regulation in HFSCs regardless direct or indirect effects.

8. The amount of PKC inhibitor GF109203X used was 5uM and as authors indicated, for 24h. This is quite high conc. prone to unspecific responses, especially for an ATP-competitive type of inhibitor. Usually 1uM max should be used, therefore is highly probable that off-target inhibition took place and many pathways were then indirectly affected/inhibited.

We thank the reviewer for this suggestion. In this manuscript, we used small molecular inhibitors (CHIR99021-GSK3i and GF109203X-PKCi) at 1uM concentration to avoid off-target effects, especially on the signaling of our concerns. As shown, treatment with 1uM GSK3 inhibitor elevated the expression of Wnt target genes (Fig. 4d), as well as increased stability of NRF2 (Fig. 5j) in HFSCs, indicating the efficiency of GSK3i. Similarly, treatment with 1uM PKC inhibitor efficiently reduced PKC phosphorylation and total protein levels, which in turn suppressed phosphorylation of GSK3 β at Ser9 in control HFSCs to the similarly low level as those in *Ror2*^{-/-} HFSCs (Fig. 7c). Together, our experiments validate the efficacy of 1uM small molecule inhibitors on blocking protein activities and downstream signaling.

Minor points:

1. more updated references should be used.

Thanks for the suggestion. We included several updated references throughout the revised manuscript.

2. WB should be improved significantly, as many blots have signal almost at background level.

As described in Q#2, we repeated the WB analysis with independent sets of cultured HFSCs that were transduced with lentiviral Cre (shown in Fig. 4a). We could obtain similar results as those shown in the previous manuscript, as well as provide additional data regarding proteins involved in canonical Wnt signaling. The full-scan of blots is provided in Fig. S11.

3. discussion should also focus more on how ROR2 could act specifically within canonical and non-canonical signaling, and not just a general hypothesis.

We added additional information in the 1st paragraph of the Discussion section as the following: “Further analyses with cultured HFSCs demonstrated that ROR2 not only transduces Wnt5a-activated non-canonical Wnt signaling and migration through activating PKC, JNK and small GTPases, but also is required for Wnt3a-induced expression of canonical Wnt target genes in HFSCs via regulating GSK3 β stability.” to be more specifically point out the function of ROR2 within canonical and non-canonical signaling.

REVIEWERS' COMMENTS

Reviewer #4 (Remarks to the Author):

The authors responded to all my questions/queries. The manuscript has improved significantly now and is more clear on the molecular mechanisms. The figures have also improved.